# METACoCo: A New Few-Shot Classification Benchmark with Spurious Correlation

**Min Zhang**[1]   **Haoxuan Li**[2]   **Fei Wu**[1]   **Kun Kuang**[1*]
[1]Zhejiang University   [2]Peking University
{zhangmin.milab, wufei, kunkuang}@zju.edu.cn, hxli@stu.pku.edu.cn

## ABSTRACT

Out-of-distribution (OOD) problems in few-shot classification (FSC) occur when novel classes sampled from testing distributions differ from base classes drawn from training distributions, which considerably degrades the performance of deep learning models deployed in real-world applications. Recent studies suggest that the OOD problems in FSC mainly including: (a) cross-domain few-shot classification (CD-FSC) and (b) spurious-correlation few-shot classification (SC-FSC). Specifically, CD-FSC occurs when a classifier learns transferring knowledge from base classes drawn from seen training distributions but recognizes novel classes sampled from unseen testing distributions. In contrast, SC-FSC arises when a classifier relies on non-causal features (or contexts) that happen to be correlated with the labels (or concepts) in base classes but such relationships no longer hold during the model deployment. Despite CD-FSC has been extensively studied, SC-FSC remains understudied due to lack of the corresponding evaluation benchmarks. To this end, we present **Meta Co**ncept **Co**ntext (MetaCoCo), a benchmark with spurious-correlation shifts collected from real-world scenarios. Moreover, to quantify the extent of spurious-correlation shifts of the presented MetaCoCo, we further propose a metric by using CLIP as a pre-trained vision-language model. Extensive experiments on the proposed benchmark are performed to evaluate the state-of-the-art methods in FSC, cross-domain shifts, and self-supervised learning. The experimental results show that the performance of the existing methods degrades significantly in the presence of spurious-correlation shifts. We open-source all codes of our benchmark and hope that the proposed MetaCoCo can facilitate future research on spurious-correlation shifts problems in FSC. The code is available at: https://github.com/remiMZ/MetaCoCo-ICLR24.

## 1 INTRODUCTION

Few-shot classification (FSC) aims to recognize unlabeled images (or query sets) from novel classes with only a few labeled images (or support sets) by transferring knowledge learned from base classes. Despite the impressive advances in the FSC, in real-world applications, out-of-distribution (OOD) problems in FSC occur when the novel classes sampled from testing distributions differ from the base classes drawn from training distributions, which significantly degrades the performance and robustness of deep learning models, and has gained increasing attention in recent years (Song et al., 2022; Li et al., 2023d). As shown in Figure 1, the OOD problems in FSC can be broadly categorized into two categories with different forms of distribution shifts: (a) cross-domain few-shot classification (CD-FSC) and (b) spurious-correlation few-shot classification (SC-FSC), as established by previous works (Triantafillou et al., 2020; Yue et al., 2020; Luo et al., 2021; Li et al., 2022).

**Cross-domain few-shot classification (CD-FSC).** Cross-domain shifts occur when a classifier learns transferring knowledge from base classes drawn from seen training distributions but recognizes novel classes sampled from unseen testing distributions. For example, in COVID-19 predictions, we may want to train a model on patients from a few sampled countries and then deploy the trained model to a broader set of countries. Existing OOD methods in FSC have shown considerable progress in solving the cross-domain shifts problem (Hou et al., 2019; Doersch et al.,

---

*Corresponding author.

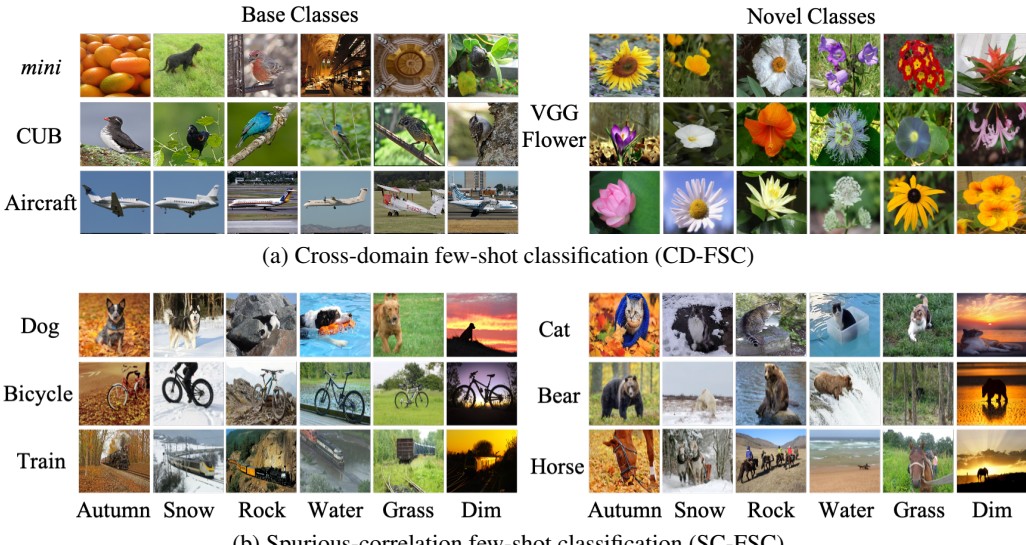

Figure 1: Example of cross-domain shifts and spurious-correlation shifts in FSC. (a) In Meta-dataset with cross-domain shifts (Triantafillou et al., 2020), the model is trained on base classes sampled from three datasets including *mini*ImageNet, CUB-200-2011 and Aircraft, then tested on novel classes drawn from VGG Flower. (b) In our proposed MetaCoCo with spurious-correlation shifts, each class (or concept, *e.g.*, dog) consists of different backgrounds (or context, *e.g.*, autumn).

2020; Guo et al., 2020; Wang & Deng, 2021; Sun et al., 2021; Liang et al., 2021; Wang & Deng, 2021; Li et al., 2023a;c; Oh et al., 2022; Zhang et al., 2020; 2022b). Meanwhile, two standard cross-domain benchmarks have been proposed to evaluate the effectiveness of these methods, *i.e.*, Meta-dataset (Triantafillou et al., 2020) consisting of 10 existing datasets, and BSCD-FSL (Guo et al., 2020) consisting of 4 existing datasets. Figure 1(a) shows the example of cross-domain shifts on Meta-dataset, where *mini* (*mini*ImageNet), CUB (CUB-200-2011) and Aircraft are used as the base classes with VGG Flower as the novel classes, with each dataset exhibits a distinct distribution.

**Spurious-correlation few-shot classification (SC-FSC).** Spurious-correlation shifts arise when a classifier relies on spurious, non-causal context features that are not essential to the true label or concept, which can significantly reduce the robustness and generalization ability of the model. In the COVID-19 example, a recent nationwide cross-sectional study found spurious correlations between long-term $PM_{2.5}$ exposure and COVID-19 deaths in the United States due to county-level socioeconomic and demographic variables as confounders (Wu et al., 2020). To this end, models trained on base classes with spurious features and evaluated on novel classes without the relationship suffer substantial drops in performance. As shown in Figure 1(b), we show the example of spurious-correlation shifts in our proposed benchmark, where each class presents a range of non-causal contexts, such as autumn or snow. Meanwhile, the concepts of the base classes and the novel classes would be distinct in the FSC problem, *e.g.*, "dog in the autumn" in the base class and "cat in the autumn" in the novel class, which emphasizes the impact of spurious correlation between concepts and contexts in the proposed benchmark. Despite the widespread of spurious-correlation shifts in the real-world FSC problems (Wang et al., 2017a; Yue et al., 2020; Luo et al., 2021; Zhang et al., 2023b), SC-FSC remains understudied due to lack of the corresponding evaluation benchmarks.

**Shortcomings of spurious-correlation shifts benchmarks in traditional machine learning.** Recently, spurious-correlation shifts in traditional machine learning (TML) have been investigated extensively (Arjovsky et al., 2019; Sagawa et al., 2019; Rosenfeld et al., 2020; Ahmed et al., 2020; Bae et al., 2021; Shen et al., 2021), and various benchmarks have been created, including toy datasets, *e.g.*, ColoredMNIST (Arjovsky et al., 2019), and real-world datasets, *e.g.*, NICO (He et al., 2021). These TML benchmarks cannot be used directly to evaluate the performance in FSC problems with spurious-correlation shifts, following the reasons below: (1) **The number of classes.** Most TML benchmarks are the binary classification problem, but for FSC problems, we need enough classes to split base and novel classes. (2) **The number of samples.** FSC needs adequate samples from base classes to learn the transferring knowledge to novel classes with a few labeled images. (3) **The num-**

**ber of contexts.** Contexts in TML benchmarks are commonly limited, but FSC with many classes requires more contexts to build stronger spurious-correlation shifts. To the best of our knowledge, there does not exist a unified study and the benchmark of spurious-correlation shifts for FSC.

In this paper, we present **Meta Concept Context** (MetaCoCo), a large-scale benchmark with a total of 175,637 images, 155 contexts and 100 classes, with spurious-correlation shifts arising from various contexts in the real-world scenarios. The basic idea of constructing spurious-correlation shifts is to label the images with the main concepts and contexts. For example, in the category with "dog" as the main concept, the images are categorized into different contexts such as "autumn", "snow", and "rock", which denotes that the "dog" is in the autumn, in the snow, or on the rock, respectively. With the help of these contexts, one can easily design a spurious-correlation-shift setting by training the model in some contexts and testing the model in other unseen contexts for studying spurious-correlation shifts as well as the unseen concepts for studying few-shot classification problems.

Furthermore, we propose a metric by using CLIP as a pre-trained vision-language model to quantify and compare the extent of spurious correlations on MetaCoCo and other FSC benchmarks. We conduct extensive experiments on MetaCoCo to evaluate the state-of-the-art methods in FSC, cross-domain shifts, and self-supervised learning. We open-source all codes for our benchmark and hope the proposed MetaCoCo will facilitate the development of spurious-correlation robust models.

## 2    COMPARISON WITH EXISTING BENCHMARKS

MetaCoCo provides a unified framework to facilitate the development of models robust to spurious-correlation shifts in FSC. We next discuss how MetaCoCo is related to existing benchmarks.

**Relation to few-shot classification benchmarks.** Few-shot classification (FSC) has attracted attention for its ability to recognize novel classes using few labeled images. Many methods have been proposed to solve the FSC problems, including (1) *Fine-tuning based methods* (Chen et al., 2019; Tian et al., 2020a; Chen et al., 2021), which address the problem by *learn to transfer*. (2) *Metric-based methods* (Vinyals et al., 2016; Snell et al., 2017; Li et al., 2019a; Zhang et al., 2022a), which solve the problem by *learn to compare*. (3) *Meta-based methods* (Finn et al., 2017; Rusu et al., 2019; Bae et al., 2021; Zhang et al., 2020), which tackle the problem by *learn to learn*.

Many FSC benchmarks have been proposed to evaluate the effectiveness of these methods, including *mini*ImageNet (Vinyals et al., 2016), Places (Zhou et al., 2017), CIFAR-FS (Bertinetto et al., 2019), Plantae (Van Horn et al., 2018), CUB-200-2011 (Wah et al., 2011), Stanford Dogs (Khosla et al., 2011), Stanford Cars (Krause et al., 2013), etc. These datasets are generally divided into training, validation and testing sets with non-overlap classes. While these datasets are useful testbeds for verifying FSC methods, they follow the independent and identically distributed (IID) assumption.

**Relation to cross-domain shifts FSC benchmarks.** Cross-domain shifts have been widely studied in the FSC community, which aims to learn the transferring knowledge from seen training distributions to recognize unseen testing distributions. Many CD-FSC methods have been proposed to address the cross-domain problem (Tseng et al., 2020; Sun et al., 2021; Liang et al., 2021; Wang & Deng, 2021; Li et al., 2022; Zhang et al., 2022b), which can be mainly divided into bi-level optimization (Tseng et al., 2020; Triantafillou et al., 2021; Li et al., 2023b; Zhang et al., 2023c), domain adversarial learning (Motiian et al., 2017; Zhao et al., 2021), adversarial data augmentation (Wang & Deng, 2021; Sun et al., 2021), and module modulation (Liu et al., 2021; Li et al., 2022). Some benchmarks have been proposed to evaluate the effectiveness of these CD-FSC methods, including Meta-dataset (Triantafillou et al., 2020) consisting of 10 existing datasets, and BSCD-FSL (Guo et al., 2020) consisting of 4 existing datasets. They usually use the leave-one-domain-out setting as the testing domain and the others as training domains. However, these benchmarks use different datasets as domains to construct cross-domain distribution shifts, causing them to fail to reflect spurious correlation shifts that occur in real-world applications (see more discussion in Appendix A).

**Relation to spurious-correlation shifts TML benchmarks.** Spurious-correlation shifts have been studied recently in traditional machine learning (TML) (Sagawa et al., 2019; Krueger et al., 2021; Yao et al., 2022; Bai et al., 2024; Tang et al., 2024). Many methods mainly focus on causal learning (Peters et al., 2015; Kuang et al., 2018; Kamath et al., 2021; Wu et al., 2022; Wang et al., 2024; Li et al., 2024; Zhu et al., 2024), invariant learning (Arjovsky et al., 2019; Chang et al., 2020; Rosenfeld et al., 2020; Huang et al., 2023), and distributionally robust optimization (Arjovsky et al., 2019),

Table 1: A summary of the existing benchmarks and our proposed spurious-correlation benchmark, *i.e.*, MetaCoCo. $\mathcal{C}$ and $\mathcal{N}$ are the number of classes and samples, respectively. The subscripts "all", "train", "val" and "test" mean the all dataset, training set, validation, and testing set, respectively.

| Dataset | $\mathcal{C}_{all}$ | $\mathcal{C}_{train}$ | $\mathcal{C}_{val}$ | $\mathcal{C}_{test}$ | $\mathcal{N}_{all}$ | $\mathcal{N}_{train}$ | $\mathcal{N}_{val}$ | $\mathcal{N}_{test}$ | Context | Similarity |
|---|---|---|---|---|---|---|---|---|---|---|
| *mini*ImageNet (Vinyals et al., 2016) | 100 | 64 | 16 | 20 | 60,000 | 38,400 | 9,600 | 12,000 | 0 | 0.211 |
| CIFAR-FS (Krizhevsky et al., 2009) | 100 | 64 | 16 | 20 | 60,000 | 38,400 | 9,600 | 12,000 | 0 | 0.181 |
| Stanford Dogs (Khosla et al., 2011) | 120 | 70 | 20 | 30 | 20,580 | 12,165 | 3,312 | 5,103 | 0 | 0.244 |
| Stanford Cars (Krause et al., 2013) | 196 | 130 | 17 | 49 | 16,185 | 10,766 | 1,394 | 4,025 | 0 | 0.164 |
| Aircraft (Wah et al., 2011) | 100 | 70 | 15 | 15 | 10,000 | 5,000 | 2,500 | 2,500 | 0 | 0.228 |
| CUB-200-2011 (Wah et al., 2011) | 200 | 140 | 30 | 30 | 11,788 | 7,648 | 1,182 | 2,958 | 0 | 0.266 |
| Describable Textures (Cimpoi et al., 2014) | 47 | 33 | 7 | 7 | 5,640 | 3,960 | 840 | 840 | 0 | 0.194 |
| Traffic Signs (Houben et al., 2013) | 43 | - | - | 43 | 50,000 | - | - | 50,000 | 0 | 0.193 |
| Omniglot (Lake et al., 2015) | 50 | 25 | 5 | 20 | 32,460 | 17,660 | 1,620 | 13,180 | 0 | 0.212 |
| Fungi (Schroeder & Cui, 2018) | 1394 | 994 | 200 | 200 | 89,760 | 64,449 | 12,195 | 13,116 | 0 | 0.191 |
| VGG Flower (Nilsback & Zisserman, 2008) | 102 | 71 | 15 | 16 | 8,189 | 5,655 | 1,109 | 1,425 | 0 | 0.177 |
| MSCOCO (Lin et al., 2014) | 80 | - | 40 | 40 | 860,001 | - | 513,021 | 346,980 | 0 | 0.173 |
| Quick Draw (Jongejan et al., 2016) | 345 | 241 | 52 | 52 | 50,426,266 | 34,776,331 | 7,939,640 | 7,710,295 | 0 | 0.168 |
| CropDiseases (Mohanty et al., 2016) | 38 | - | - | 38 | 43,456 | - | - | 43,456 | 0 | 0.213 |
| ChestX (Wang et al., 2017b) | 8 | - | - | 8 | 25,848 | - | - | 25,848 | 0 | 0.183 |
| EuroSAT (Helber et al., 2019) | 10 | - | - | 10 | 27,000 | - | - | 27,000 | 0 | 0.173 |
| ISIC2018 (Codella et al., 2019) | 7 | - | - | 7 | 10,015 | - | - | 10,015 | 0 | 0.186 |
| MetaCoCo (Ours) | 100 | 64 | 16 | 20 | 175,637 | 156,666 | 5,839 | 12,268 | 155 | 0.142 |

etc. Some toy benchmarks, *e.g.*, ColoredMNIST (Arjovsky et al., 2019) and real-world benchmarks, *e.g.*, NICO (He et al., 2021) and MetaShift (Liang & Zou, 2022), have been proposed to evaluate the performance of these methods. These TML benchmarks do not be used directly in the FSC setting, due to lack of sufficient classes, number of samples, and number of contexts. Although IFSL (Yue et al., 2020) and COSOC (Luo et al., 2021) have experimentally proved the importance of spurious-correlation shifts, there is still a lack of a benchmark for evaluation. Therefore, we propose MetaCoCo in this paper to reflect spurious-correlation shifts arising in real-world scenarios.

## 3 PROBLEM AND EVALUATION SETTINGS

FSC aims to recognize unlabeled images (or query sets) from novel classes with only few labeled images (or support sets). Following the previous studies (Vinyals et al., 2016; Tian et al., 2020b), we adopt an episodic paradigm to train and evaluate the few-shot models. Specifically, each $N$-way $K$-shot episode $\mathcal{T}_e$ has a support set $\mathcal{S}_e = \{(x_i, y_i) : i = 1, \ldots, I_s\}$ and a query set $\mathcal{Q}_e = \{(x_i, y_i) : i = I_s + 1, \ldots, I_s + I_q\}$, where $x_i \in \mathcal{X}$ is the image and $y_i \in \mathcal{Y}$ is the label from a set of $N$ classes $\mathcal{C}_e$, with $I_s = N \cdot K$ and $I_q$ be the image numbers in the support and query set, respectively.

Let $\mathcal{S}_e(\mathcal{X})$ and $\mathcal{Q}_e(\mathcal{X})$ be the image spaces of $\mathcal{S}_e$ and $\mathcal{Q}_e$, and $\mathcal{S}_e(\mathcal{Y})$ and $\mathcal{Q}_e(\mathcal{Y})$ be the corresponding label spaces, respectively. The label space of $\mathcal{S}_e$ and $\mathcal{Q}_e$ is same but the image space is different, *i.e.*, $\mathcal{S}_e(\mathcal{X}) \neq \mathcal{Q}_e(\mathcal{X})$ and $\mathcal{S}_e(\mathcal{Y}) = \mathcal{Q}_e(\mathcal{Y})$. During the training phase, for meta-based and metric-based methods, episodes are randomly sampled from the base classes set $\mathcal{D}_b$ to train the model. Instead, for fine-tuning based methods, a mini-batch images is randomly sampled from $\mathcal{D}_b$ to train the model. During the testing phase, the trained model is fine tuned with $\mathcal{S}_e$ and evaluated with $\mathcal{Q}_e$ in novel episodes sampled from the novel classes set $\mathcal{D}_n$. Note that $\mathcal{D}_b$ contains more images and classes compared with $\mathcal{D}_n$ but label spaces are disjoint, *i.e.*, $\mathcal{D}_b(\mathcal{Y}) \neq \mathcal{D}_n(\mathcal{Y})$[1]. The model architectures have a feature encoder $f_\theta$ and a classifier $c_\phi$ parameterized by $\theta$ and $\phi$. The $f_\theta$ aims to extract features, $f_\theta : \mathcal{X} \to \mathcal{Z}$, and the $c_\phi$ predicts the class of extracted features, $c_\phi : \mathcal{Z} \to \mathcal{Y}$.

### 3.1 CROSS-DOMAIN SHIFTS AND SPURIOUS-CORRELATION SHIFTS

In Table 1, we summarize the statistics of the existing benchmarks and our proposed spurious-correlation benchmark, *i.e.*, MetaCoCo. Specifically, Meta-dataset (Triantafillou et al., 2020) and BSCD-FSL (Guo et al., 2020) are two commonly used cross-domain benchmarks, where Meta-dataset has *10 existing datasets*, including ILSVRC-2012 (Deng et al., 2009), Omniglot (Lake et al., 2015), Aircraft (Wah et al., 2011), CUB-200-2011 (Wah et al., 2011), Describable Textures (Cimpoi et al., 2014), Quick Draw (Jongejan et al., 2016), Fungi (Schroeder & Cui, 2018), VGG Flower (Nilsback & Zisserman, 2008), Traffic Signs (Houben et al., 2013) and MSCOCO (Lin et al., 2014). BSCD-FSL (Guo et al., 2020) has *4 existing datasets*, including CropDiseases (Mohanty et al., 2016), EuroSAT (Helber et al., 2019), ISIC2018 (Codella et al., 2019; Tschandl et al., 2018), and

---

[1]$\mathcal{D}_b(\mathcal{Y})$ and $\mathcal{D}_n(\mathcal{Y})$ can be defined similarly, meaning the label spaces of $\mathcal{D}_b$ and $\mathcal{D}_n$, respectively.

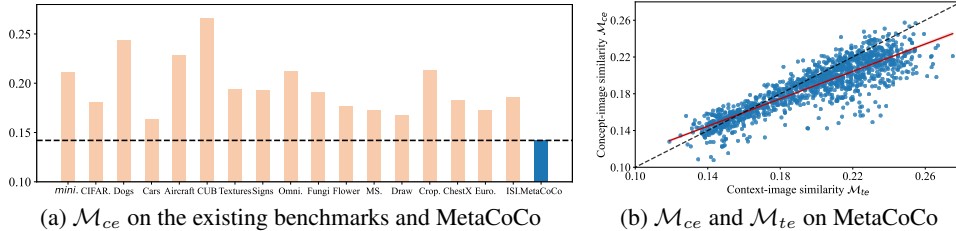

(a) $\mathcal{M}_{ce}$ on the existing benchmarks and MetaCoCo      (b) $\mathcal{M}_{ce}$ and $\mathcal{M}_{te}$ on MetaCoCo

Figure 2: (a) The sample-averaged similarity $\mathcal{M}_{ce}$ between concepts and images on the existing FSC benchmarks and the proposed MetaCoCo, where MetaCoCo has significantly lower similarity between contexts and images. (b) The context-image similarities $\mathcal{M}_{te}$ (horizontal axis) versus the concept-image similarities $\mathcal{M}_{ce}$ (vertical axis) of the sample points in the MetaCoCo.

ChestX (Wang et al., 2017b). The main differences between cross-domain benchmarks and our proposed MetaCoCo benchmark are as follows: (1) **The cause of shifts.** The shifts in cross-domain benchmarks are caused by varying distributions between various datasets. Instead, the shifts in MetaCoCo are caused by varying both concepts and contexts. For example, for cross-domain shifts, the FSL model is trained on *mini*ImageNet and tested on EuroSAT. Whereas for spurious-correlation shifts, the FSL model is trained and tested on images that have distinct associations with the contexts. (2) **The use of contexts.** In contrast to the existing few-shot classification benchmarks, as shown in Table 1, the proposed MetaCoCo benchmark further uses context information collected from real-world scenarios to reflect the spurious-correlation shifts.

## 3.2 SIMILARITY BETWEEN THE CONCEPT AND CONTEXT INFORMATION

For images containing both conceptual and contextual information, a greater similarity between image and context implies that the benchmark has more spurious-correlation shifts. To intuitively show that MetaCoCo has considerably more spurious-correlation shifts than the existing FSC benchmarks including cross-domain-shift benchmarks, we introduce a novel metric that uses CLIP (Radford et al., 2021) as a pre-trained vision-language model. By calculating the cosine distance of text and image features extracted by pre-trained text and image encoder from CLIP, the similarity $\mathcal{M}_{ce}$ between conceptual language information and image visual knowledge, and the similarity $\mathcal{M}_{te}$ between contextual language expression and image visual knowledge are calculated as follows:

$$\mathcal{M}_{ce} = d(z_x, z_t^{ce}), \quad \mathcal{M}_{te} = d(z_x, z_t^{te}), \tag{1}$$

where $d(\cdot, \cdot)$ is the cosine distance measurement, $z_x$ is the image features extracted by pre-trained image encoder by CLIP, and $z_t^{ce}$ and $z_t^{te}$ represent the text features of concept and context extracted by pre-trained text encoder by CLIP, respectively. Figure 2(a) shows the sample-averaged similarity $\mathcal{M}_{ce}$ between concepts[2] and images on the existing FSC benchmarks as well as the proposed MetaCoCo. It can be seen that MetaCoCo has significantly lower similarity between concepts and images. This is because the added context information in the image introduces spurious-correlations with the concepts, *e.g.*, "grass" and "dog", thus weakening the direct correlation between the images and the concepts or labels, and presenting a more challenging evaluating benchmark for the FSC. Figure 2(b) further shows the context-image similarities $\mathcal{M}_{te}$ (horizontal axis) versus the concept-image similarities $\mathcal{M}_{ce}$ (vertical axis) of the sample points in the MetaCoCo. We find that the overall context-image similarities are slightly higher than the concept-image similarities, suggesting that spurious-correlation shifts are substantial in the proposed benchmark.

## 3.3 EVALUATION STRATEGIES

Before presenting the datasets, we first discuss the evaluation strategies in MetaCoCo, including:

(1) **Fine-tuning based methods.** Fine-tuning based methods follow the transfer learning procedure, including two phases: pre-training with base classes and test-tuning with novel classes. In the pre-

---

[2]Since the existing FSC benchmarks lack context information as shown in Table 1, we are not able to compute their sample-averaged similarity $\mathcal{M}_{te}$ between contexts and images.

training with base classes phase, the base classes $\mathcal{D}_b$ is used to train a $\mathcal{C}_{base}$-class classifier as below:

$$\Gamma = \arg\min_{\theta,\phi} \sum_{i=1}^{T} \mathcal{L}_{\text{CE}}\Big( c_\phi(f_\theta(x_i)), y_i \Big), \tag{2}$$

where $T$ is the sample number of $\mathcal{D}_b$, and $\mathcal{L}_{\text{CE}}(\cdot, \cdot)$ is the cross-entropy loss. In the test-tuning with novel classes phase, each episode $\mathcal{T}_e = \langle \mathcal{S}_e, \mathcal{Q}_e \rangle$ is sampled from novel classes $\mathcal{D}_n$ and a new $\mathcal{C}_e$-class classifier is re-learned based on a few labeled images $\mathcal{S}_e$ and tested on $\mathcal{Q}_e$.

**(2) Metric-based methods.** Metric-based methods directly compare the similarities (or distance) between query images and support classes, *i.e.*, learning to compare, through the episodic training mechanism. Taking Prototypical Network (ProtoNet) (Snell et al., 2017) as an example, it takes the mean vector of each support class as its corresponding prototype representation, and then compares the relationships between query images and prototypes. The prototype $p_n$ of each class in the support set $\mathcal{S}_e$ can be formulated as $p_n = \frac{1}{K} \sum_{(x_i,y_i)\in\mathcal{S}_e} f_\theta(x_i) \cdot \mathbb{I}(y_i = n)$, where $\mathbb{I}(\cdot)$ is the indicator function, then the the metric loss on $\mathcal{Q}_e$ can be computed as:

$$\mathcal{L}(\theta) = -\frac{1}{I_q} \sum_{(x_i,y_i)\in\mathcal{Q}_e} \log P(y_i|\mathcal{Q}_e), \quad \text{where } P(y_i|\mathcal{Q}_e) = \frac{\exp(-D(f_\theta(x_i), p_{y_i}))}{\sum_{n=1}^{N} \exp(-D(f_\theta(x_i), p_n))}, \tag{3}$$

and $D(\cdot, \cdot)$ denotes a distance measurement, *e.g.*, the squared euclidean distance in the ProtoNet.

**(3) Meta-based methods.** Meta-based methods aim to make the trained model able to quickly adapt to unseen novel tasks by a few gradient steps in the testing phase. Specifically, the learning paradigm of meta-based methods has two levels, *i.e.*, inner-level and outer-level, to update the base and meta learner, respectively. Model-agnostic meta-learning (MAML) (Finn et al., 2017) is one representative method, whose core idea is to train a model's initial parameters by using the two levels. Specifically, the base learner is optimized on the support set $\mathcal{S}_e$ that

$$\begin{aligned} \{\theta, \phi\} &\leftarrow \{\theta, \phi\} - \eta_{out}\nabla_{\{\theta,\phi\}}\mathcal{L}_{ce}(c_{\phi'}(f_{\theta'}(x_i), y_i)), \\ \text{where} \quad \{\theta', \phi'\} &= \{\theta, \phi\} - \eta_{in}\nabla_{\{\theta,\phi\}}\mathcal{L}_{ce}(c_\phi(f_\theta(x_i), y_i)), \end{aligned} \tag{4}$$

and the $\eta_{in}$ and $\eta_{out}$ are the learning rates of the inner level and the outer level, respectively.

## 4 METACOCO: A NEW FEW-SHOT CLASSIFICATION BENCHMARK WITH SPURIOUS CORRELATION

MetaCoCo aims to present an environment for evaluating the fine control of spurious-correlation shifts in the FSC problems. Specifically, our approach consists of (1) dataset generating, and (2) episode sampling, whose operational procedures are detailed below.

**Dataset generating.** Compared with the existing benchmarks, the samples in MetaCoCo consist of both conceptual and contextual information, and many of these images exhibit a strong correlation with the context, which increases the impact of spurious-correlation shifts between the training data and the testing data on the prediction performance. Specifically, we first select 100 categories of common objects following DomainNet (Peng et al., 2019). These categories include 155 contexts, which are collected from the adjectives or nouns appeared more frequently with these categories from WordNet (Miller, 1995). Then the images are collected by searching a category name combined with a context name (*e.g.*, "dog on grass") in various image search engines. One of the main challenges is that the downloaded data contains a large portion of outliers. To clean the dataset, we manually filter out the outliers, which takes around 2,500 hours in total. To control the annotation quality, we assign two annotators to each image and only take the images agreed by both annotators. After the filtering process, we kept 17.6k images from the 1.0 million images crawled from the web. The dataset has an average of around 1,000 images per category (see Appendix B for more details).

**Episode sampling.** MetaCoCo has 100 categories, and the number of matching contexts for each category is inconsistent, resulting in an inconsistent number of samples for each category. We sort the samples from most to least. The first 64 categories with the largest number of samples are used as training data, then 20 categories are selected as testing data, and the last 16 categories are used as validation data. FSC adopts an episodic paradigm to train and test the model. Each $N$-way $K$-shot

Table 2: Experiments in state-of-the-art few-shot classification and self-supervised learning methods. "rot." and "jig." mean using the Rotation and Jigsaw self-supervised pretext tasks, respectively.

| Method | Conference | Backbone | Type | GL | LL | TT | 1-shot | 5-shot |
|---|---|---|---|---|---|---|---|---|
| Baseline (Chen et al., 2019) | ICLR 2019 | ResNet12 | Fine-tuning | ✓ | | ✓ | 46.78 | 60.78 |
| Baseline++ (Chen et al., 2019) | ICLR 2019 | ResNet12 | Fine-tuning | ✓ | | ✓ | 46.95 | 58.50 |
| RFS-simple (Tian et al., 2020a) | ECCV 2020 | ResNet12 | Fine-tuning | ✓ | | ✓ | 47.02 | 56.71 |
| Neg-Cosine (Liu et al., 2020) | ECCV 2020 | ResNet12 | Fine-tuning | ✓ | | ✓ | 50.78 | 62.34 |
| SKD-GEN0 (Rajasegaran et al., 2020) | BMVC 2021 | ResNet12 | Fine-tuning | ✓ | | ✓ | 51.34 | 63.21 |
| FRN (Wertheimer et al., 2021) | CVPR 2021 | ResNet12 | Fine-tuning | ✓ | | ✓ | 50.23 | 60.56 |
| Yang et al (Yang et al., 2022) | ECCV 2022 | ResNet12 | Fine-tuning | ✓ | | ✓ | 58.01 | 69.32 |
| LP-FT-FB (Wang et al., 2023) | ICLR 2023 | ResNet12 | Fine-tuning | ✓ | ✓ | ✓ | 56.21 | 70.21 |
| MAML (Finn et al., 2017) | ICML 2017 | ResNet12 | Meta | | ✓ | ✓ | 45.01 | 54.21 |
| Versa (Gordon et al., 2018) | NeurIPS 2018 | ResNet12 | Meta | | ✓ | ✓ | 39.64 | 53.06 |
| R2D2 (Bertinetto et al., 2019) | ICLR 2019 | ResNet12 | Meta | | ✓ | ✓ | 45.25 | 60.14 |
| MTL (Sun et al., 2019) | CVPR 2019 | ResNet12 | Meta | ✓ | ✓ | ✓ | 44.23 | 58.04 |
| ANIL (Raghu et al., 2020) | ICLR 2020 | ResNet12 | Meta | | ✓ | ✓ | 36.58 | 50.54 |
| BOIL (Oh et al., 2020) | ICLR 2021 | ResNet12 | Meta | | ✓ | ✓ | 44.09 | 55.61 |
| CDKT+ML (Ke et al., 2023) | NeurIPS 2023 | ResNet18 | Meta | | ✓ | ✓ | 44.86 | 61.42 |
| CDKT+PL (Ke et al., 2023) | NeurIPS 2023 | ResNet18 | Meta | | ✓ | ✓ | 43.21 | 59.87 |
| CovaMNet (Li et al., 2019b) | AAAI 2019 | ResNet12 | Metric | ✓ | | | 47.81 | 58.43 |
| DN4 (Li et al., 2019a) | CVPR 2019 | ResNet12 | Metric | ✓ | | | 45.04 | 57.68 |
| CAN (Hou et al., 2019) | NeurIPS 2019 | ResNet12 | Metric | ✓ | ✓ | | 48.93 | 62.36 |
| DeepBDC (Xie et al., 2022) | CVPR 2022 | ResNet12 | Metric | | ✓ | ✓ | 46.78 | 62.54 |
| FGFL (Cheng et al., 2023) | ICCV 2023 | ResNet12 | Metric | | ✓ | ✓ | 46.78 | 64.32 |
| PUTM (Tian et al., 2023) | ICCV 2023 | ResNet18 | Metric | | ✓ | ✓ | 60.23 | 72.36 |
| TSA+DETA (Zhang et al., 2023a) | ICCV 2023 | ResNet18 | Metric | | ✓ | ✓ | 51.42 | 61.58 |
| MoCo (He et al., 2020) | CVPR 2020 | ResNet50 | Self-supervised learning | | ✓ | ✓ | 56.90 | 70.65 |
| SimCLR (Chen et al., 2020) | ICML 2020 | ResNet50 | Self-supervised learning | | ✓ | ✓ | 58.12 | 71.21 |
| ProtoNet (Snell et al., 2017) | NeurIPS 2017 | ResNet18 | Metric | | ✓ | | 43.14 | 57.84 |
| + rot. + SSFSL (Su et al., 2020) | ECCV 2020 | ResNet18 | Self-supervised learning | | ✓ | | 40.65 | 54.31 |
| + rot. + HTS (Zhang et al., 2022a) | ECCV 2022 | ResNet18 | Self-supervised learning | | ✓ | | 42.06 | 55.13 |
| + jig. + SSFSL (Su et al., 2020) | ECCV 2020 | ResNet18 | Self-supervised learning | ✓ | | | 45.43 | 58.91 |
| + rot. + jig. + SSFSL (Su et al., 2020) | ECCV 2020 | ResNet18 | Self-supervised learning | | ✓ | | 44.46 | 59.01 |
| ProtoNet (Snell et al., 2017) | NeurIPS 2017 | ResNet12 | Metric | | ✓ | | 42.69 | 59.50 |
| + rot. + SLA (Lee et al., 2020) | ICML 2020 | ResNet12 | Self-supervised learning | | ✓ | | 40.29 | 58.09 |
| + rot. + HTS (Zhang et al., 2022a) | ECCV 2022 | ResNet12 | Self-supervised learning | | ✓ | ✓ | 43.19 | 60.50 |
| ProtoNet (Snell et al., 2017) | NeurIPS 2017 | WRN-28-10 | Metric | | ✓ | | 43.67 | 60.78 |
| + rot. + BF3S (Gidaris et al., 2019) | ICCV 2019 | WRN-28-10 | Self-supervised learning | | ✓ | | 43.78 | 57.64 |
| + rot. + HTS (Zhang et al., 2022a) | ECCV 2022 | WRN-28-10 | Self-supervised learning | | ✓ | | 45.31 | 62.31 |

episode $\mathcal{T}_e$ has a support set $\mathcal{S}_e$ and a query set $\mathcal{Q}_e$, where $\mathcal{S}_e$ and $\mathcal{Q}_e$ share the same categories but different images. Therefore, we have two sample episodic strategies: independent and identically distributed (IID) episode, *i.e.*, the support and query images with the *same* contexts, and out-of-distribution (OOD) episode, *i.e.*, the support and query images with the *different* contexts.

## 5 EXPERIMENTS

In this section, we evaluate the spurious-correlation performance of the state-of-the-art methods optimized with different learning strategies. These experiments further demonstrate that SC-FSC is still a major challenge. (see Appendix C and D for more experimental details and results).

### 5.1 EXPERIMENTAL SETUP

**Few-shot classification methods.** We evaluate the performance with a large number of algorithms that span different learning strategies, including: (1) Five *fine-tuning based methods*: Baseline (Chen et al., 2019), Baseline++ (Chen et al., 2019), RFS-simple (Tian et al., 2020a), Neg-Cosine (Liu et al., 2020) and SKD-GEN0 (Rajasegaran et al., 2020). (2) Six *metric-based methods*: ProtoNet (Snell et al., 2017), RelationNet (Sung et al., 2018), CovaMNet (Li et al., 2019b), DN4 (Li et al., 2019a), CAN (Hou et al., 2019) and RENet (Kang et al., 2021). (3) Six *meta-based methods*: MAML (Finn et al., 2017), Versa (Gordon et al., 2018), R2D2 (Bertinetto et al., 2019), MTL (Sun et al., 2019), ANIL (Raghu et al., 2020) and BOIL (Oh et al., 2020). (4) Six *self-supervised learning methods*: MoCo (He et al., 2020), SimCLR (Chen et al., 2020), SSFSL (Su et al., 2020), HTS (Zhang et al., 2022a), SLA (Lee et al., 2020) and BF3S (Gidaris et al., 2019). (5) Seven *cross-domain methods*: Linear (Yue et al., 2020), Cosine (Yue et al., 2020), $k$-NN (Yue et al., 2020), ATA (Wang & Deng, 2021), FT (Tseng et al., 2020), LRP (Sun et al., 2021) and IFSL (Yue et al., 2020).

**Backbone architectures.** Following prior literatures (Li et al., 2023d), all fine-tuning based methods, metric-based methods and meta-based methods adopt three different embedding backbones from shallow to deep, *i.e.*, Conv64F, ResNet12 and ResNet18. For other learning strategy methods,

Table 3: Experiments of cross-domain and spurious-correlation few-shot classification methods.

| Method | Conference | Type | GL | LL | TT | 5-way 1-shot | 5-way 5-shot |
|---|---|---|---|---|---|---|---|
| RelationNet (Sung et al., 2018) | CVPR 2018 | Metric | | ✓ | | 45.32 ± 0.48 | 57.73 ± 0.45 |
| +ATA (Wang & Deng, 2021) | IJCAI 2021 | CD-FSC | | ✓ | | 43.24 ± 0.47 | 56.94 ± 0.47 |
| +FT (Tseng et al., 2020) | ICLR 2020 | CD-FSC | | ✓ | | 45.37 ± 0.50 | 58.74 ± 0.48 |
| GNN (Satorras & Estrach, 2018) | ICLR 2018 | Metric | | ✓ | | 48.14 ± 0.55 | 61.94 ± 0.56 |
| +ATA (Wang & Deng, 2021) | IJCAI 2021 | CD-FSC | | ✓ | | 46.78 ± 0.55 | 61.78 ± 0.52 |
| + FT (Tseng et al., 2020) | ICLR 2020 | CD-FSC | | ✓ | | 47.30 ± 0.56 | 65.90 ± 0.56 |
| TPN (Liu et al., 2018) | ICLR 2019 | Metric | | ✓ | | 49.65 ± 0.51 | 60.62 ± 0.47 |
| +ATA (Wang & Deng, 2021) | IJCAI 2021 | CD-FSC | | ✓ | | 47.15 ± 0.53 | 60.33 ± 0.31 |
| +FT (Tseng et al., 2020) | ICLR 2020 | CD-FSC | | ✓ | | 45.62 ± 0.51 | 55.78 ± 0.52 |
| Linear (Yue et al., 2020) | NeurIPS 2020 | Fine-tuning | | | | 43.31 ± 0.40 | 57.87 ± 0.41 |
| Cosine (Yue et al., 2020) | NeurIPS 2020 | Fine-tuning | ✓ | | ✓ | 42.81 ± 0.42 | 56.33 ± 0.41 |
| $k$-NN (Yue et al., 2020) | NeurIPS 2020 | Fine-tuning | ✓ | | ✓ | 42.22 ± 0.42 | 57.93 ± 0.42 |
| MAML (Finn et al., 2017) | ICML 2017 | Meta | | ✓ | ✓ | 44.09 ± 0.52 | 53.98 ± 0.48 |
| +IFSL (Yue et al., 2020) | NeurIPS 2020 | SC-FSC | | ✓ | ✓ | 43.42 ± 0.51 | 55.00 ± 0.48 |
| MTL (Sun et al., 2019) | CVPR 2019 | Meta | ✓ | ✓ | ✓ | 43.80 ± 0.48 | 57.18 ± 0.48 |
| +IFSL (Yue et al., 2020) | NeurIPS 2020 | SC-FSC | ✓ | ✓ | ✓ | 43.42 ± 0.48 | 56.90 ± 0.48 |
| MatchingNet (Vinyals et al., 2016) | NeurIPS 2016 | Metric | | ✓ | | 43.72 ± 0.49 | 56.12 ± 0.49 |
| +IFSL (Yue et al., 2020) | NeurIPS 2020 | SC-FSC | | ✓ | | 44.11 ± 0.49 | 55.86 ± 0.49 |
| SIB (Hu et al., 2020) | ICLR 2020 | Meta | | ✓ | ✓ | 48.43 ± 0.57 | 58.53 ± 0.51 |
| +IFSL (Yue et al., 2020) | NeurIPS 2020 | SC-FSC | | ✓ | ✓ | 47.97 ± 0.54 | 58.41 ± 0.50 |

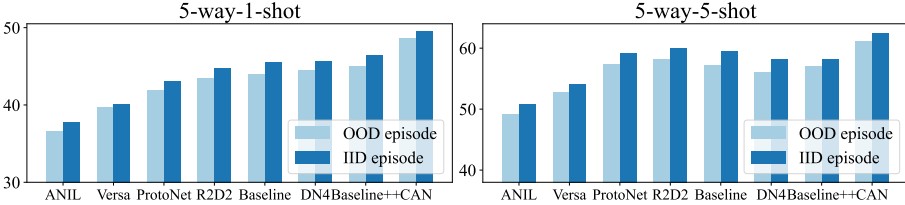

Figure 3: Experiments of the test-tuning phase with different sampling episodes, *i.e.*, IID and OOD.

we adopt different feature backbones based on the corresponding original papers, *e.g.*, ResNet10 for cross-domain few-shot classification methods, WRN-28-10 for self-supervised learning methods.

**Evaluation protocols.** Following the prior work (Li et al., 2023d), in this paper, we control the evaluation for all methods , evaluate them on 600 sampled tasks and repeat this process five times, *i.e.*, a total of 3,000 tasks. The top-1 mean accuracy will be reported. All images are resized into 84 × 84 by using the single center crop (Li et al., 2019b). Three common tricks are used: (1) *Global-label (GL)* indicates that the global labels of the training set are used for pre-training during the training phase. (2) *Local-label (LL)* means that only the specific local labels are used in the episodic training phase. (3) *Test-tune (TT)* means test-tuning of using the support set at the testing stage.

## 5.2 MAIN RESULTS

In this section, we conduct extensive experiments on various methods with six learning strategies.

**Experiments in fine-tuning, metric- and meta-based methods and self-supervised methods.** We evaluate the performance of 17 competing few-shot methods and six self-supervised methods in our MetaCoCo. The results of the 5-way 1- or 5-shot setting are shown in Table 2. From Table 2, we have the following findings: (1) We find that the performance of all methods decreases compared with existing FSC benchmarks (Li et al., 2023d), which demonstrates that these methods are insufficient in solving the spurious-correlation-shift problem. (2) Previous works introduced self-supervised learning to improve the generalization of FSC models, but experiments have shown that this is not suitable for the SC-FSC problem. In some cases, using self-supervised learning even damages the performance, *i.e.*, ProtoNet has 43.14% in 1-shot, but the accuracy by using rotation is 40.65%.

**Experiments in CD-FSC and SC-FSC methods.** Table 3 displays the accuracy of seven CD-FSC methods. These methods have a significant performance on solving the cross-domain-shift problem on the Meta-dataset (Triantafillou et al., 2020) and BSCD-FSL (Guo et al., 2020). However, in MetaCoCo, the advantages of these methods disappear, resulting in weaker performance, even worse than non-cross-domain FSC methods. It is worth noting that the main motivation of IFSL (Yue et al., 2020) is to use the idea of causality to solve the impact of spurious correlation between contextual information and images on the model training phase. However, we observe a substantial decrease of the performance on the real-world spurious-correlation benchmark, *i.e.*, MetaCoCo.

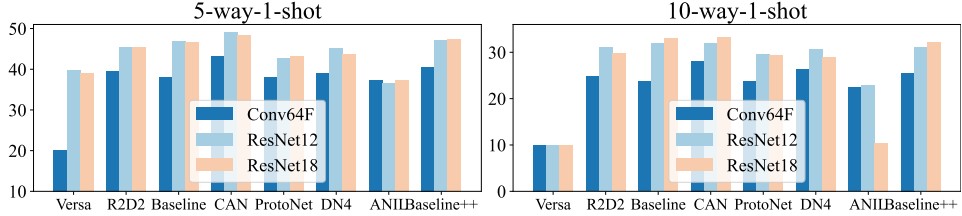

Figure 4: Experiments of different backbone architectures under 5-way and 10-way 1-shot settings.

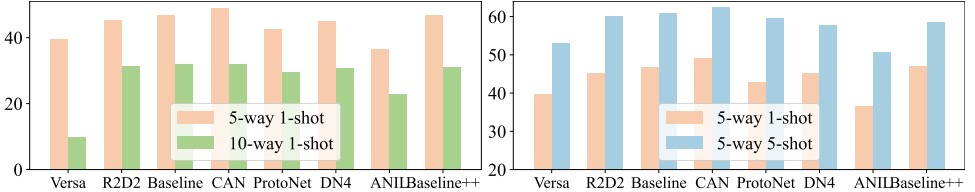

Figure 5: Experimental results of different ways (left) and shots (right) on testing performance.

To this end, according to these experimental results, we observe that most methods are insufficient to solve the spurious-correlation-shift FSC problem. We hope the proposed MetaCoCo can facilitate future research on the important and real-world problem for few-shot classification.

### 5.3 IN-DEPTH STUDY

To further analyze the influence of spurious shifts in MetaCoCo, we conduct in-depth experiments.

**Effect of the IID and OOD episodes.** Figure 3 shows the results of FSC methods under 5-way 1- and 5-shot settings. The IID and OOD episodes represent the same and different contexts of the support and query sets during the test-tuning phase, respectively (see Section 4). These results clearly denote that the learning process of the IID episode is better than the optimization process of the OOD episode. This further demonstrates that the model tends to utilize contextual information during the learning process. Once images do not match the contexts, the performance will deteriorate.

**Effect of different backbone architectures.** In Chen et al. (2019), they change the depth of the feature backbone to reduce intra-class variation for all methods. Following this paper, we start from Conv64F and gradually increase the backbone to ResNet12 and 18. The experiments under 5-way and 10-way 1-shot settings are shown in Figure 4. It is arguably a common sense that the stronger backbone is used, the performance is best. However, we surprisingly find that this may not be always in the SC-FSC problem. Figure 4 shows the performance degradation in some settings.

**Ways and shots analysis.** We further study the performance of "ways" (Figure 5 left) and "shots" (Figure 5 right). As expected, we found that the difficulty increases as the way increases, and performance degrades. More examples per class, on the other hand, indeed make it easier to correctly classify that class. Interestingly, Versa presents a poor performance with increasing the way but it improves at a high rate when the shot increases, which further represents that the contextual effects become larger when the task becomes difficult. CAN has the best accuracy under all settings because it uses a transduction strategy to introduce query samples in the training phase, which destroys the strong spurious correlations between contexts and images.

## 6 CONCLUSION

In this paper, we present Meta Concept Context (MetaCoCo), a large-scale, diverse and realistic environment benchmark for spurious-correlation few-shot classification. We believe that our exploration of various modes on MetaCoCo has uncovered interesting directions for future works: it remains unclear what is the best learning strategy for avoiding the effect of spurious-correlation contexts and the most appropriate episodic sample. Current models even including these cross-domain FSC models don't work when trained on mismatching contexts. Current models are also not robust to the amount of data in testing episodes, each excelling in a different part of the spectrum. We believe that addressing these shortcomings constitutes an important research goal moving forward.

ACKNOWLEDGMENTS

This work was supported in part by National Natural Science Foundation of China (No. U20A20387, 62376243, 62037001, 623B2002), the StarryNight Science Fund of Zhejiang University Shanghai Institute for Advanced Study (SN-ZJU-SIAS-0010) and Project by Shanghai AI Laboratory (P22KS00111). All opinions in this paper are those of the authors and donot necessarily reflect the views of the funding agencies.

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

## A    MORE DISCUSSION ON THE EXISTING BENCHMARKS

In Table 1, we have summarized statistics of existing benchmarks. A brief introduction of benchmarks mentioned in this paper is the following. For more details, please refer to the original paper.

*mini*ImageNet (Vinyals et al., 2016). *mini*ImageNet is the subsets of the ILSVRC-12 dataset (Russakovsky et al., 2015).

**Fine-grained benchmarks.** CUB-200-2011 (Wah et al., 2011), Stanford Dogs (Khosla et al., 2011) and Stanford Cars (Krause et al., 2013) are initially designed for fine-grained classification.

**Meta-dataset (Triantafillou et al., 2020).** Meta-dataset is a cross-domain FSC benchmark and has *10 existing datasets*, including ILSVRC-2012 (Deng et al., 2009), Omniglot (Lake et al., 2015), Aircraft (Wah et al., 2011), CUB-200-2011 (Wah et al., 2011), Describable Textures (Cimpoi et al., 2014), Quick Draw (Jongejan et al., 2016), Fungi (Schroeder & Cui, 2018), VGG Flower (Nilsback & Zisserman, 2008), Traffic Signs (Houben et al., 2013) and MSCOCO (Lin et al., 2014).

**BSCD-FSL (Guo et al., 2020)** BSCD-FSL is also a cross-domain FSC benchmark and has *4 existing datasets*, including CropDiseases (Mohanty et al., 2016), EuroSAT (Helber et al., 2019), ISIC2018 (Codella et al., 2019; Tschandl et al., 2018), and ChestX (Wang et al., 2017b).

## B    DETAILED DATASET STATISTICS

Tables 4 and 5 show the number of samples for concepts and detailed statistics of the MetaCoCo benchmark, respectively. In particular, our benchmark contains 100 concepts (or categories), 155 contexts and 17.6k images. These concepts are from common objects following DomainNet (Peng et al., 2019). The 155 contexts are collected from the adjectives or nouns appeared more frequently with these concepts from WordNet (Miller, 1995). In addition, we show the statistics of samples in each concept in Table 4.

## C    EXPERIMENTAL DETAILS

In this paper, many feature backbones are used to fair evaluate the performance of few-shot classification methods. Specifically, Conv64F contains four convolutional blocks, each of which consists of a convolutional (Conv) layer, a batch-normalization (BN) layer, a ReLU/LeakyReLU layer and a max-pooling (MP) layer, where the numbers of filters of these blocks are {64, 64, 64, 64}. ResNet12 consists of four residual blocks, each of which further contains three convolutional blocks (each is built as Conv-BN-ReLU-MP) along with a skip connection layer, where the numbers of filters of these blocks are {64, 160, 320, 640}. ResNet18 is the standard architecture used in previous works. One important difference between ResNet12 and ResNet18 is that ResNet12 uses Dropblock in each residual block, while ResNet18 does not. In addition, the number of filters of these blocks in ResNet18 is {64, 128, 256, 512}. ResNet10 is a common backbone architecture in cross-domain few-shot classification methods. It has the same number of filters of blocks as ResNet18, but the number of layers is 1 in each stage. WRN-28-10 is frequently used in self-supervised learning methods, where 28 means the number of layers and 10 is the number of the width.

## D    MORE EXPERIMENTS

In Tables 6 and 7 and Figures 6 and 7, we show the additional experiments to supplement the results in our main paper. From these additional experiments, we find that most of the existing few-shot classification methods are not robust in the spurious-correlation problem. We hope that these studies and the proposed MetaCOCo can facilitate future research on real-world problems.

Table 4: Number of samples for concepts in the MetaCoCo benchmark.

| Training Concepts | dog | cat | bird | table | tree | bear | horse | fence | car | bicycle | motorcycle | train | cow | elephant | bus | chair | truck | airplane | pants | sheep | helicopter | door | monkey |
|---|---|---|---|---|---|---|---|---|---|---|---|---|---|---|---|---|---|---|---|---|---|---|---|
| Training Samples | 5820 | 4915 | 4397 | 4365 | 4035 | 4004 | 3980 | 3953 | 3873 | 3787 | 3755 | 3726 | 3612 | 3519 | 3491 | 3454 | 3393 | 3208 | 3080 | 3031 | 2994 | 2581 | 2578 |
| Training Concepts | umbrella | lion | squirrel | boat | wolf | lizard | tiger | giraffe | tent | hot air balloon | owl | sailboat | seal | frog | jacket | rabbit | goose | kangaroo | flower | ship | cactus | hat | fox |
| Training Samples | 2499 | 2470 | 2312 | 2310 | 2293 | 2215 | 2184 | 2115 | 2038 | 2017 | 2002 | 2001 | 1976 | 1943 | 1892 | 1877 | 1862 | 1859 | 1835 | 1815 | 1801 | 1770 | 1751 |
| Training Concepts | clock | spider | ostrich | tortoise | butterfly | pumpkin | sunflower | crocodile | bench | mailbox | lifeboat | dolphin | crab | window | pineapple | shorts | bag | toilet | | | | | |
| Training Samples | 1729 | 1723 | 1703 | 1672 | 1619 | 1588 | 1548 | 1499 | 1491 | 1466 | 1451 | 1182 | 1155 | 1141 | 1072 | 1026 | 993 | 864 | | | | | |
| Validation Concepts | carpet | cup | refrigerator | house | zebra | tower | ocean | spoon | suit | fire hydrant | skateboard | pillow | bed | knife | backpack | bridge | | | | | | | |
| Validation Samples | 437 | 434 | 431 | 430 | 420 | 381 | 379 | 362 | 357 | 347 | 340 | 329 | 317 | 314 | 312 | 249 | | | | | | | |
| Testing Concepts | rat | laptop | sink | frame | bowl | coat | bush | cloud | cabinet | shrimp | dress | television | t-shirt | sweater | surfboard | tie | fork | couch | keyboard | curtain | | | |
| Testing Samples | 857 | 841 | 774 | 763 | 719 | 703 | 661 | 630 | 620 | 552 | 546 | 541 | 540 | 531 | 530 | 506 | 497 | 490 | 483 | 482 | | | |

Table 5: Detailed statistics of the MetaCoCo benchmark.

| Contexts | dog | cat | bird | table | tree | bear | horse | fence | car | bicycle | motorcycle | train | cow | elephant | bus | chair | truck | airplane | pants | sheep | helicopter | door | monkey |
|---|---|---|---|---|---|---|---|---|---|---|---|---|---|---|---|---|---|---|---|---|---|---|---|
| grass | 649 | 597 | 839 | 0 | 0 | 665 | 604 | 0 | 321 | 583 | 452 | 346 | 1,009 | 559 | 332 | 213 | 440 | 331 | 0 | 573 | 453 | 0 | 428 |
| water | 886 | 635 | 606 | 0 | 0 | 550 | 305 | 0 | 306 | 327 | 390 | 271 | 387 | 402 | 348 | 503 | 442 | 256 | 0 | 292 | 249 | 0 | 251 |
| outdoor | 456 | 249 | 203 | 0 | 0 | 131 | 375 | 0 | 402 | 415 | 347 | 202 | 263 | 286 | 367 | 276 | 246 | 345 | 0 | 395 | 436 | 0 | 231 |
| dim | 322 | 241 | 414 | 0 | 0 | 164 | 318 | 0 | 390 | 521 | 489 | 333 | 268 | 338 | 202 | 145 | 486 | 299 | 0 | 303 | 477 | 0 | 184 |
| white | 218 | 153 | 63 | 426 | 22 | 156 | 110 | 256 | 209 | 0 | 37 | 136 | 53 | 0 | 252 | 163 | 238 | 263 | 505 | 60 | 0 | 537 | 0 |
| rock | 240 | 187 | 301 | 0 | 0 | 193 | 174 | 0 | 285 | 327 | 266 | 219 | 150 | 128 | 187 | 206 | 232 | 268 | 0 | 335 | 225 | 0 | 337 |
| autumn | 311 | 224 | 180 | 0 | 0 | 145 | 239 | 0 | 206 | 208 | 258 | 243 | 233 | 98 | 162 | 234 | 174 | 124 | 0 | 273 | 101 | 0 | 136 |
| black | 172 | 195 | 49 | 185 | 0 | 335 | 42 | 425 | 195 | 48 | 119 | 147 | 32 | 0 | 40 | 287 | 69 | 0 | 1,052 | 0 | 0 | 76 | 0 |
| brown | 357 | 145 | 66 | 967 | 246 | 383 | 407 | 214 | 0 | 0 | 0 | 25 | 106 | 55 | 0 | 294 | 0 | 0 | 211 | 0 | 0 | 235 | 0 |
| wood | 0 | 0 | 0 | 1,583 | 0 | 0 | 0 | 769 | 0 | 0 | 0 | 0 | 0 | 0 | 0 | 446 | 0 | 0 | 0 | 0 | 0 | 385 | 0 |
| blue | 0 | 0 | 30 | 78 | 0 | 0 | 0 | 73 | 137 | 35 | 47 | 263 | 0 | 0 | 217 | 118 | 84 | 157 | 314 | 0 | 0 | 59 | 0 |
| green | 0 | 0 | 0 | 53 | 1,453 | 0 | 0 | 227 | 34 | 0 | 0 | 86 | 0 | 0 | 62 | 73 | 36 | 0 | 60 | 0 | 0 | 47 | 0 |
| gray | 24 | 192 | 50 | 105 | 31 | 0 | 21 | 275 | 106 | 0 | 0 | 112 | 0 | 196 | 0 | 66 | 26 | 61 | 446 | 0 | 0 | 35 | 0 |
| on snow | 250 | 141 | 0 | 0 | 0 | 112 | 140 | 0 | 188 | 169 | 136 | 128 | 136 | 71 | 128 | 0 | 118 | 0 | 0 | 87 | 181 | 0 | 102 |
| large | 87 | 57 | 30 | 58 | 698 | 91 | 55 | 63 | 0 | 0 | 0 | 109 | 0 | 169 | 0 | 27 | 0 | 159 | 0 | 0 | 0 | 0 | 62 | 0 |
| eating | 277 | 299 | 201 | 0 | 0 | 0 | 0 | 0 | 0 | 0 | 0 | 0 | 147 | 123 | 0 | 0 | 0 | 0 | 0 | 116 | 0 | 0 | 169 |
| red | 0 | 0 | 0 | 69 | 0 | 0 | 0 | 59 | 117 | 0 | 47 | 126 | 0 | 0 | 179 | 98 | 93 | 23 | 68 | 0 | 0 | 76 | 0 |
| metal | 0 | 0 | 0 | 67 | 0 | 0 | 0 | 993 | 0 | 0 | 0 | 0 | 0 | 0 | 0 | 77 | 0 | 0 | 0 | 0 | 0 | 87 | 0 |
| lying | 234 | 146 | 0 | 0 | 0 | 229 | 142 | 0 | 0 | 0 | 0 | 0 | 173 | 73 | 0 | 0 | 0 | 0 | 0 | 116 | 0 | 0 | 0 |
| with people | 0 | 0 | 0 | 0 | 0 | 0 | 126 | 0 | 39 | 173 | 173 | 0 | 56 | 0 | 53 | 0 | 0 | 129 | 0 | 50 | 58 | 0 | 0 |
| on beach | 285 | 0 | 0 | 0 | 0 | 0 | 167 | 0 | 103 | 133 | 122 | 46 | 0 | 0 | 0 | 0 | 97 | 106 | 0 | 0 | 111 | 0 | 50 |
| small | 113 | 38 | 64 | 144 | 215 | 30 | 19 | 70 | 47 | 0 | 0 | 24 | 0 | 30 | 0 | 30 | 0 | 82 | 0 | 0 | 0 | 0 | 0 |
| in forest | 0 | 0 | 0 | 0 | 0 | 246 | 146 | 0 | 0 | 0 | 0 | 101 | 134 | 162 | 0 | 0 | 95 | 0 | 0 | 95 | 126 | 0 | 140 |
| on road | 87 | 179 | 0 | 0 | 0 | 0 | 77 | 0 | 149 | 241 | 338 | 0 | 0 | 91 | 0 | 0 | 149 | 0 | 0 | 111 | 0 | 0 | 0 |
| silver | 0 | 0 | 0 | 21 | 0 | 0 | 0 | 108 | 111 | 0 | 0 | 67 | 0 | 0 | 0 | 0 | 0 | 20 | 0 | 0 | 0 | 24 | 0 |
| running | 130 | 0 | 0 | 0 | 0 | 0 | 182 | 0 | 0 | 0 | 0 | 0 | 0 | 0 | 0 | 0 | 0 | 0 | 0 | 0 | 0 | 0 | 0 |
| in cage | 122 | 110 | 90 | 0 | 0 | 0 | 0 | 0 | 0 | 0 | 0 | 0 | 0 | 0 | 0 | 0 | 0 | 0 | 0 | 0 | 0 | 0 | 77 |
| open | 0 | 0 | 0 | 0 | 0 | 0 | 0 | 0 | 0 | 0 | 0 | 0 | 0 | 0 | 0 | 0 | 0 | 0 | 0 | 0 | 0 | 490 | 0 |
| yellow | 0 | 0 | 0 | 28 | 0 | 0 | 0 | 20 | 33 | 0 | 0 | 98 | 0 | 0 | 75 | 0 | 34 | 0 | 0 | 0 | 0 | 55 | 0 |
| standing | 74 | 38 | 167 | 0 | 0 | 0 | 32 | 63 | 0 | 0 | 0 | 0 | 173 | 200 | 0 | 0 | 0 | 0 | 0 | 0 | 0 | 0 | 0 |
| in city | 0 | 0 | 0 | 0 | 0 | 0 | 0 | 0 | 149 | 0 | 197 | 0 | 0 | 0 | 201 | 0 | 75 | 59 | 0 | 0 | 71 | 0 | 0 |
| sitting | 113 | 140 | 25 | 0 | 0 | 27 | 0 | 0 | 0 | 0 | 0 | 0 | 0 | 0 | 0 | 0 | 0 | 0 | 0 | 0 | 0 | 0 | 168 |
| tall | 0 | 0 | 0 | 0 | 641 | 0 | 0 | 166 | 0 | 0 | 0 | 0 | 0 | 0 | 0 | 0 | 0 | 0 | 0 | 0 | 0 | 0 | 0 |
| in river | 0 | 146 | 0 | 0 | 0 | 0 | 73 | 0 | 0 | 0 | 0 | 0 | 139 | 176 | 0 | 0 | 0 | 0 | 0 | 0 | 0 | 0 | 0 |
| in water | 140 | 50 | 81 | 0 | 0 | 174 | 0 | 0 | 0 | 0 | 0 | 0 | 0 | 0 | 0 | 0 | 0 | 0 | 0 | 71 | 0 | 0 | 117 |
| dark | 0 | 0 | 98 | 38 | 0 | 0 | 19 | 0 | 29 | 0 | 0 | 0 | 0 | 0 | 0 | 21 | 0 | 164 | 0 | 0 | 26 | 0 | 0 |
| at home | 93 | 279 | 0 | 0 | 0 | 0 | 90 | 0 | 0 | 0 | 0 | 0 | 78 | 0 | 0 | 0 | 0 | 0 | 0 | 0 | 0 | 0 | 0 |
| walking | 24 | 141 | 0 | 0 | 0 | 39 | 35 | 0 | 0 | 0 | 0 | 0 | 0 | 87 | 0 | 0 | 0 | 0 | 0 | 88 | 0 | 0 | 100 |
| glass | 0 | 0 | 0 | 71 | 0 | 0 | 0 | 0 | 0 | 0 | 0 | 0 | 0 | 0 | 0 | 0 | 0 | 0 | 0 | 0 | 0 | 211 | 0 |
| orange | 0 | 107 | 0 | 16 | 0 | 0 | 0 | 45 | 0 | 0 | 0 | 37 | 0 | 0 | 41 | 32 | 23 | 0 | 0 | 0 | 0 | 0 | 0 |
| on tree | 0 | 50 | 0 | 0 | 0 | 71 | 0 | 0 | 0 | 0 | 0 | 0 | 0 | 0 | 0 | 0 | 0 | 0 | 0 | 0 | 0 | 0 | 0 |
| in hand | 0 | 0 | 95 | 0 | 0 | 0 | 0 | 0 | 0 | 0 | 0 | 0 | 0 | 0 | 0 | 0 | 0 | 0 | 0 | 0 | 0 | 0 | 0 |
| flying | 0 | 0 | 245 | 0 | 0 | 0 | 0 | 0 | 0 | 0 | 0 | 0 | 0 | 0 | 0 | 0 | 0 | 0 | 0 | 0 | 0 | 0 | 0 |
| plastic | 0 | 0 | 0 | 22 | 0 | 0 | 0 | 0 | 0 | 0 | 0 | 0 | 0 | 0 | 0 | 61 | 0 | 0 | 0 | 0 | 0 | 0 | 0 |
| long | 0 | 0 | 0 | 0 | 0 | 0 | 0 | 130 | 0 | 0 | 0 | 195 | 0 | 0 | 0 | 0 | 0 | 0 | 79 | 0 | 0 | 0 | 0 |
| closed | 0 | 0 | 0 | 0 | 0 | 0 | 0 | 0 | 0 | 0 | 0 | 0 | 0 | 0 | 0 | 0 | 0 | 0 | 0 | 0 | 0 | 176 | 0 |
| on ground | 0 | 0 | 278 | 0 | 0 | 98 | 0 | 0 | 0 | 0 | 0 | 0 | 0 | 0 | 0 | 0 | 0 | 0 | 0 | 0 | 0 | 0 | 0 |
| aside mountain | 0 | 0 | 0 | 0 | 0 | 0 | 0 | 0 | 0 | 0 | 0 | 63 | 0 | 0 | 0 | 0 | 62 | 76 | 0 | 0 | 163 | 0 | 0 |
| pink | 0 | 0 | 0 | 0 | 0 | 0 | 0 | 0 | 0 | 0 | 0 | 0 | 0 | 0 | 0 | 0 | 0 | 0 | 0 | 0 | 0 | 0 | 0 |
| round | 0 | 0 | 0 | 187 | 0 | 0 | 0 | 0 | 0 | 0 | 0 | 0 | 0 | 0 | 0 | 0 | 0 | 0 | 0 | 0 | 0 | 0 | 0 |
| bare | 0 | 0 | 0 | 0 | 329 | 0 | 0 | 0 | 0 | 0 | 0 | 0 | 0 | 0 | 0 | 0 | 0 | 0 | 0 | 0 | 0 | 0 | 0 |
| tan | 27 | 0 | 0 | 38 | 0 | 0 | 0 | 0 | 0 | 0 | 0 | 0 | 0 | 0 | 0 | 26 | 0 | 0 | 110 | 0 | 0 | 0 | 0 |
| baby | 0 | 0 | 0 | 0 | 0 | 0 | 0 | 0 | 0 | 0 | 0 | 0 | 0 | 0 | 0 | 0 | 0 | 0 | 0 | 0 | 0 | 0 | 0 |
| in garage | 0 | 0 | 0 | 0 | 0 | 0 | 0 | 0 | 0 | 145 | 150 | 0 | 0 | 0 | 0 | 0 | 0 | 0 | 0 | 0 | 0 | 0 | 0 |
| yacht | 0 | 0 | 0 | 0 | 0 | 0 | 0 | 0 | 0 | 0 | 0 | 0 | 0 | 0 | 0 | 0 | 0 | 0 | 0 | 0 | 0 | 0 | 0 |
| on track | 0 | 0 | 0 | 0 | 0 | 0 | 0 | 0 | 88 | 0 | 187 | 0 | 0 | 0 | 0 | 0 | 0 | 0 | 0 | 0 | 0 | 0 | 0 |
| at station | 0 | 0 | 0 | 0 | 0 | 0 | 0 | 0 | 0 | 0 | 0 | 158 | 0 | 0 | 97 | 0 | 0 | 0 | 0 | 0 | 0 | 0 | 0 |
| wooden | 0 | 0 | 0 | 0 | 0 | 0 | 0 | 0 | 0 | 0 | 0 | 0 | 0 | 0 | 0 | 0 | 0 | 0 | 0 | 0 | 0 | 0 | 0 |
| in farm | 0 | 0 | 0 | 0 | 0 | 0 | 0 | 0 | 0 | 0 | 0 | 0 | 0 | 0 | 0 | 0 | 0 | 0 | 0 | 0 | 0 | 0 | 0 |
| on branch | 0 | 0 | 240 | 0 | 0 | 0 | 0 | 0 | 0 | 0 | 0 | 0 | 0 | 0 | 0 | 0 | 0 | 0 | 0 | 0 | 0 | 0 | 0 |
| leafy | 0 | 0 | 0 | 0 | 236 | 0 | 0 | 0 | 0 | 0 | 0 | 0 | 0 | 0 | 0 | 0 | 0 | 0 | 0 | 0 | 0 | 0 | 0 |
| shared | 0 | 0 | 0 | 0 | 0 | 0 | 0 | 0 | 0 | 233 | 0 | 0 | 0 | 0 | 0 | 0 | 0 | 0 | 0 | 0 | 0 | 0 | 0 |
| velodrome | 0 | 0 | 0 | 0 | 0 | 0 | 0 | 0 | 0 | 229 | 0 | 0 | 0 | 0 | 0 | 0 | 0 | 0 | 0 | 0 | 0 | 0 | 0 |
| at wharf | 0 | 0 | 0 | 0 | 0 | 0 | 0 | 0 | 0 | 0 | 0 | 0 | 0 | 0 | 0 | 0 | 0 | 0 | 0 | 0 | 0 | 0 | 0 |
| double decker | 0 | 0 | 0 | 0 | 0 | 0 | 0 | 0 | 0 | 0 | 0 | 0 | 0 | 0 | 223 | 0 | 0 | 0 | 0 | 0 | 0 | 0 | 0 |
| on | 0 | 0 | 0 | 0 | 0 | 0 | 0 | 0 | 0 | 0 | 0 | 0 | 0 | 0 | 0 | 0 | 0 | 0 | 0 | 0 | 0 | 0 | 0 |
| cross bridge | 0 | 0 | 0 | 0 | 0 | 0 | 0 | 0 | 0 | 0 | 0 | 0 | 0 | 0 | 0 | 0 | 0 | 0 | 0 | 0 | 0 | 0 | 0 |
| in sunset | 0 | 0 | 0 | 0 | 0 | 0 | 0 | 0 | 0 | 0 | 0 | 0 | 0 | 0 | 0 | 0 | 0 | 0 | 0 | 0 | 0 | 0 | 0 |
| at heliport | 0 | 0 | 0 | 0 | 0 | 0 | 0 | 0 | 0 | 0 | 0 | 0 | 0 | 0 | 0 | 0 | 0 | 0 | 0 | 0 | 188 | 0 | 0 |
| on bridge | 0 | 0 | 0 | 0 | 0 | 0 | 0 | 0 | 36 | 0 | 0 | 56 | 0 | 0 | 45 | 0 | 44 | 0 | 0 | 0 | 0 | 0 | 0 |
| in burrow | 0 | 0 | 0 | 0 | 0 | 0 | 0 | 0 | 0 | 0 | 0 | 0 | 0 | 0 | 0 | 0 | 0 | 0 | 0 | 0 | 0 | 0 | 0 |
| aside tree | 0 | 0 | 0 | 0 | 0 | 0 | 0 | 0 | 0 | 0 | 0 | 0 | 0 | 0 | 169 | 0 | 0 | 0 | 0 | 0 | 0 | 0 | 0 |
| in zoo | 0 | 0 | 0 | 0 | 0 | 0 | 0 | 0 | 0 | 0 | 0 | 0 | 0 | 160 | 0 | 0 | 0 | 0 | 0 | 0 | 0 | 0 | 0 |
| in pot | 0 | 0 | 0 | 0 | 0 | 0 | 0 | 0 | 0 | 0 | 0 | 0 | 0 | 0 | 0 | 0 | 0 | 0 | 0 | 0 | 0 | 0 | 0 |
| at airport | 0 | 0 | 0 | 0 | 0 | 0 | 0 | 0 | 0 | 0 | 0 | 0 | 0 | 0 | 0 | 0 | 0 | 156 | 0 | 0 | 0 | 0 | 0 |
| on sea | 0 | 0 | 0 | 0 | 0 | 0 | 0 | 0 | 0 | 0 | 0 | 0 | 0 | 0 | 0 | 0 | 0 | 0 | 0 | 0 | 155 | 0 | 0 |
| sleeping | 36 | 110 | 0 | 0 | 0 | 0 | 0 | 0 | 0 | 0 | 0 | 0 | 0 | 0 | 0 | 0 | 0 | 0 | 0 | 0 | 0 | 0 | 0 |
| in desert | 0 | 0 | 0 | 0 | 0 | 0 | 0 | 0 | 0 | 0 | 0 | 0 | 0 | 0 | 0 | 0 | 0 | 0 | 0 | 0 | 0 | 0 | 0 |
| eating grass | 0 | 0 | 0 | 0 | 0 | 133 | 0 | 0 | 0 | 0 | 0 | 0 | 0 | 0 | 0 | 0 | 0 | 0 | 0 | 0 | 0 | 0 | 0 |
| in race | 0 | 0 | 0 | 0 | 0 | 0 | 0 | 0 | 0 | 0 | 0 | 0 | 0 | 0 | 0 | 0 | 130 | 0 | 68 | 0 | 0 | 0 | 0 |
| taking off | 0 | 0 | 0 | 0 | 0 | 0 | 0 | 0 | 0 | 0 | 0 | 0 | 0 | 0 | 0 | 0 | 0 | 129 | 0 | 0 | 0 | 0 | 0 |
| on power line | 0 | 0 | 0 | 0 | 0 | 0 | 0 | 0 | 0 | 0 | 0 | 0 | 0 | 0 | 0 | 0 | 0 | 0 | 0 | 0 | 0 | 0 | 0 |
| in circus | 0 | 0 | 0 | 0 | 0 | 0 | 0 | 0 | 0 | 0 | 0 | 0 | 0 | 115 | 0 | 0 | 0 | 0 | 0 | 0 | 0 | 0 | 0 |
| mountain | 0 | 0 | 0 | 0 | 0 | 0 | 0 | 0 | 0 | 0 | 0 | 0 | 0 | 0 | 0 | 0 | 0 | 0 | 0 | 0 | 0 | 0 | 0 |
| short | 0 | 0 | 0 | 0 | 25 | 0 | 0 | 60 | 0 | 0 | 0 | 28 | 0 | 0 | 0 | 0 | 0 | 0 | 0 | 0 | 0 | 0 | 0 |

| Contexts | dog | cat | bird | table | tree | bear | horse | fence | car | bicycle | motorcycle | train | cow | elephant | bus | chair | truck | airplane | pants | sheep | helicopter | door | monkey |
|---|---|---|---|---|---|---|---|---|---|---|---|---|---|---|---|---|---|---|---|---|---|---|---|
| open mouth | 0 | 0 | 0 | 0 | 0 | 0 | 0 | 0 | 0 | 0 | 0 | 0 | 0 | 0 | 0 | 0 | 0 | 0 | 0 | 0 | 0 | 0 | 0 |
| on booth | 0 | 0 | 0 | 0 | 0 | 0 | 0 | 0 | 112 | 0 | 0 | 0 | 0 | 0 | 0 | 0 | 0 | 0 | 0 | 0 | 0 | 0 | 0 |
| howling | 0 | 0 | 0 | 0 | 0 | 0 | 0 | 0 | 0 | 0 | 0 | 0 | 0 | 0 | 0 | 0 | 0 | 0 | 0 | 0 | 0 | 0 | 0 |
| brick | 0 | 0 | 0 | 0 | 0 | 0 | 0 | 0 | 0 | 0 | 0 | 0 | 0 | 0 | 0 | 0 | 0 | 0 | 0 | 0 | 0 | 0 | 0 |
| khaki | 0 | 0 | 0 | 0 | 0 | 0 | 0 | 0 | 0 | 0 | 0 | 0 | 0 | 0 | 0 | 0 | 0 | 0 | 71 | 0 | 0 | 0 | 0 |
| at dock | 0 | 0 | 0 | 0 | 0 | 0 | 0 | 0 | 0 | 0 | 0 | 0 | 0 | 0 | 0 | 0 | 0 | 0 | 0 | 0 | 0 | 0 | 0 |
| on head | 0 | 0 | 0 | 0 | 0 | 0 | 0 | 0 | 0 | 0 | 0 | 0 | 0 | 0 | 0 | 0 | 0 | 0 | 0 | 0 | 0 | 0 | 0 |
| around cloud | 0 | 0 | 0 | 0 | 0 | 0 | 0 | 0 | 0 | 0 | 0 | 0 | 0 | 0 | 0 | 0 | 0 | 88 | 0 | 0 | 0 | 0 | 0 |
| climbing | 0 | 0 | 0 | 0 | 0 | 0 | 0 | 0 | 0 | 0 | 0 | 0 | 0 | 0 | 0 | 0 | 0 | 0 | 0 | 0 | 0 | 0 | 88 |
| stone | 0 | 0 | 0 | 0 | 0 | 0 | 0 | 0 | 0 | 0 | 0 | 0 | 0 | 0 | 0 | 0 | 0 | 0 | 0 | 0 | 0 | 0 | 0 |
| on web | 0 | 0 | 0 | 0 | 0 | 0 | 0 | 0 | 0 | 0 | 0 | 0 | 0 | 0 | 0 | 0 | 0 | 0 | 0 | 0 | 0 | 0 | 0 |
| on a stick | 0 | 0 | 0 | 0 | 0 | 0 | 0 | 0 | 0 | 0 | 0 | 0 | 0 | 0 | 0 | 0 | 0 | 0 | 0 | 0 | 0 | 0 | 0 |
| leather | 0 | 0 | 0 | 0 | 0 | 0 | 0 | 0 | 0 | 0 | 0 | 0 | 0 | 0 | 0 | 58 | 0 | 0 | 0 | 0 | 0 | 0 | 0 |
| cutting | 0 | 0 | 0 | 0 | 0 | 0 | 0 | 0 | 0 | 0 | 0 | 0 | 0 | 0 | 0 | 0 | 0 | 0 | 0 | 0 | 0 | 0 | 0 |
| concrete | 0 | 0 | 0 | 0 | 0 | 0 | 0 | 0 | 0 | 0 | 0 | 0 | 0 | 0 | 0 | 0 | 0 | 0 | 0 | 0 | 0 | 0 | 0 |
| at park | 0 | 0 | 0 | 0 | 0 | 0 | 0 | 0 | 81 | 0 | 0 | 0 | 0 | 0 | 0 | 0 | 0 | 0 | 0 | 0 | 0 | 0 | 0 |
| on iceberg | 0 | 0 | 0 | 0 | 0 | 0 | 0 | 0 | 0 | 0 | 0 | 0 | 0 | 0 | 0 | 0 | 0 | 0 | 0 | 0 | 0 | 0 | 0 |
| on shoulder | 0 | 0 | 80 | 0 | 0 | 0 | 0 | 0 | 0 | 0 | 0 | 0 | 0 | 0 | 0 | 0 | 0 | 0 | 0 | 0 | 0 | 0 | 0 |
| at night | 0 | 0 | 0 | 0 | 0 | 0 | 0 | 0 | 0 | 0 | 0 | 0 | 0 | 0 | 0 | 0 | 0 | 77 | 0 | 0 | 0 | 0 | 0 |
| with flower | 0 | 0 | 0 | 0 | 0 | 0 | 0 | 0 | 0 | 0 | 0 | 0 | 0 | 0 | 0 | 0 | 0 | 0 | 0 | 0 | 0 | 0 | 0 |
| grazing | 0 | 0 | 0 | 0 | 0 | 0 | 22 | 0 | 0 | 0 | 0 | 0 | 0 | 0 | 0 | 0 | 0 | 0 | 0 | 0 | 0 | 0 | 0 |
| paper | 0 | 0 | 0 | 0 | 0 | 0 | 0 | 0 | 0 | 0 | 0 | 0 | 0 | 0 | 0 | 0 | 0 | 0 | 0 | 0 | 0 | 0 | 0 |
| spotted | 0 | 0 | 0 | 0 | 0 | 0 | 0 | 0 | 0 | 0 | 0 | 0 | 75 | 0 | 0 | 0 | 0 | 0 | 0 | 0 | 0 | 0 | 0 |
| at yard | 0 | 0 | 0 | 0 | 0 | 0 | 0 | 0 | 0 | 0 | 0 | 0 | 0 | 0 | 75 | 0 | 0 | 0 | 0 | 0 | 0 | 0 | 0 |
| with bee | 0 | 0 | 0 | 0 | 0 | 0 | 0 | 0 | 0 | 0 | 0 | 0 | 0 | 0 | 0 | 0 | 0 | 0 | 0 | 0 | 0 | 0 | 0 |
| hanging | 0 | 0 | 0 | 0 | 0 | 0 | 0 | 0 | 0 | 0 | 0 | 0 | 0 | 0 | 0 | 0 | 0 | 0 | 0 | 0 | 0 | 0 | 0 |
| clean | 0 | 0 | 0 | 0 | 0 | 0 | 0 | 0 | 0 | 0 | 0 | 0 | 0 | 0 | 0 | 0 | 0 | 0 | 0 | 0 | 0 | 0 | 0 |
| square | 0 | 0 | 0 | 38 | 0 | 0 | 0 | 0 | 0 | 0 | 0 | 0 | 0 | 0 | 0 | 0 | 0 | 0 | 0 | 0 | 0 | 0 | 0 |
| subway | 0 | 0 | 0 | 0 | 0 | 0 | 0 | 0 | 0 | 0 | 0 | 71 | 0 | 0 | 0 | 0 | 0 | 0 | 0 | 0 | 0 | 0 | 0 |
| jumping | 41 | 0 | 0 | 0 | 0 | 0 | 29 | 0 | 0 | 0 | 0 | 0 | 0 | 0 | 0 | 0 | 0 | 0 | 0 | 0 | 0 | 0 | 0 |
| on flower | 0 | 0 | 0 | 0 | 0 | 0 | 0 | 0 | 0 | 0 | 0 | 0 | 0 | 0 | 0 | 0 | 0 | 0 | 0 | 0 | 0 | 0 | 0 |
| landing | 0 | 0 | 0 | 0 | 0 | 0 | 0 | 0 | 0 | 0 | 0 | 0 | 0 | 0 | 0 | 0 | 0 | 0 | 0 | 0 | 0 | 0 | 0 |
| porcelain | 0 | 0 | 0 | 0 | 0 | 0 | 0 | 0 | 0 | 0 | 0 | 0 | 0 | 0 | 0 | 0 | 0 | 0 | 0 | 0 | 0 | 0 | 0 |
| at sunset | 0 | 0 | 0 | 0 | 0 | 0 | 0 | 0 | 0 | 0 | 0 | 0 | 0 | 0 | 0 | 0 | 0 | 0 | 0 | 66 | 0 | 0 | 0 |
| on desk | 0 | 0 | 0 | 0 | 0 | 0 | 0 | 0 | 0 | 0 | 0 | 0 | 0 | 0 | 0 | 0 | 0 | 0 | 0 | 0 | 0 | 0 | 0 |
| fighting | 0 | 0 | 0 | 0 | 0 | 0 | 0 | 0 | 0 | 0 | 0 | 0 | 0 | 0 | 0 | 0 | 0 | 0 | 0 | 0 | 0 | 0 | 0 |
| purple | 0 | 0 | 0 | 0 | 0 | 0 | 0 | 0 | 0 | 0 | 0 | 0 | 0 | 0 | 0 | 0 | 0 | 0 | 0 | 0 | 0 | 0 | 0 |
| off | 0 | 0 | 0 | 0 | 0 | 0 | 0 | 0 | 0 | 0 | 0 | 0 | 0 | 0 | 0 | 0 | 0 | 0 | 0 | 0 | 0 | 0 | 0 |
| gold | 0 | 0 | 0 | 0 | 0 | 0 | 0 | 0 | 0 | 0 | 0 | 0 | 0 | 0 | 0 | 0 | 0 | 0 | 0 | 0 | 0 | 0 | 0 |
| dirty | 0 | 0 | 0 | 0 | 0 | 0 | 0 | 0 | 0 | 0 | 0 | 0 | 0 | 0 | 0 | 0 | 0 | 0 | 0 | 0 | 0 | 0 | 0 |
| resting | 21 | 37 | 0 | 0 | 0 | 0 | 0 | 0 | 0 | 0 | 0 | 0 | 0 | 0 | 0 | 0 | 0 | 0 | 0 | 0 | 0 | 0 | 0 |
| beige | 0 | 0 | 0 | 0 | 0 | 0 | 0 | 0 | 0 | 0 | 0 | 0 | 0 | 0 | 0 | 0 | 0 | 0 | 0 | 0 | 0 | 0 | 0 |
| rectangular | 0 | 0 | 0 | 19 | 0 | 0 | 0 | 0 | 0 | 0 | 0 | 0 | 0 | 0 | 0 | 0 | 0 | 0 | 0 | 0 | 0 | 0 | 0 |
| thin | 0 | 0 | 0 | 0 | 37 | 0 | 0 | 0 | 0 | 0 | 0 | 0 | 0 | 0 | 0 | 0 | 0 | 0 | 0 | 0 | 0 | 0 | 0 |
| thick | 0 | 0 | 0 | 0 | 43 | 0 | 0 | 0 | 0 | 0 | 0 | 0 | 0 | 0 | 0 | 0 | 0 | 0 | 0 | 0 | 0 | 0 | 0 |
| in bucket | 0 | 0 | 0 | 0 | 0 | 0 | 0 | 0 | 0 | 0 | 0 | 0 | 0 | 0 | 0 | 0 | 0 | 0 | 0 | 0 | 0 | 0 | 0 |
| city | 0 | 0 | 0 | 0 | 0 | 0 | 0 | 0 | 0 | 0 | 0 | 0 | 0 | 0 | 0 | 0 | 0 | 0 | 0 | 0 | 0 | 0 | 0 |
| in pouch | 0 | 0 | 0 | 0 | 0 | 0 | 0 | 0 | 0 | 0 | 0 | 0 | 0 | 0 | 0 | 0 | 0 | 0 | 0 | 0 | 0 | 0 | 0 |
| in hole | 0 | 0 | 0 | 0 | 0 | 0 | 0 | 0 | 0 | 0 | 0 | 0 | 0 | 0 | 0 | 0 | 0 | 0 | 0 | 0 | 0 | 0 | 0 |
| on bird feeder | 0 | 0 | 0 | 0 | 0 | 0 | 0 | 0 | 0 | 0 | 0 | 0 | 0 | 0 | 0 | 0 | 0 | 0 | 0 | 0 | 0 | 0 | 0 |
| on shelves | 0 | 0 | 0 | 0 | 0 | 0 | 0 | 0 | 0 | 0 | 0 | 0 | 0 | 0 | 0 | 0 | 0 | 0 | 0 | 0 | 0 | 0 | 0 |
| on wall | 0 | 0 | 0 | 0 | 0 | 0 | 0 | 0 | 0 | 0 | 0 | 0 | 0 | 0 | 0 | 0 | 0 | 0 | 0 | 0 | 0 | 0 | 0 |
| on post | 0 | 0 | 0 | 0 | 0 | 0 | 0 | 0 | 0 | 0 | 0 | 0 | 0 | 0 | 0 | 0 | 0 | 0 | 0 | 0 | 0 | 0 | 0 |
| in box | 0 | 0 | 0 | 0 | 0 | 0 | 0 | 0 | 0 | 0 | 0 | 0 | 0 | 0 | 0 | 0 | 0 | 0 | 0 | 0 | 0 | 0 | 0 |
| shiny | 0 | 0 | 0 | 39 | 0 | 0 | 0 | 0 | 0 | 0 | 0 | 0 | 0 | 0 | 0 | 0 | 0 | 0 | 0 | 0 | 0 | 0 | 0 |
| sinking | 0 | 0 | 0 | 0 | 0 | 0 | 0 | 0 | 0 | 0 | 0 | 0 | 0 | 0 | 0 | 0 | 0 | 0 | 0 | 0 | 0 | 0 | 0 |
| with cargo | 0 | 0 | 0 | 0 | 0 | 0 | 0 | 0 | 0 | 0 | 0 | 0 | 0 | 0 | 0 | 0 | 0 | 0 | 0 | 0 | 0 | 0 | 0 |
| bright | 0 | 0 | 0 | 0 | 0 | 0 | 0 | 0 | 0 | 0 | 0 | 0 | 0 | 0 | 0 | 0 | 0 | 0 | 0 | 0 | 0 | 0 | 0 |
| in shell | 0 | 0 | 0 | 0 | 0 | 0 | 0 | 0 | 0 | 0 | 0 | 0 | 0 | 0 | 0 | 0 | 0 | 0 | 0 | 0 | 0 | 0 | 0 |
| aside traffic light | 0 | 0 | 0 | 0 | 0 | 0 | 0 | 0 | 0 | 0 | 0 | 0 | 0 | 0 | 36 | 0 | 0 | 0 | 0 | 0 | 0 | 0 | 0 |
| empty | 0 | 0 | 0 | 0 | 0 | 0 | 0 | 0 | 0 | 0 | 0 | 0 | 0 | 0 | 0 | 0 | 0 | 0 | 0 | 0 | 0 | 0 | 0 |
| cross tunnel | 0 | 0 | 0 | 0 | 0 | 0 | 0 | 0 | 0 | 0 | 0 | 36 | 0 | 0 | 0 | 0 | 0 | 0 | 0 | 0 | 0 | 0 | 0 |
| full | 0 | 0 | 0 | 0 | 0 | 0 | 0 | 0 | 0 | 0 | 0 | 0 | 0 | 0 | 0 | 0 | 0 | 0 | 0 | 0 | 0 | 0 | 0 |
| playing | 31 | 0 | 0 | 0 | 0 | 0 | 0 | 0 | 0 | 0 | 0 | 0 | 0 | 0 | 0 | 0 | 0 | 0 | 0 | 0 | 0 | 0 | 0 |
| light brown | 0 | 0 | 30 | 0 | 0 | 0 | 0 | 0 | 0 | 0 | 0 | 0 | 0 | 0 | 0 | 0 | 0 | 0 | 0 | 0 | 0 | 0 | 0 |
| staring | 0 | 29 | 0 | 0 | 0 | 0 | 0 | 0 | 0 | 0 | 0 | 0 | 0 | 0 | 0 | 0 | 0 | 0 | 0 | 0 | 0 | 0 | 0 |
| colorful | 0 | 0 | 0 | 0 | 0 | 0 | 0 | 0 | 0 | 0 | 0 | 0 | 0 | 0 | 0 | 0 | 0 | 0 | 0 | 0 | 0 | 0 | 0 |
| dark brown | 0 | 0 | 0 | 23 | 0 | 0 | 0 | 0 | 0 | 0 | 0 | 0 | 0 | 0 | 0 | 0 | 0 | 0 | 0 | 0 | 0 | 0 | 0 |
| young | 0 | 0 | 0 | 0 | 21 | 0 | 0 | 0 | 0 | 0 | 0 | 0 | 0 | 0 | 0 | 0 | 0 | 0 | 0 | 0 | 0 | 0 | 0 |
| dark blue | 0 | 0 | 0 | 0 | 0 | 0 | 0 | 0 | 0 | 0 | 0 | 0 | 0 | 0 | 0 | 0 | 0 | 0 | 0 | 0 | 0 | 0 | 0 |

| Contexts | umbrella | lion | squirrel | boat | wolf | lizard | tiger | giraffe | tent | hot air balloon | owl | sailboat | seal | frog | jacket | rabbit | goose | kangaroo | flower | ship | cactus | hat | fox |
|---|---|---|---|---|---|---|---|---|---|---|---|---|---|---|---|---|---|---|---|---|---|---|---|
| grass | 460 | 437 | 0 | 238 | 371 | 293 | 444 | 389 | 269 | 376 | 332 | 290 | 351 | 0 | 457 | 391 | 336 | 419 | 202 | 318 | 285 | 398 | 283 |
| water | 243 | 209 | 0 | 276 | 368 | 374 | 368 | 441 | 459 | 229 | 434 | 414 | 296 | 0 | 160 | 278 | 246 | 358 | 401 | 298 | 280 | 183 | 213 |
| outdoor | 299 | 265 | 0 | 324 | 344 | 369 | 214 | 288 | 367 | 197 | 402 | 269 | 103 | 0 | 165 | 349 | 196 | 341 | 378 | 203 | 404 | 160 | 267 |
| dim | 294 | 107 | 0 | 179 | 56 | 166 | 277 | 280 | 227 | 288 | 251 | 115 | 139 | 0 | 131 | 193 | 257 | 221 | 302 | 310 | 147 | 133 | 94 |
| white | 0 | 116 | 138 | 76 | 0 | 65 | 33 | 74 | 0 | 107 | 0 | 95 | 0 | 146 | 131 | 94 | 37 | 20 | 0 | 0 | 82 | 91 | 147 |
| rock | 253 | 241 | 0 | 265 | 344 | 201 | 149 | 250 | 254 | 123 | 236 | 272 | 258 | 0 | 122 | 86 | 110 | 322 | 203 | 205 | 73 | 151 | 84 |
| autumn | 121 | 272 | 0 | 234 | 130 | 121 | 121 | 265 | 147 | 193 | 135 | 32 | 205 | 0 | 125 | 145 | 76 | 128 | 225 | 44 | 210 | 216 | 252 |
| black | 0 | 0 | 0 | 0 | 0 | 0 | 0 | 0 | 0 | 0 | 0 | 0 | 0 | 631 | 0 | 0 | 0 | 0 | 0 | 0 | 115 | 0 | 65 |
| brown | 0 | 0 | 0 | 0 | 0 | 0 | 229 | 0 | 0 | 0 | 0 | 0 | 0 | 135 | 0 | 0 | 0 | 0 | 0 | 0 | 45 | 0 | 22 |
| wood | 0 | 0 | 0 | 0 | 0 | 0 | 0 | 0 | 0 | 0 | 0 | 0 | 0 | 0 | 0 | 0 | 0 | 0 | 0 | 0 | 0 | 0 | 28 |
| blue | 0 | 0 | 86 | 0 | 88 | 0 | 0 | 36 | 0 | 0 | 0 | 0 | 104 | 348 | 0 | 0 | 0 | 0 | 0 | 0 | 53 | 0 | 23 |
| green | 0 | 0 | 0 | 0 | 110 | 0 | 0 | 15 | 0 | 0 | 0 | 0 | 106 | 114 | 0 | 0 | 0 | 0 | 0 | 0 | 0 | 0 | 0 |
| gray | 0 | 0 | 0 | 0 | 0 | 0 | 0 | 0 | 0 | 0 | 0 | 0 | 0 | 162 | 119 | 0 | 0 | 0 | 0 | 0 | 25 | 0 | 0 |
| on snow | 89 | 135 | 0 | 81 | 0 | 107 | 0 | 0 | 0 | 74 | 0 | 125 | 0 | 0 | 139 | 0 | 71 | 0 | 0 | 0 | 0 | 97 | 0 |
| large | 0 | 0 | 58 | 0 | 0 | 0 | 58 | 0 | 0 | 0 | 0 | 0 | 0 | 0 | 0 | 0 | 0 | 0 | 0 | 0 | 16 | 0 | 101 |
| eating | 85 | 127 | 0 | 112 | 70 | 45 | 0 | 0 | 0 | 73 | 0 | 93 | 76 | 0 | 110 | 63 | 53 | 0 | 0 | 0 | 0 | 26 | 0 |
| red | 0 | 0 | 0 | 0 | 0 | 0 | 0 | 0 | 0 | 0 | 0 | 0 | 0 | 136 | 0 | 0 | 0 | 0 | 0 | 0 | 35 | 0 | 0 |
| metal | 0 | 0 | 0 | 0 | 0 | 0 | 0 | 0 | 0 | 0 | 0 | 0 | 0 | 0 | 0 | 0 | 0 | 0 | 0 | 0 | 0 | 0 | 24 |
| lying | 139 | 116 | 0 | 151 | 0 | 86 | 0 | 0 | 0 | 0 | 0 | 0 | 0 | 0 | 0 | 81 | 0 | 0 | 0 | 0 | 0 | 88 | 0 |
| with people | 40 | 25 | 158 | 34 | 0 | 20 | 0 | 0 | 61 | 19 | 69 | 0 | 0 | 0 | 32 | 0 | 63 | 0 | 0 | 0 | 0 | 0 | 0 |
| on beach | 0 | 0 | 168 | 0 | 0 | 0 | 0 | 0 | 0 | 0 | 67 | 50 | 0 | 0 | 0 | 0 | 0 | 0 | 0 | 0 | 0 | 0 | 0 |
| small | 0 | 0 | 71 | 0 | 0 | 0 | 17 | 0 | 0 | 0 | 0 | 0 | 0 | 0 | 0 | 0 | 0 | 0 | 109 | 0 | 0 | 0 | 0 |
| in forest | 0 | 0 | 0 | 0 | 0 | 0 | 0 | 0 | 0 | 0 | 0 | 0 | 0 | 0 | 0 | 0 | 102 | 0 | 0 | 0 | 0 | 0 | 0 |
| on road | 0 | 0 | 0 | 0 | 0 | 0 | 0 | 0 | 0 | 0 | 0 | 0 | 0 | 0 | 0 | 0 | 0 | 0 | 0 | 0 | 0 | 0 | 0 |
| silver | 0 | 0 | 0 | 0 | 0 | 0 | 0 | 0 | 0 | 0 | 0 | 0 | 0 | 0 | 0 | 0 | 0 | 0 | 0 | 0 | 0 | 0 | 0 |
| running | 99 | 94 | 0 | 69 | 74 | 118 | 0 | 0 | 0 | 39 | 0 | 0 | 0 | 66 | 0 | 48 | 0 | 0 | 0 | 0 | 0 | 36 | 0 |
| in cage | 36 | 0 | 0 | 93 | 54 | 59 | 0 | 0 | 0 | 45 | 0 | 24 | 0 | 0 | 65 | 65 | 0 | 0 | 0 | 0 | 0 | 68 | 0 |
| open | 0 | 0 | 0 | 0 | 0 | 0 | 0 | 0 | 0 | 0 | 0 | 0 | 0 | 0 | 0 | 0 | 0 | 0 | 0 | 0 | 0 | 0 | 0 |
| yellow | 0 | 0 | 0 | 0 | 110 | 0 | 0 | 0 | 0 | 0 | 0 | 0 | 150 | 56 | 0 | 0 | 0 | 0 | 0 | 0 | 0 | 0 | 0 |
| standing | 0 | 0 | 0 | 0 | 0 | 0 | 153 | 0 | 0 | 0 | 0 | 0 | 0 | 0 | 0 | 0 | 0 | 0 | 0 | 0 | 0 | 0 | 0 |
| in city | 0 | 0 | 202 | 0 | 0 | 0 | 0 | 0 | 0 | 0 | 0 | 0 | 0 | 0 | 0 | 0 | 0 | 0 | 0 | 0 | 0 | 0 | 0 |
| sitting | 56 | 0 | 0 | 51 | 0 | 84 | 0 | 0 | 0 | 0 | 0 | 0 | 0 | 0 | 0 | 103 | 0 | 0 | 0 | 0 | 0 | 56 | 0 |
| tall | 0 | 0 | 0 | 0 | 0 | 0 | 0 | 0 | 0 | 0 | 0 | 0 | 0 | 0 | 0 | 0 | 0 | 0 | 0 | 0 | 0 | 0 | 0 |
| in river | 0 | 0 | 271 | 0 | 0 | 0 | 0 | 0 | 0 | 0 | 0 | 0 | 0 | 0 | 0 | 0 | 0 | 0 | 0 | 0 | 0 | 0 | 0 |
| in water | 0 | 0 | 0 | 0 | 0 | 0 | 0 | 0 | 0 | 0 | 0 | 0 | 0 | 0 | 0 | 0 | 0 | 0 | 0 | 0 | 0 | 0 | 0 |
| dark | 0 | 0 | 0 | 0 | 0 | 0 | 0 | 0 | 0 | 0 | 0 | 0 | 0 | 70 | 0 | 0 | 0 | 0 | 0 | 0 | 0 | 0 | 0 |
| at home | 0 | 0 | 0 | 0 | 0 | 0 | 0 | 0 | 0 | 0 | 0 | 0 | 0 | 0 | 0 | 0 | 0 | 0 | 0 | 0 | 0 | 0 | 0 |
| walking | 0 | 0 | 0 | 0 | 0 | 0 | 52 | 0 | 0 | 0 | 0 | 0 | 0 | 0 | 0 | 0 | 0 | 0 | 0 | 0 | 0 | 0 | 0 |
| glass | 0 | 0 | 0 | 0 | 0 | 0 | 0 | 0 | 0 | 0 | 0 | 0 | 0 | 0 | 0 | 0 | 0 | 0 | 0 | 0 | 0 | 0 | 0 |
| orange | 0 | 0 | 0 | 0 | 0 | 0 | 0 | 0 | 0 | 0 | 0 | 0 | 0 | 43 | 0 | 0 | 0 | 0 | 0 | 0 | 0 | 0 | 0 |
| on tree | 132 | 0 | 0 | 0 | 0 | 0 | 0 | 0 | 0 | 113 | 0 | 0 | 85 | 0 | 0 | 0 | 0 | 0 | 0 | 0 | 0 | 0 | 0 |
| in hand | 0 | 0 | 0 | 0 | 96 | 0 | 0 | 0 | 0 | 0 | 0 | 0 | 70 | 0 | 0 | 0 | 0 | 0 | 0 | 0 | 0 | 0 | 0 |
| flying | 0 | 0 | 0 | 0 | 0 | 0 | 0 | 0 | 0 | 126 | 0 | 0 | 0 | 0 | 95 | 0 | 0 | 0 | 0 | 0 | 0 | 0 | 0 |
| plastic | 0 | 0 | 0 | 0 | 0 | 0 | 0 | 0 | 0 | 0 | 0 | 0 | 0 | 0 | 0 | 0 | 0 | 0 | 0 | 0 | 0 | 0 | 0 |
| long | 0 | 0 | 0 | 0 | 0 | 0 | 0 | 0 | 0 | 0 | 0 | 0 | 0 | 0 | 0 | 0 | 0 | 0 | 0 | 0 | 0 | 0 | 0 |
| closed | 0 | 0 | 0 | 0 | 0 | 0 | 0 | 0 | 0 | 0 | 0 | 0 | 0 | 0 | 0 | 0 | 0 | 0 | 0 | 0 | 0 | 0 | 0 |
| on ground | 0 | 0 | 0 | 0 | 0 | 0 | 0 | 0 | 0 | 0 | 0 | 0 | 0 | 0 | 0 | 0 | 0 | 0 | 0 | 0 | 0 | 0 | 0 |
| aside mountain | 0 | 0 | 0 | 0 | 0 | 0 | 0 | 0 | 0 | 0 | 0 | 0 | 0 | 0 | 0 | 0 | 0 | 0 | 0 | 0 | 0 | 0 | 0 |
| pink | 0 | 0 | 0 | 0 | 0 | 0 | 0 | 0 | 0 | 0 | 0 | 0 | 0 | 51 | 0 | 0 | 0 | 26 | 0 | 0 | 0 | 0 | 0 |
| round | 0 | 0 | 0 | 0 | 0 | 0 | 0 | 0 | 0 | 0 | 0 | 0 | 0 | 0 | 0 | 0 | 0 | 0 | 0 | 0 | 0 | 0 | 103 |
| bare | 0 | 0 | 0 | 0 | 0 | 0 | 0 | 0 | 0 | 0 | 0 | 0 | 0 | 0 | 0 | 0 | 0 | 0 | 0 | 0 | 0 | 0 | 0 |
| tan | 0 | 0 | 0 | 0 | 0 | 0 | 0 | 0 | 0 | 0 | 0 | 0 | 0 | 0 | 0 | 0 | 0 | 0 | 0 | 0 | 0 | 0 | 0 |
| baby | 124 | 0 | 0 | 0 | 0 | 76 | 0 | 0 | 0 | 0 | 0 | 116 | 0 | 0 | 0 | 0 | 0 | 0 | 0 | 0 | 0 | 0 | 0 |
| in garage | 0 | 0 | 0 | 0 | 0 | 0 | 0 | 0 | 0 | 0 | 0 | 0 | 0 | 0 | 0 | 0 | 0 | 0 | 0 | 0 | 0 | 0 | 0 |
| yacht | 0 | 0 | 287 | 0 | 0 | 0 | 0 | 0 | 0 | 0 | 0 | 0 | 0 | 0 | 0 | 0 | 0 | 0 | 0 | 0 | 0 | 0 | 0 |
| on track | 0 | 0 | 0 | 0 | 0 | 0 | 0 | 0 | 0 | 0 | 0 | 0 | 0 | 0 | 0 | 0 | 0 | 0 | 0 | 0 | 0 | 0 | 0 |
| at station | 0 | 0 | 0 | 0 | 0 | 0 | 0 | 0 | 0 | 0 | 0 | 0 | 0 | 0 | 0 | 0 | 0 | 0 | 0 | 0 | 0 | 0 | 0 |
| wooden | 0 | 0 | 250 | 0 | 0 | 0 | 0 | 0 | 0 | 0 | 0 | 0 | 0 | 0 | 0 | 0 | 0 | 0 | 0 | 0 | 0 | 0 | 0 |
| in farm | 0 | 0 | 0 | 0 | 0 | 0 | 0 | 0 | 0 | 0 | 0 | 0 | 0 | 0 | 0 | 0 | 0 | 0 | 0 | 0 | 0 | 0 | 0 |
| on branch | 0 | 0 | 0 | 0 | 0 | 0 | 0 | 0 | 0 | 0 | 0 | 0 | 0 | 0 | 0 | 0 | 0 | 0 | 0 | 0 | 0 | 0 | 0 |
| leafy | 0 | 0 | 0 | 0 | 0 | 0 | 0 | 0 | 0 | 0 | 0 | 0 | 0 | 0 | 0 | 0 | 0 | 0 | 0 | 0 | 0 | 0 | 0 |
| shared | 0 | 0 | 0 | 0 | 0 | 0 | 0 | 0 | 0 | 0 | 0 | 0 | 0 | 0 | 0 | 0 | 0 | 0 | 0 | 0 | 0 | 0 | 0 |
| velodrome | 0 | 0 | 0 | 0 | 0 | 0 | 0 | 0 | 0 | 0 | 0 | 0 | 0 | 0 | 0 | 0 | 0 | 0 | 0 | 0 | 0 | 0 | 0 |
| at wharf | 0 | 0 | 226 | 0 | 0 | 0 | 0 | 0 | 0 | 0 | 0 | 0 | 0 | 0 | 0 | 0 | 0 | 0 | 0 | 0 | 0 | 0 | 0 |
| double decker | 0 | 0 | 0 | 0 | 0 | 0 | 0 | 0 | 0 | 0 | 0 | 0 | 0 | 0 | 0 | 0 | 0 | 0 | 0 | 0 | 0 | 0 | 0 |
| on | 0 | 0 | 0 | 0 | 0 | 0 | 0 | 0 | 0 | 0 | 0 | 0 | 0 | 0 | 0 | 0 | 0 | 0 | 0 | 0 | 0 | 0 | 0 |
| cross bridge | 0 | 0 | 198 | 0 | 0 | 0 | 0 | 0 | 0 | 0 | 0 | 0 | 0 | 0 | 0 | 0 | 0 | 0 | 0 | 0 | 0 | 0 | 0 |
| in sunset | 0 | 0 | 197 | 0 | 0 | 0 | 0 | 0 | 0 | 0 | 0 | 0 | 0 | 0 | 0 | 0 | 0 | 0 | 0 | 0 | 0 | 0 | 0 |
| at heliport | 0 | 0 | 0 | 0 | 0 | 0 | 0 | 0 | 0 | 0 | 0 | 0 | 0 | 0 | 0 | 0 | 0 | 0 | 0 | 0 | 0 | 0 | 0 |
| on bridge | 0 | 0 | 0 | 0 | 0 | 0 | 0 | 0 | 0 | 0 | 0 | 0 | 0 | 0 | 0 | 0 | 0 | 0 | 0 | 0 | 0 | 0 | 0 |
| in burrow | 0 | 0 | 0 | 0 | 0 | 0 | 0 | 0 | 0 | 0 | 0 | 0 | 0 | 0 | 55 | 0 | 0 | 0 | 0 | 0 | 0 | 48 | 0 |
| aside tree | 0 | 0 | 0 | 0 | 0 | 0 | 0 | 0 | 0 | 0 | 0 | 0 | 0 | 0 | 0 | 0 | 0 | 0 | 0 | 0 | 0 | 0 | 0 |
| in zoo | 0 | 0 | 0 | 0 | 0 | 0 | 0 | 0 | 0 | 0 | 0 | 0 | 0 | 0 | 0 | 0 | 0 | 0 | 0 | 0 | 0 | 0 | 0 |
| in pot | 0 | 0 | 0 | 0 | 0 | 0 | 0 | 0 | 0 | 0 | 0 | 0 | 0 | 0 | 0 | 0 | 0 | 0 | 0 | 0 | 99 | 0 | 0 |
| at airport | 0 | 0 | 0 | 0 | 0 | 0 | 0 | 0 | 0 | 0 | 0 | 0 | 0 | 0 | 0 | 0 | 0 | 0 | 0 | 0 | 0 | 0 | 0 |
| on sea | 0 | 0 | 0 | 0 | 0 | 0 | 0 | 0 | 0 | 0 | 0 | 0 | 0 | 0 | 0 | 0 | 0 | 0 | 0 | 0 | 0 | 0 | 0 |
| sleeping | 0 | 0 | 0 | 0 | 0 | 0 | 0 | 0 | 0 | 0 | 0 | 0 | 0 | 0 | 0 | 0 | 0 | 0 | 0 | 0 | 0 | 0 | 0 |
| in desert | 0 | 0 | 0 | 0 | 0 | 0 | 0 | 0 | 0 | 0 | 0 | 0 | 0 | 0 | 0 | 0 | 0 | 68 | 0 | 0 | 74 | 0 | 0 |
| eating grass | 0 | 0 | 0 | 0 | 0 | 0 | 0 | 0 | 0 | 0 | 0 | 0 | 0 | 0 | 0 | 0 | 0 | 0 | 0 | 0 | 0 | 0 | 0 |
| in race | 0 | 0 | 0 | 0 | 0 | 0 | 0 | 0 | 0 | 0 | 0 | 0 | 0 | 0 | 0 | 0 | 0 | 0 | 0 | 0 | 0 | 0 | 0 |
| taking off | 0 | 0 | 0 | 0 | 0 | 0 | 0 | 0 | 0 | 0 | 0 | 0 | 0 | 0 | 0 | 0 | 0 | 0 | 0 | 0 | 0 | 0 | 0 |
| on power line | 0 | 123 | 0 | 0 | 0 | 0 | 0 | 0 | 0 | 0 | 0 | 0 | 0 | 0 | 0 | 0 | 0 | 0 | 0 | 0 | 0 | 0 | 0 |
| in circus | 0 | 0 | 0 | 0 | 0 | 0 | 0 | 0 | 0 | 0 | 0 | 0 | 0 | 0 | 0 | 0 | 0 | 0 | 0 | 0 | 0 | 0 | 0 |
| mountain | 0 | 0 | 0 | 0 | 0 | 0 | 0 | 0 | 113 | 0 | 0 | 0 | 0 | 0 | 0 | 0 | 0 | 0 | 0 | 0 | 0 | 0 | 0 |
| short | 0 | 0 | 0 | 0 | 0 | 0 | 0 | 0 | 0 | 0 | 0 | 0 | 0 | 0 | 0 | 0 | 0 | 0 | 0 | 0 | 0 | 0 | 0 |

| Contexts | | | | | | | | | | Concepts | | | | | | | | | | | | | |
|---|---|---|---|---|---|---|---|---|---|---|---|---|---|---|---|---|---|---|---|---|---|---|---|
| | umbrella | lion | squirrel | boat | wolf | lizard | tiger | giraffe | tent | hot air balloon | owl | sailboat | seal | frog | jacket | rabbit | goose | kangaroo | flower | ship | cactus | hat | fox |
| open mouth | 0 | 0 | 0 | 0 | 0 | 0 | 0 | 0 | 0 | 0 | 0 | 0 | 0 | 0 | 0 | 0 | 0 | 0 | 0 | 0 | 0 | 0 | 0 |
| on booth | 0 | 0 | 0 | 0 | 0 | 0 | 0 | 0 | 0 | 0 | 0 | 0 | 0 | 0 | 0 | 0 | 0 | 0 | 0 | 0 | 0 | 0 | 0 |
| howling | 0 | 0 | 0 | 110 | 0 | 0 | 0 | 0 | 0 | 0 | 0 | 0 | 0 | 0 | 0 | 0 | 0 | 0 | 0 | 0 | 0 | 0 | 0 |
| brick | 0 | 0 | 0 | 0 | 0 | 0 | 0 | 0 | 0 | 0 | 0 | 0 | 0 | 0 | 0 | 0 | 0 | 0 | 0 | 0 | 0 | 0 | 0 |
| khaki | 0 | 0 | 0 | 0 | 0 | 0 | 0 | 0 | 0 | 0 | 0 | 0 | 0 | 0 | 0 | 0 | 0 | 0 | 0 | 0 | 0 | 0 | 0 |
| at dock | 0 | 0 | 0 | 0 | 0 | 0 | 0 | 0 | 0 | 0 | 75 | 0 | 0 | 0 | 0 | 0 | 0 | 0 | 28 | 0 | 0 | 0 | 0 |
| on head | 0 | 0 | 0 | 0 | 0 | 0 | 0 | 0 | 0 | 0 | 0 | 0 | 0 | 0 | 0 | 0 | 0 | 0 | 0 | 0 | 0 | 0 | 0 |
| around cloud | 0 | 0 | 0 | 0 | 0 | 0 | 0 | 0 | 0 | 0 | 0 | 0 | 0 | 0 | 0 | 0 | 0 | 0 | 0 | 0 | 0 | 0 | 0 |
| climbing | 0 | 0 | 0 | 0 | 0 | 0 | 0 | 0 | 0 | 0 | 0 | 0 | 0 | 0 | 0 | 0 | 0 | 0 | 0 | 0 | 0 | 0 | 0 |
| stone | 0 | 0 | 0 | 0 | 0 | 0 | 0 | 0 | 0 | 0 | 0 | 0 | 0 | 0 | 0 | 0 | 0 | 0 | 0 | 0 | 0 | 0 | 0 |
| on web | 0 | 0 | 0 | 0 | 0 | 0 | 0 | 0 | 0 | 0 | 0 | 0 | 0 | 0 | 0 | 0 | 0 | 0 | 0 | 0 | 0 | 0 | 0 |
| on a stick | 0 | 0 | 0 | 0 | 0 | 0 | 0 | 0 | 0 | 0 | 0 | 0 | 0 | 0 | 0 | 0 | 0 | 0 | 0 | 0 | 0 | 0 | 0 |
| leather | 0 | 0 | 0 | 0 | 0 | 0 | 0 | 0 | 0 | 0 | 0 | 0 | 0 | 0 | 0 | 0 | 0 | 0 | 0 | 0 | 0 | 0 | 0 |
| cutting | 0 | 0 | 0 | 0 | 0 | 0 | 0 | 0 | 0 | 0 | 0 | 0 | 0 | 0 | 0 | 0 | 0 | 0 | 0 | 0 | 0 | 0 | 0 |
| concrete | 0 | 0 | 0 | 0 | 0 | 0 | 0 | 0 | 0 | 0 | 0 | 0 | 0 | 0 | 0 | 0 | 0 | 0 | 0 | 0 | 0 | 0 | 0 |
| at park | 0 | 0 | 0 | 0 | 0 | 0 | 0 | 0 | 0 | 0 | 0 | 0 | 0 | 0 | 0 | 0 | 0 | 0 | 0 | 0 | 0 | 0 | 0 |
| on iceberg | 0 | 0 | 0 | 0 | 0 | 0 | 0 | 0 | 0 | 0 | 0 | 81 | 0 | 0 | 0 | 0 | 0 | 0 | 0 | 0 | 0 | 0 | 0 |
| on shoulder | 0 | 0 | 0 | 0 | 0 | 0 | 0 | 0 | 0 | 0 | 0 | 0 | 0 | 0 | 0 | 0 | 0 | 0 | 0 | 0 | 0 | 0 | 0 |
| at night | 0 | 0 | 0 | 0 | 0 | 0 | 0 | 0 | 0 | 0 | 0 | 0 | 0 | 0 | 0 | 0 | 0 | 0 | 0 | 0 | 0 | 0 | 0 |
| with flower | 0 | 0 | 0 | 0 | 0 | 0 | 0 | 0 | 0 | 0 | 0 | 0 | 0 | 0 | 0 | 0 | 0 | 0 | 0 | 76 | 0 | 0 | 0 |
| grazing | 0 | 0 | 0 | 0 | 0 | 0 | 0 | 0 | 0 | 0 | 0 | 0 | 0 | 0 | 0 | 0 | 0 | 0 | 0 | 0 | 0 | 0 | 0 |
| paper | 0 | 0 | 0 | 0 | 0 | 0 | 0 | 0 | 0 | 0 | 0 | 0 | 0 | 0 | 0 | 0 | 0 | 0 | 0 | 0 | 0 | 0 | 0 |
| spotted | 0 | 0 | 0 | 0 | 0 | 0 | 0 | 0 | 0 | 0 | 0 | 0 | 0 | 0 | 0 | 0 | 0 | 0 | 0 | 0 | 0 | 0 | 0 |
| at yard | 0 | 0 | 0 | 0 | 0 | 0 | 0 | 0 | 0 | 0 | 0 | 0 | 0 | 0 | 0 | 0 | 0 | 0 | 0 | 0 | 0 | 0 | 0 |
| with bee | 0 | 0 | 0 | 0 | 0 | 0 | 0 | 0 | 0 | 0 | 0 | 0 | 0 | 0 | 0 | 0 | 0 | 0 | 0 | 0 | 0 | 0 | 0 |
| hanging | 0 | 0 | 0 | 0 | 0 | 0 | 0 | 0 | 0 | 0 | 0 | 0 | 0 | 0 | 0 | 0 | 0 | 0 | 0 | 0 | 0 | 0 | 0 |
| clean | 0 | 0 | 0 | 0 | 0 | 0 | 0 | 0 | 0 | 0 | 0 | 0 | 0 | 0 | 0 | 0 | 0 | 0 | 0 | 0 | 0 | 0 | 0 |
| square | 0 | 0 | 0 | 0 | 0 | 0 | 0 | 0 | 0 | 0 | 0 | 0 | 0 | 0 | 0 | 0 | 0 | 0 | 0 | 0 | 0 | 0 | 0 |
| subway | 0 | 0 | 0 | 0 | 0 | 0 | 0 | 0 | 0 | 0 | 0 | 0 | 0 | 0 | 0 | 0 | 0 | 0 | 0 | 0 | 0 | 0 | 0 |
| jumping | 0 | 0 | 0 | 0 | 0 | 0 | 0 | 0 | 0 | 0 | 0 | 0 | 0 | 0 | 0 | 0 | 0 | 0 | 0 | 0 | 0 | 0 | 0 |
| on flower | 0 | 0 | 0 | 0 | 0 | 0 | 0 | 0 | 0 | 0 | 0 | 0 | 0 | 0 | 0 | 0 | 0 | 0 | 0 | 0 | 0 | 0 | 0 |
| landing | 0 | 0 | 0 | 0 | 0 | 0 | 0 | 0 | 67 | 0 | 0 | 0 | 0 | 0 | 0 | 0 | 0 | 0 | 0 | 0 | 0 | 0 | 0 |
| porcelain | 0 | 0 | 0 | 0 | 0 | 0 | 0 | 0 | 0 | 0 | 0 | 0 | 0 | 0 | 0 | 0 | 0 | 0 | 0 | 0 | 0 | 0 | 0 |
| at sunset | 0 | 0 | 0 | 0 | 0 | 0 | 0 | 0 | 0 | 0 | 0 | 0 | 0 | 0 | 0 | 0 | 0 | 0 | 0 | 0 | 0 | 0 | 0 |
| on desk | 0 | 0 | 0 | 0 | 0 | 0 | 0 | 0 | 0 | 0 | 0 | 0 | 0 | 0 | 0 | 0 | 0 | 0 | 0 | 65 | 0 | 0 | 0 |
| fighting | 0 | 0 | 0 | 0 | 0 | 0 | 0 | 0 | 0 | 0 | 0 | 0 | 0 | 0 | 0 | 64 | 0 | 0 | 0 | 0 | 0 | 0 | 0 |
| purple | 0 | 0 | 0 | 0 | 0 | 0 | 0 | 0 | 0 | 0 | 0 | 0 | 0 | 0 | 0 | 0 | 0 | 0 | 0 | 0 | 0 | 0 | 0 |
| off | 0 | 0 | 0 | 0 | 0 | 0 | 0 | 0 | 0 | 0 | 0 | 0 | 0 | 0 | 0 | 0 | 0 | 0 | 0 | 0 | 0 | 0 | 0 |
| gold | 0 | 0 | 0 | 0 | 0 | 0 | 0 | 0 | 0 | 0 | 0 | 0 | 0 | 0 | 0 | 0 | 0 | 0 | 0 | 0 | 0 | 0 | 23 |
| dirty | 0 | 0 | 0 | 0 | 0 | 0 | 0 | 0 | 0 | 0 | 0 | 0 | 0 | 0 | 0 | 0 | 0 | 0 | 0 | 0 | 0 | 0 | 0 |
| resting | 0 | 0 | 0 | 0 | 0 | 0 | 0 | 0 | 0 | 0 | 0 | 0 | 0 | 0 | 0 | 0 | 0 | 0 | 0 | 0 | 0 | 0 | 0 |
| beige | 0 | 0 | 0 | 0 | 0 | 0 | 0 | 0 | 0 | 0 | 0 | 0 | 0 | 0 | 0 | 0 | 0 | 0 | 0 | 0 | 0 | 0 | 0 |
| rectangular | 0 | 0 | 0 | 0 | 0 | 0 | 0 | 0 | 0 | 0 | 0 | 0 | 0 | 0 | 0 | 0 | 0 | 0 | 0 | 0 | 0 | 0 | 0 |
| thin | 0 | 0 | 0 | 0 | 0 | 0 | 0 | 0 | 0 | 0 | 0 | 0 | 0 | 0 | 0 | 0 | 0 | 0 | 0 | 0 | 0 | 0 | 0 |
| thick | 0 | 0 | 0 | 0 | 0 | 0 | 0 | 0 | 0 | 0 | 0 | 0 | 0 | 0 | 0 | 0 | 0 | 0 | 0 | 0 | 0 | 0 | 0 |
| in bucket | 0 | 0 | 0 | 0 | 0 | 0 | 0 | 0 | 0 | 0 | 0 | 0 | 0 | 0 | 0 | 0 | 0 | 0 | 0 | 0 | 0 | 0 | 0 |
| city | 0 | 0 | 0 | 0 | 0 | 0 | 0 | 0 | 53 | 0 | 0 | 0 | 0 | 0 | 0 | 0 | 0 | 0 | 0 | 0 | 0 | 0 | 0 |
| in pouch | 0 | 0 | 0 | 0 | 0 | 0 | 0 | 0 | 0 | 0 | 0 | 0 | 0 | 0 | 0 | 0 | 0 | 51 | 0 | 0 | 0 | 0 | 0 |
| in hole | 0 | 0 | 0 | 0 | 0 | 0 | 0 | 0 | 0 | 0 | 0 | 0 | 0 | 0 | 0 | 0 | 0 | 0 | 0 | 0 | 0 | 0 | 0 |
| on bird feeder | 0 | 45 | 0 | 0 | 0 | 0 | 0 | 0 | 0 | 0 | 0 | 0 | 0 | 0 | 0 | 0 | 0 | 0 | 0 | 0 | 0 | 0 | 0 |
| on shelves | 0 | 0 | 0 | 0 | 0 | 0 | 0 | 0 | 0 | 0 | 0 | 0 | 0 | 0 | 0 | 0 | 0 | 0 | 0 | 0 | 0 | 0 | 0 |
| on wall | 0 | 0 | 0 | 0 | 0 | 0 | 0 | 0 | 0 | 0 | 0 | 0 | 0 | 0 | 0 | 0 | 0 | 0 | 0 | 0 | 0 | 0 | 0 |
| on post | 0 | 0 | 0 | 0 | 0 | 0 | 0 | 0 | 0 | 0 | 0 | 0 | 0 | 0 | 0 | 0 | 0 | 0 | 0 | 0 | 0 | 0 | 0 |
| in box | 0 | 0 | 0 | 0 | 0 | 0 | 0 | 0 | 0 | 0 | 0 | 0 | 0 | 0 | 0 | 0 | 0 | 0 | 0 | 0 | 0 | 0 | 0 |
| shiny | 0 | 0 | 0 | 0 | 0 | 0 | 0 | 0 | 0 | 0 | 0 | 0 | 0 | 0 | 0 | 0 | 0 | 0 | 0 | 0 | 0 | 0 | 0 |
| sinking | 0 | 0 | 0 | 0 | 0 | 0 | 0 | 0 | 0 | 0 | 0 | 0 | 0 | 0 | 0 | 0 | 0 | 0 | 38 | 0 | 0 | 0 | 0 |
| with cargo | 0 | 0 | 0 | 0 | 0 | 0 | 0 | 0 | 0 | 0 | 0 | 0 | 0 | 0 | 0 | 0 | 0 | 0 | 38 | 0 | 0 | 0 | 0 |
| bright | 0 | 0 | 0 | 0 | 0 | 0 | 0 | 0 | 0 | 0 | 0 | 0 | 0 | 0 | 0 | 0 | 0 | 0 | 0 | 0 | 0 | 0 | 0 |
| in shell | 0 | 0 | 0 | 0 | 0 | 0 | 0 | 0 | 0 | 0 | 0 | 0 | 0 | 0 | 0 | 0 | 0 | 0 | 0 | 0 | 0 | 0 | 0 |
| aside traffic light | 0 | 0 | 0 | 0 | 0 | 0 | 0 | 0 | 0 | 0 | 0 | 0 | 0 | 0 | 0 | 0 | 0 | 0 | 0 | 0 | 0 | 0 | 0 |
| empty | 0 | 0 | 0 | 0 | 0 | 0 | 0 | 0 | 0 | 0 | 0 | 0 | 0 | 0 | 0 | 0 | 0 | 0 | 0 | 0 | 0 | 0 | 0 |
| cross tunnel | 0 | 0 | 0 | 0 | 0 | 0 | 0 | 0 | 0 | 0 | 0 | 0 | 0 | 0 | 0 | 0 | 0 | 0 | 0 | 0 | 0 | 0 | 0 |
| full | 0 | 0 | 0 | 0 | 0 | 0 | 0 | 0 | 0 | 0 | 0 | 0 | 0 | 0 | 0 | 0 | 0 | 0 | 0 | 0 | 0 | 0 | 0 |
| playing | 0 | 0 | 0 | 0 | 0 | 0 | 0 | 0 | 0 | 0 | 0 | 0 | 0 | 0 | 0 | 0 | 0 | 0 | 0 | 0 | 0 | 0 | 0 |
| light brown | 0 | 0 | 0 | 0 | 0 | 0 | 0 | 0 | 0 | 0 | 0 | 0 | 0 | 0 | 0 | 0 | 0 | 0 | 0 | 0 | 0 | 0 | 0 |
| staring | 0 | 0 | 0 | 0 | 0 | 0 | 0 | 0 | 0 | 0 | 0 | 0 | 0 | 0 | 0 | 0 | 0 | 0 | 0 | 0 | 0 | 0 | 0 |
| colorful | 0 | 0 | 0 | 0 | 0 | 0 | 0 | 0 | 0 | 0 | 0 | 0 | 0 | 0 | 0 | 0 | 0 | 0 | 0 | 0 | 0 | 0 | 0 |
| dark brown | 0 | 0 | 0 | 0 | 0 | 0 | 0 | 0 | 0 | 0 | 0 | 0 | 0 | 0 | 0 | 0 | 0 | 0 | 0 | 0 | 0 | 0 | 0 |
| young | 0 | 0 | 0 | 0 | 0 | 0 | 0 | 0 | 0 | 0 | 0 | 0 | 0 | 0 | 0 | 0 | 0 | 0 | 0 | 0 | 0 | 0 | 0 |
| dark blue | 0 | 0 | 0 | 0 | 0 | 0 | 0 | 0 | 0 | 0 | 0 | 0 | 0 | 0 | 0 | 0 | 0 | 0 | 0 | 0 | 0 | 0 | 0 |

| Contexts | | | | | | | | | | | | Concepts | | | | | | | | | | |
|---|---|---|---|---|---|---|---|---|---|---|---|---|---|---|---|---|---|---|---|---|---|---|
| | clock | spider | ostrich | tortoise | butterfly | pumpkin | sunflower | crocodile | bench | mailbox | lifeboat | dolphin | crab | window | pineapple | shorts | bag | toilet | carpet | cup | refrigerator | house | zebra |
| grass | 267 | 286 | 357 | 388 | 235 | 327 | 255 | 0 | 307 | 124 | 99 | 281 | 0 | 240 | 0 | 0 | 0 | 0 | 0 | 0 | 0 | 0 | 0 |
| water | 246 | 251 | 292 | 111 | 211 | 197 | 343 | 0 | 91 | 543 | 340 | 178 | 0 | 360 | 0 | 0 | 0 | 0 | 0 | 0 | 0 | 0 | 0 |
| outdoor | 248 | 326 | 291 | 248 | 240 | 209 | 326 | 0 | 229 | 374 | 309 | 126 | 0 | 152 | 0 | 0 | 0 | 0 | 0 | 0 | 0 | 0 | 0 |
| dim | 194 | 151 | 185 | 182 | 167 | 289 | 100 | 0 | 78 | 135 | 176 | 115 | 0 | 57 | 0 | 0 | 0 | 0 | 0 | 0 | 0 | 0 | 0 |
| white | 76 | 0 | 0 | 0 | 0 | 0 | 40 | 47 | 0 | 0 | 54 | 0 | 150 | 0 | 278 | 108 | 663 | 46 | 138 | 271 | 140 | 93 | 60 |
| rock | 98 | 113 | 198 | 255 | 118 | 137 | 151 | 0 | 234 | 123 | 93 | 183 | 0 | 53 | 0 | 0 | 0 | 0 | 0 | 0 | 0 | 0 | 0 |
| autumn | 54 | 36 | 171 | 132 | 289 | 67 | 38 | 0 | 89 | 4 | 7 | 15 | 0 | 17 | 0 | 0 | 0 | 0 | 0 | 0 | 0 | 0 | 0 |
| black | 0 | 0 | 0 | 0 | 0 | 0 | 0 | 80 | 99 | 0 | 0 | 0 | 34 | 0 | 244 | 259 | 0 | 0 | 0 | 43 | 0 | 102 | 21 |
| brown | 0 | 0 | 0 | 0 | 0 | 0 | 0 | 214 | 0 | 0 | 0 | 0 | 19 | 0 | 50 | 77 | 8 | 91 | 16 | 0 | 55 | 0 | 49 |
| wood | 0 | 0 | 0 | 0 | 0 | 0 | 0 | 601 | 36 | 0 | 0 | 0 | 33 | 0 | 0 | 0 | 0 | 0 | 0 | 0 | 0 | 0 | 0 |
| blue | 0 | 0 | 0 | 77 | 0 | 0 | 0 | 46 | 34 | 0 | 0 | 0 | 0 | 0 | 198 | 89 | 13 | 61 | 33 | 0 | 28 | 0 | 0 |
| green | 0 | 0 | 0 | 0 | 0 | 0 | 0 | 101 | 0 | 0 | 0 | 0 | 0 | 0 | 38 | 47 | 0 | 30 | 16 | 0 | 0 | 0 | 0 |
| gray | 0 | 0 | 0 | 0 | 0 | 0 | 0 | 63 | 0 | 0 | 0 | 0 | 0 | 0 | 86 | 27 | 8 | 94 | 0 | 11 | 27 | 0 | 25 |
| on snow | 0 | 80 | 0 | 0 | 0 | 0 | 0 | 0 | 0 | 0 | 0 | 0 | 0 | 0 | 0 | 0 | 0 | 0 | 0 | 0 | 0 | 0 | 0 |
| large | 0 | 0 | 0 | 0 | 0 | 0 | 0 | 33 | 0 | 0 | 0 | 0 | 275 | 0 | 0 | 42 | 0 | 0 | 0 | 33 | 60 | 22 | 67 |
| eating | 122 | 109 | 77 | 0 | 0 | 0 | 52 | 0 | 0 | 0 | 52 | 0 | 0 | 0 | 0 | 0 | 0 | 0 | 0 | 0 | 0 | 0 | 0 |
| red | 68 | 0 | 0 | 25 | 0 | 0 | 0 | 44 | 92 | 0 | 0 | 0 | 0 | 0 | 34 | 31 | 0 | 47 | 21 | 0 | 18 | 0 | 20 |
| metal | 0 | 0 | 0 | 0 | 0 | 0 | 0 | 141 | 92 | 0 | 0 | 0 | 0 | 0 | 0 | 0 | 0 | 0 | 0 | 0 | 0 | 0 | 42 |
| lying | 0 | 0 | 0 | 0 | 0 | 0 | 0 | 0 | 0 | 0 | 0 | 0 | 0 | 0 | 0 | 0 | 0 | 0 | 0 | 0 | 0 | 0 | 0 |
| with people | 0 | 40 | 36 | 0 | 0 | 118 | 27 | 0 | 75 | 52 | 21 | 0 | 0 | 0 | 0 | 0 | 0 | 0 | 0 | 0 | 0 | 0 | 0 |
| on beach | 0 | 0 | 28 | 0 | 0 | 0 | 54 | 0 | 0 | 0 | 78 | 0 | 0 | 0 | 0 | 0 | 0 | 0 | 0 | 0 | 0 | 0 | 0 |
| small | 0 | 0 | 0 | 0 | 0 | 0 | 0 | 17 | 0 | 0 | 0 | 0 | 53 | 0 | 0 | 26 | 0 | 0 | 29 | 34 | 27 | 14 | 12 |
| in forest | 0 | 0 | 0 | 0 | 0 | 0 | 0 | 0 | 0 | 0 | 0 | 0 | 0 | 0 | 0 | 0 | 0 | 0 | 0 | 0 | 0 | 0 | 0 |
| on road | 0 | 0 | 0 | 0 | 0 | 0 | 0 | 0 | 0 | 0 | 0 | 0 | 0 | 0 | 0 | 0 | 0 | 0 | 0 | 0 | 0 | 0 | 0 |
| silver | 0 | 0 | 0 | 0 | 0 | 0 | 0 | 0 | 0 | 0 | 0 | 0 | 0 | 0 | 0 | 0 | 0 | 0 | 0 | 39 | 0 | 0 | 0 |
| running | 0 | 86 | 0 | 0 | 0 | 0 | 0 | 0 | 0 | 0 | 0 | 0 | 0 | 0 | 0 | 0 | 0 | 0 | 0 | 0 | 0 | 0 | 0 |
| in cage | 0 | 118 | 0 | 0 | 0 | 0 | 0 | 0 | 0 | 0 | 0 | 0 | 0 | 0 | 0 | 0 | 0 | 0 | 0 | 0 | 0 | 0 | 0 |
| open | 0 | 0 | 0 | 0 | 0 | 0 | 0 | 0 | 0 | 0 | 0 | 0 | 127 | 0 | 0 | 0 | 34 | 0 | 0 | 0 | 0 | 0 | 0 |
| yellow | 131 | 0 | 0 | 49 | 0 | 0 | 0 | 0 | 0 | 0 | 0 | 0 | 0 | 0 | 0 | 0 | 0 | 0 | 0 | 0 | 24 | 0 | 0 |
| standing | 0 | 0 | 0 | 0 | 0 | 0 | 0 | 0 | 0 | 0 | 0 | 0 | 0 | 0 | 0 | 0 | 0 | 0 | 0 | 0 | 0 | 103 | 0 |
| in city | 0 | 0 | 0 | 0 | 0 | 0 | 0 | 0 | 0 | 0 | 0 | 0 | 0 | 0 | 0 | 0 | 0 | 0 | 0 | 0 | 0 | 0 | 0 |
| sitting | 0 | 107 | 0 | 0 | 0 | 0 | 0 | 0 | 0 | 0 | 0 | 0 | 0 | 0 | 0 | 0 | 0 | 0 | 0 | 0 | 0 | 0 | 0 |
| tall | 0 | 0 | 0 | 0 | 0 | 0 | 0 | 0 | 0 | 0 | 0 | 0 | 0 | 0 | 0 | 0 | 0 | 0 | 0 | 0 | 0 | 0 | 0 |
| in river | 0 | 0 | 0 | 0 | 0 | 0 | 0 | 0 | 0 | 0 | 0 | 0 | 0 | 0 | 0 | 0 | 0 | 0 | 0 | 0 | 0 | 0 | 0 |
| in water | 0 | 0 | 0 | 0 | 0 | 0 | 0 | 0 | 0 | 0 | 0 | 0 | 0 | 0 | 0 | 0 | 0 | 0 | 0 | 0 | 0 | 0 | 0 |
| dark | 0 | 0 | 0 | 0 | 0 | 0 | 0 | 17 | 0 | 0 | 0 | 0 | 22 | 0 | 31 | 0 | 0 | 0 | 0 | 0 | 0 | 0 | 0 |
| at home | 0 | 0 | 0 | 0 | 0 | 0 | 0 | 0 | 0 | 0 | 0 | 0 | 0 | 0 | 0 | 0 | 0 | 0 | 0 | 0 | 0 | 0 | 0 |
| walking | 0 | 0 | 0 | 0 | 0 | 0 | 0 | 0 | 0 | 0 | 0 | 0 | 0 | 0 | 0 | 0 | 0 | 0 | 0 | 0 | 0 | 32 | 0 |
| glass | 0 | 0 | 0 | 0 | 0 | 0 | 0 | 0 | 0 | 0 | 0 | 0 | 207 | 0 | 0 | 0 | 0 | 0 | 37 | 0 | 0 | 0 | 0 |
| orange | 0 | 0 | 0 | 0 | 0 | 0 | 0 | 0 | 0 | 0 | 0 | 0 | 0 | 0 | 0 | 23 | 0 | 0 | 0 | 0 | 0 | 0 | 0 |
| on tree | 0 | 0 | 0 | 0 | 0 | 0 | 0 | 0 | 0 | 0 | 0 | 0 | 28 | 0 | 0 | 0 | 0 | 0 | 0 | 0 | 0 | 0 | 0 |
| in hand | 132 | 0 | 0 | 85 | 0 | 0 | 0 | 0 | 0 | 0 | 0 | 0 | 0 | 0 | 0 | 0 | 0 | 0 | 0 | 0 | 0 | 0 | 0 |
| flying | 0 | 0 | 0 | 0 | 0 | 0 | 0 | 0 | 0 | 0 | 0 | 0 | 0 | 0 | 0 | 0 | 0 | 0 | 0 | 0 | 0 | 0 | 0 |
| plastic | 0 | 0 | 0 | 0 | 0 | 0 | 0 | 0 | 0 | 0 | 0 | 0 | 0 | 0 | 0 | 166 | 0 | 0 | 78 | 0 | 0 | 0 | 0 |
| long | 0 | 0 | 0 | 0 | 0 | 0 | 0 | 0 | 0 | 0 | 0 | 0 | 0 | 0 | 0 | 0 | 0 | 0 | 0 | 0 | 0 | 0 | 0 |
| closed | 0 | 0 | 0 | 0 | 0 | 0 | 0 | 0 | 0 | 0 | 0 | 0 | 125 | 0 | 0 | 0 | 38 | 0 | 0 | 0 | 0 | 0 | 0 |
| on ground | 0 | 0 | 0 | 0 | 0 | 0 | 0 | 0 | 0 | 0 | 0 | 0 | 0 | 0 | 0 | 0 | 0 | 0 | 0 | 0 | 0 | 0 | 0 |
| aside mountain | 0 | 0 | 0 | 0 | 0 | 0 | 0 | 0 | 0 | 0 | 0 | 0 | 0 | 0 | 0 | 0 | 0 | 0 | 0 | 0 | 0 | 0 | 0 |
| pink | 0 | 0 | 0 | 0 | 0 | 0 | 0 | 0 | 0 | 0 | 0 | 0 | 0 | 0 | 0 | 25 | 6 | 0 | 0 | 0 | 0 | 0 | 0 |
| round | 0 | 0 | 0 | 0 | 0 | 0 | 0 | 0 | 0 | 0 | 0 | 0 | 14 | 0 | 0 | 0 | 0 | 0 | 0 | 0 | 0 | 0 | 0 |
| bare | 0 | 0 | 0 | 0 | 0 | 0 | 0 | 0 | 0 | 0 | 0 | 0 | 0 | 0 | 0 | 0 | 0 | 0 | 0 | 0 | 0 | 0 | 0 |
| tan | 0 | 0 | 0 | 0 | 0 | 0 | 0 | 0 | 0 | 0 | 0 | 0 | 0 | 0 | 33 | 0 | 0 | 27 | 0 | 0 | 0 | 0 | 0 |
| baby | 0 | 0 | 0 | 0 | 0 | 0 | 0 | 0 | 0 | 0 | 0 | 0 | 0 | 0 | 0 | 0 | 0 | 0 | 0 | 0 | 0 | 0 | 0 |
| in garage | 0 | 0 | 0 | 0 | 0 | 0 | 0 | 0 | 0 | 0 | 0 | 0 | 0 | 0 | 0 | 0 | 0 | 0 | 0 | 0 | 0 | 0 | 0 |
| yacht | 0 | 0 | 0 | 0 | 0 | 0 | 0 | 0 | 0 | 0 | 0 | 0 | 0 | 0 | 0 | 0 | 0 | 0 | 0 | 0 | 0 | 0 | 0 |
| on track | 0 | 0 | 0 | 0 | 0 | 0 | 0 | 0 | 0 | 0 | 0 | 0 | 0 | 0 | 0 | 0 | 0 | 0 | 0 | 0 | 0 | 0 | 0 |
| at station | 0 | 0 | 0 | 0 | 0 | 0 | 0 | 0 | 0 | 0 | 0 | 0 | 0 | 0 | 0 | 0 | 0 | 0 | 0 | 0 | 0 | 0 | 0 |
| wooden | 0 | 0 | 0 | 0 | 0 | 0 | 0 | 0 | 0 | 0 | 0 | 0 | 0 | 0 | 0 | 0 | 0 | 0 | 0 | 0 | 0 | 0 | 0 |
| in farm | 0 | 0 | 0 | 0 | 104 | 72 | 0 | 0 | 0 | 0 | 0 | 0 | 0 | 68 | 0 | 0 | 0 | 0 | 0 | 0 | 0 | 0 | 0 |
| on branch | 0 | 0 | 0 | 0 | 0 | 0 | 0 | 0 | 0 | 0 | 0 | 0 | 0 | 0 | 0 | 0 | 0 | 0 | 0 | 0 | 0 | 0 | 0 |
| leafy | 0 | 0 | 0 | 0 | 0 | 0 | 0 | 0 | 0 | 0 | 0 | 0 | 0 | 0 | 0 | 0 | 0 | 0 | 0 | 0 | 0 | 0 | 0 |
| shared | 0 | 0 | 0 | 0 | 0 | 0 | 0 | 0 | 0 | 0 | 0 | 0 | 0 | 0 | 0 | 0 | 0 | 0 | 0 | 0 | 0 | 0 | 0 |
| velodrome | 0 | 0 | 0 | 0 | 0 | 0 | 0 | 0 | 0 | 0 | 0 | 0 | 0 | 0 | 0 | 0 | 0 | 0 | 0 | 0 | 0 | 0 | 0 |
| at wharf | 0 | 0 | 0 | 0 | 0 | 0 | 0 | 0 | 0 | 0 | 0 | 0 | 0 | 0 | 0 | 0 | 0 | 0 | 0 | 0 | 0 | 0 | 0 |
| double decker | 0 | 0 | 0 | 0 | 0 | 0 | 0 | 0 | 0 | 0 | 0 | 0 | 0 | 0 | 0 | 0 | 0 | 0 | 0 | 0 | 0 | 0 | 0 |
| on | 0 | 0 | 0 | 0 | 0 | 0 | 0 | 0 | 0 | 0 | 0 | 0 | 0 | 0 | 0 | 0 | 0 | 0 | 0 | 0 | 0 | 0 | 0 |
| cross bridge | 0 | 0 | 0 | 0 | 0 | 0 | 0 | 0 | 0 | 0 | 0 | 0 | 0 | 0 | 0 | 0 | 0 | 0 | 0 | 0 | 0 | 0 | 0 |
| in sunset | 0 | 0 | 0 | 0 | 0 | 0 | 0 | 0 | 0 | 0 | 0 | 0 | 0 | 0 | 0 | 0 | 0 | 0 | 0 | 0 | 0 | 0 | 0 |
| at heliport | 0 | 0 | 0 | 0 | 0 | 0 | 0 | 0 | 0 | 0 | 0 | 0 | 0 | 0 | 0 | 0 | 0 | 0 | 0 | 0 | 0 | 0 | 0 |
| on bridge | 0 | 0 | 0 | 0 | 0 | 0 | 0 | 0 | 0 | 0 | 0 | 0 | 0 | 0 | 0 | 0 | 0 | 0 | 0 | 0 | 0 | 0 | 0 |
| in burrow | 0 | 0 | 0 | 0 | 0 | 0 | 0 | 0 | 0 | 0 | 0 | 0 | 76 | 0 | 0 | 0 | 0 | 0 | 0 | 0 | 0 | 0 | 0 |
| aside tree | 0 | 0 | 0 | 0 | 0 | 0 | 0 | 0 | 0 | 0 | 0 | 0 | 0 | 0 | 0 | 0 | 0 | 0 | 0 | 0 | 0 | 0 | 0 |
| in zoo | 0 | 0 | 0 | 0 | 0 | 0 | 0 | 0 | 0 | 0 | 0 | 0 | 0 | 0 | 0 | 0 | 0 | 0 | 0 | 0 | 0 | 0 | 0 |
| in pot | 0 | 0 | 0 | 0 | 0 | 59 | 0 | 0 | 0 | 0 | 0 | 0 | 0 | 0 | 0 | 0 | 0 | 0 | 0 | 0 | 0 | 0 | 0 |
| at airport | 0 | 0 | 0 | 0 | 0 | 0 | 0 | 0 | 0 | 0 | 126 | 0 | 0 | 0 | 0 | 0 | 0 | 0 | 0 | 0 | 0 | 0 | 0 |
| on sea | 0 | 0 | 0 | 0 | 0 | 0 | 0 | 0 | 0 | 0 | 0 | 0 | 0 | 0 | 0 | 0 | 0 | 0 | 0 | 0 | 0 | 0 | 0 |
| sleeping | 0 | 0 | 0 | 0 | 0 | 0 | 0 | 0 | 0 | 0 | 0 | 0 | 0 | 0 | 0 | 0 | 0 | 0 | 0 | 0 | 0 | 0 | 0 |
| in desert | 0 | 0 | 0 | 0 | 0 | 0 | 0 | 0 | 0 | 0 | 0 | 0 | 0 | 0 | 0 | 0 | 0 | 0 | 0 | 0 | 0 | 0 | 0 |
| eating grass | 0 | 0 | 0 | 0 | 0 | 0 | 0 | 0 | 0 | 0 | 0 | 0 | 0 | 0 | 0 | 0 | 0 | 0 | 0 | 0 | 0 | 0 | 0 |
| in race | 0 | 0 | 0 | 0 | 0 | 0 | 0 | 0 | 0 | 0 | 0 | 0 | 0 | 0 | 0 | 0 | 0 | 0 | 0 | 0 | 0 | 0 | 0 |
| taking off | 0 | 0 | 0 | 0 | 0 | 0 | 0 | 0 | 0 | 0 | 0 | 0 | 0 | 0 | 0 | 0 | 0 | 0 | 0 | 0 | 0 | 0 | 0 |
| on power line | 0 | 0 | 0 | 0 | 0 | 0 | 0 | 0 | 0 | 0 | 0 | 0 | 0 | 0 | 0 | 0 | 0 | 0 | 0 | 0 | 0 | 0 | 0 |
| in circus | 0 | 0 | 0 | 0 | 0 | 0 | 0 | 0 | 0 | 0 | 0 | 0 | 0 | 0 | 0 | 0 | 0 | 0 | 0 | 0 | 0 | 0 | 0 |
| mountain | 0 | 0 | 0 | 0 | 0 | 0 | 0 | 0 | 0 | 0 | 0 | 0 | 0 | 0 | 0 | 0 | 0 | 0 | 0 | 0 | 0 | 0 | 0 |
| short | 0 | 0 | 0 | 0 | 0 | 0 | 0 | 0 | 0 | 0 | 0 | 0 | 0 | 0 | 0 | 0 | 0 | 0 | 0 | 0 | 0 | 0 | 0 |

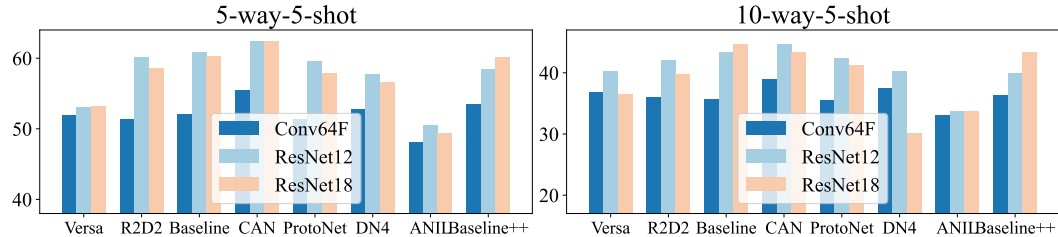

Figure 6: Experiments of different backbone architectures under 5-way and 10-way 5- shot settings.

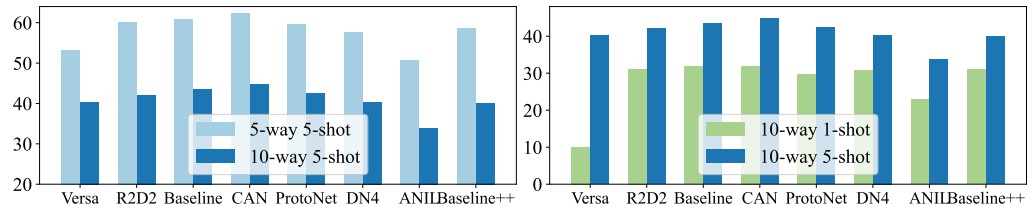

Figure 7: Experimental results of different ways (left) and shots (right) on testing performance.

| Contexts | clock | spider | ostrich | tortoise | butterfly | pumpkin | sunflower | crocodile | bench | mailbox | lifeboat | dolphin | crab | window | pineapple | shorts | bag | toilet | carpet | cup | refrigerator | house | zebra |
|---|---|---|---|---|---|---|---|---|---|---|---|---|---|---|---|---|---|---|---|---|---|---|---|
| open mouth | 0 | 0 | 0 | 0 | 0 | 0 | 113 | 0 | 0 | 0 | 0 | 0 | 0 | 0 | 0 | 0 | 0 | 0 | 0 | 0 | 0 | 0 | 0 |
| on booth | 0 | 0 | 0 | 0 | 0 | 0 | 0 | 0 | 0 | 0 | 0 | 0 | 0 | 0 | 0 | 0 | 0 | 0 | 0 | 0 | 0 | 0 | 0 |
| howling | 0 | 0 | 0 | 0 | 0 | 0 | 0 | 0 | 0 | 0 | 0 | 0 | 0 | 0 | 0 | 0 | 0 | 0 | 0 | 0 | 0 | 0 | 0 |
| brick | 0 | 0 | 0 | 0 | 0 | 0 | 0 | 0 | 0 | 0 | 0 | 0 | 0 | 0 | 0 | 0 | 0 | 0 | 0 | 0 | 51 | 0 | 57 |
| khaki | 0 | 0 | 0 | 0 | 0 | 0 | 0 | 0 | 0 | 0 | 0 | 0 | 0 | 0 | 34 | 0 | 0 | 0 | 0 | 0 | 0 | 0 | 0 |
| at dock | 0 | 0 | 0 | 0 | 0 | 0 | 0 | 0 | 0 | 0 | 0 | 0 | 0 | 0 | 0 | 0 | 0 | 0 | 0 | 0 | 0 | 0 | 0 |
| on head | 0 | 0 | 0 | 0 | 95 | 0 | 0 | 0 | 0 | 0 | 0 | 0 | 0 | 0 | 0 | 0 | 0 | 0 | 0 | 0 | 0 | 0 | 0 |
| around cloud | 0 | 0 | 0 | 0 | 0 | 0 | 0 | 0 | 0 | 0 | 0 | 0 | 0 | 0 | 0 | 0 | 0 | 0 | 0 | 0 | 0 | 0 | 0 |
| climbing | 0 | 0 | 0 | 0 | 0 | 0 | 0 | 0 | 0 | 0 | 0 | 0 | 0 | 0 | 0 | 0 | 0 | 0 | 0 | 0 | 0 | 0 | 0 |
| stone | 0 | 0 | 0 | 0 | 0 | 0 | 0 | 34 | 0 | 0 | 0 | 0 | 0 | 0 | 0 | 0 | 0 | 0 | 0 | 0 | 0 | 0 | 28 |
| on web | 87 | 0 | 0 | 0 | 0 | 0 | 0 | 0 | 0 | 0 | 0 | 0 | 0 | 0 | 0 | 0 | 0 | 0 | 0 | 0 | 0 | 0 | 0 |
| on a stick | 0 | 0 | 0 | 0 | 85 | 0 | 0 | 0 | 0 | 0 | 0 | 0 | 0 | 0 | 0 | 0 | 0 | 0 | 0 | 0 | 0 | 0 | 0 |
| leather | 0 | 0 | 0 | 0 | 0 | 0 | 0 | 0 | 0 | 0 | 0 | 0 | 0 | 0 | 0 | 27 | 0 | 0 | 0 | 0 | 0 | 0 | 0 |
| cutting | 0 | 0 | 0 | 0 | 0 | 0 | 0 | 0 | 0 | 0 | 0 | 0 | 0 | 84 | 0 | 0 | 0 | 0 | 0 | 0 | 0 | 0 | 0 |
| concrete | 0 | 0 | 0 | 0 | 0 | 0 | 0 | 53 | 0 | 0 | 0 | 0 | 0 | 0 | 0 | 0 | 0 | 0 | 0 | 0 | 0 | 0 | 0 |
| at park | 0 | 0 | 0 | 0 | 0 | 0 | 0 | 0 | 0 | 0 | 0 | 0 | 0 | 0 | 0 | 0 | 0 | 0 | 0 | 0 | 0 | 0 | 0 |
| on iceberg | 0 | 0 | 0 | 0 | 0 | 0 | 0 | 0 | 0 | 0 | 0 | 0 | 0 | 0 | 0 | 0 | 0 | 0 | 0 | 0 | 0 | 0 | 0 |
| on shoulder | 0 | 0 | 0 | 0 | 0 | 0 | 0 | 0 | 0 | 0 | 0 | 0 | 0 | 0 | 0 | 0 | 0 | 0 | 0 | 0 | 0 | 0 | 0 |
| at night | 0 | 0 | 0 | 0 | 0 | 0 | 0 | 0 | 0 | 0 | 0 | 0 | 0 | 0 | 0 | 0 | 0 | 0 | 0 | 0 | 0 | 0 | 0 |
| with flower | 0 | 0 | 0 | 0 | 0 | 0 | 0 | 0 | 0 | 0 | 0 | 0 | 0 | 0 | 0 | 0 | 0 | 0 | 0 | 0 | 0 | 0 | 0 |
| grazing | 0 | 0 | 0 | 0 | 0 | 0 | 0 | 0 | 0 | 0 | 0 | 0 | 0 | 0 | 0 | 0 | 0 | 0 | 0 | 0 | 0 | 54 | 0 |
| paper | 0 | 0 | 0 | 0 | 0 | 0 | 0 | 0 | 0 | 0 | 0 | 0 | 0 | 0 | 0 | 46 | 0 | 30 | 0 | 0 | 0 | 0 | 0 |
| spotted | 0 | 0 | 0 | 0 | 0 | 0 | 0 | 0 | 0 | 0 | 0 | 0 | 0 | 0 | 0 | 0 | 0 | 0 | 0 | 0 | 0 | 0 | 0 |
| at yard | 0 | 0 | 0 | 0 | 0 | 0 | 0 | 0 | 0 | 0 | 0 | 0 | 0 | 0 | 0 | 0 | 0 | 0 | 0 | 0 | 0 | 0 | 0 |
| with bee | 0 | 0 | 0 | 0 | 0 | 73 | 0 | 0 | 0 | 0 | 0 | 0 | 0 | 0 | 0 | 0 | 0 | 0 | 0 | 0 | 0 | 0 | 0 |
| hanging | 0 | 0 | 0 | 0 | 0 | 0 | 0 | 0 | 0 | 0 | 73 | 0 | 0 | 0 | 0 | 0 | 0 | 0 | 0 | 0 | 0 | 0 | 0 |
| clean | 0 | 0 | 0 | 0 | 0 | 0 | 0 | 0 | 0 | 0 | 0 | 0 | 0 | 0 | 0 | 0 | 46 | 0 | 0 | 0 | 0 | 0 | 0 |
| square | 0 | 0 | 0 | 0 | 0 | 0 | 0 | 0 | 0 | 0 | 0 | 0 | 19 | 0 | 0 | 0 | 0 | 0 | 0 | 0 | 0 | 0 | 0 |
| subway | 0 | 0 | 0 | 0 | 0 | 0 | 0 | 0 | 0 | 0 | 0 | 0 | 0 | 0 | 0 | 0 | 0 | 0 | 0 | 0 | 0 | 0 | 0 |
| jumping | 0 | 0 | 0 | 0 | 0 | 0 | 0 | 0 | 0 | 0 | 0 | 0 | 0 | 0 | 0 | 0 | 0 | 0 | 0 | 0 | 0 | 0 | 0 |
| on flower | 0 | 0 | 0 | 67 | 0 | 0 | 0 | 0 | 0 | 0 | 0 | 0 | 0 | 0 | 0 | 0 | 0 | 0 | 0 | 0 | 0 | 0 | 0 |
| landing | 0 | 0 | 0 | 0 | 0 | 0 | 0 | 0 | 0 | 0 | 0 | 0 | 0 | 0 | 0 | 0 | 0 | 0 | 0 | 0 | 0 | 0 | 0 |
| porcelain | 0 | 0 | 0 | 0 | 0 | 0 | 0 | 0 | 0 | 0 | 0 | 0 | 0 | 0 | 0 | 0 | 0 | 0 | 0 | 0 | 0 | 0 | 0 |
| at sunset | 0 | 0 | 0 | 0 | 0 | 0 | 0 | 0 | 0 | 0 | 0 | 0 | 0 | 0 | 0 | 0 | 0 | 0 | 0 | 0 | 0 | 0 | 0 |
| on desk | 0 | 0 | 0 | 0 | 0 | 0 | 0 | 0 | 0 | 0 | 0 | 0 | 0 | 0 | 0 | 0 | 0 | 0 | 0 | 0 | 0 | 0 | 0 |
| fighting | 0 | 0 | 0 | 0 | 0 | 0 | 0 | 0 | 0 | 0 | 0 | 0 | 0 | 0 | 0 | 0 | 0 | 0 | 0 | 0 | 0 | 0 | 0 |
| purple | 0 | 0 | 0 | 0 | 0 | 0 | 0 | 0 | 0 | 0 | 0 | 0 | 0 | 0 | 0 | 0 | 0 | 0 | 0 | 0 | 0 | 0 | 0 |
| off | 0 | 0 | 0 | 0 | 0 | 0 | 0 | 0 | 0 | 0 | 0 | 0 | 0 | 0 | 0 | 0 | 0 | 0 | 0 | 0 | 0 | 0 | 0 |
| gold | 0 | 0 | 0 | 0 | 0 | 0 | 0 | 0 | 0 | 0 | 0 | 0 | 0 | 0 | 0 | 0 | 0 | 0 | 0 | 0 | 0 | 0 | 0 |
| dirty | 0 | 0 | 0 | 0 | 0 | 0 | 0 | 0 | 0 | 0 | 0 | 0 | 0 | 0 | 0 | 0 | 48 | 0 | 0 | 0 | 0 | 0 | 0 |
| resting | 0 | 0 | 0 | 0 | 0 | 0 | 0 | 0 | 0 | 0 | 0 | 0 | 0 | 0 | 0 | 0 | 0 | 0 | 0 | 0 | 0 | 0 | 0 |
| beige | 0 | 0 | 0 | 0 | 0 | 0 | 0 | 0 | 0 | 0 | 0 | 0 | 0 | 0 | 0 | 0 | 0 | 41 | 0 | 0 | 0 | 0 | 0 |
| rectangular | 0 | 0 | 0 | 0 | 0 | 0 | 0 | 0 | 0 | 0 | 0 | 0 | 25 | 0 | 0 | 0 | 0 | 0 | 0 | 0 | 0 | 0 | 0 |
| thin | 0 | 0 | 0 | 0 | 0 | 0 | 0 | 0 | 0 | 0 | 0 | 0 | 0 | 0 | 0 | 0 | 0 | 0 | 0 | 0 | 0 | 0 | 0 |
| thick | 0 | 0 | 0 | 0 | 0 | 0 | 0 | 0 | 0 | 0 | 0 | 0 | 0 | 0 | 0 | 0 | 0 | 0 | 0 | 0 | 0 | 0 | 0 |
| in bucket | 0 | 0 | 0 | 0 | 0 | 0 | 0 | 0 | 0 | 0 | 0 | 54 | 0 | 0 | 0 | 0 | 0 | 0 | 0 | 0 | 0 | 0 | 0 |
| city | 0 | 0 | 0 | 0 | 0 | 0 | 0 | 0 | 0 | 0 | 0 | 0 | 0 | 0 | 0 | 0 | 0 | 0 | 0 | 0 | 0 | 0 | 0 |
| in pouch | 0 | 0 | 0 | 0 | 0 | 0 | 0 | 0 | 0 | 0 | 0 | 0 | 0 | 0 | 0 | 0 | 0 | 0 | 0 | 0 | 0 | 0 | 0 |
| in hole | 0 | 0 | 0 | 0 | 0 | 0 | 0 | 0 | 0 | 0 | 0 | 0 | 0 | 0 | 0 | 0 | 0 | 0 | 0 | 0 | 0 | 0 | 0 |
| on bird feeder | 0 | 0 | 0 | 0 | 0 | 0 | 0 | 0 | 0 | 0 | 0 | 0 | 0 | 0 | 0 | 0 | 0 | 0 | 0 | 0 | 0 | 0 | 0 |
| on shelves | 0 | 0 | 0 | 0 | 44 | 0 | 0 | 0 | 0 | 0 | 0 | 0 | 0 | 0 | 0 | 0 | 0 | 0 | 0 | 0 | 0 | 0 | 0 |
| on wall | 0 | 0 | 0 | 0 | 0 | 0 | 0 | 0 | 43 | 0 | 0 | 0 | 0 | 0 | 0 | 0 | 0 | 0 | 0 | 0 | 0 | 0 | 0 |
| on post | 0 | 0 | 0 | 0 | 0 | 0 | 0 | 0 | 42 | 0 | 0 | 0 | 0 | 0 | 0 | 0 | 0 | 0 | 0 | 0 | 0 | 0 | 0 |
| in box | 0 | 0 | 0 | 0 | 0 | 0 | 0 | 0 | 0 | 0 | 0 | 0 | 0 | 0 | 41 | 0 | 0 | 0 | 0 | 0 | 0 | 0 | 0 |
| shiny | 0 | 0 | 0 | 0 | 0 | 0 | 0 | 0 | 0 | 0 | 0 | 0 | 0 | 0 | 0 | 0 | 0 | 0 | 0 | 0 | 0 | 0 | 0 |
| sinking | 0 | 0 | 0 | 0 | 0 | 0 | 0 | 0 | 0 | 0 | 0 | 0 | 0 | 0 | 0 | 0 | 0 | 0 | 0 | 0 | 0 | 0 | 0 |
| with cargo | 0 | 0 | 0 | 0 | 0 | 0 | 0 | 0 | 0 | 0 | 0 | 0 | 0 | 0 | 0 | 0 | 0 | 0 | 0 | 0 | 0 | 0 | 0 |
| bright | 0 | 0 | 0 | 0 | 0 | 0 | 0 | 0 | 0 | 0 | 0 | 0 | 38 | 0 | 0 | 0 | 0 | 0 | 0 | 0 | 0 | 0 | 0 |
| in shell | 0 | 0 | 37 | 0 | 0 | 0 | 0 | 0 | 0 | 0 | 0 | 0 | 0 | 0 | 0 | 0 | 0 | 0 | 0 | 0 | 0 | 0 | 0 |
| aside traffic light | 0 | 0 | 0 | 0 | 0 | 0 | 0 | 0 | 0 | 0 | 0 | 0 | 0 | 0 | 0 | 0 | 0 | 0 | 0 | 0 | 0 | 0 | 0 |
| empty | 0 | 0 | 0 | 0 | 0 | 0 | 0 | 0 | 0 | 0 | 0 | 0 | 0 | 0 | 0 | 0 | 0 | 0 | 0 | 20 | 0 | 0 | 0 |
| cross tunnel | 0 | 0 | 0 | 0 | 0 | 0 | 0 | 0 | 0 | 0 | 0 | 0 | 0 | 0 | 0 | 0 | 0 | 0 | 0 | 0 | 0 | 0 | 0 |
| full | 0 | 0 | 0 | 0 | 0 | 0 | 0 | 0 | 0 | 0 | 0 | 0 | 0 | 0 | 0 | 0 | 0 | 0 | 0 | 16 | 0 | 0 | 0 |
| playing | 0 | 0 | 0 | 0 | 0 | 0 | 0 | 0 | 0 | 0 | 0 | 0 | 0 | 0 | 0 | 0 | 0 | 0 | 0 | 0 | 0 | 0 | 0 |
| light brown | 0 | 0 | 0 | 0 | 0 | 0 | 0 | 0 | 0 | 0 | 0 | 0 | 0 | 0 | 0 | 0 | 0 | 0 | 0 | 0 | 0 | 0 | 0 |
| staring | 0 | 0 | 0 | 0 | 0 | 0 | 0 | 0 | 0 | 0 | 0 | 0 | 0 | 0 | 0 | 0 | 0 | 0 | 0 | 0 | 0 | 0 | 0 |
| colorful | 0 | 0 | 0 | 0 | 0 | 0 | 0 | 0 | 0 | 0 | 0 | 0 | 0 | 0 | 0 | 0 | 0 | 0 | 0 | 0 | 0 | 0 | 0 |
| dark brown | 0 | 0 | 0 | 0 | 0 | 0 | 0 | 0 | 0 | 0 | 0 | 0 | 0 | 0 | 0 | 0 | 0 | 0 | 0 | 0 | 0 | 0 | 0 |
| young | 0 | 0 | 0 | 0 | 0 | 0 | 0 | 0 | 0 | 0 | 0 | 0 | 0 | 0 | 0 | 0 | 0 | 0 | 0 | 0 | 0 | 0 | 0 |
| dark blue | 0 | 0 | 0 | 0 | 0 | 0 | 0 | 0 | 0 | 0 | 0 | 0 | 0 | 0 | 0 | 0 | 0 | 0 | 0 | 0 | 0 | 0 | 0 |

**Concepts**

| Contexts | tower | ocean | spoon | suit | fire hydrant | skateboard | pillow | bed | knife | backpack | bridge | rat | laptop | sink | frame | bowl | coat | bush | cloud | cabinet | shrimp | dress | television |
|---|---|---|---|---|---|---|---|---|---|---|---|---|---|---|---|---|---|---|---|---|---|---|---|
| grass | 0 | 0 | 0 | 0 | 0 | 0 | 0 | 0 | 0 | 0 | 125 | 0 | 0 | 0 | 0 | 0 | 0 | 190 | 0 | 0 | 0 | 0 | 0 |
| water | 0 | 0 | 0 | 0 | 0 | 0 | 0 | 0 | 0 | 0 | 0 | 0 | 0 | 0 | 0 | 0 | 0 | 260 | 0 | 0 | 0 | 0 | 0 |
| outdoor | 0 | 0 | 0 | 0 | 0 | 0 | 0 | 0 | 0 | 0 | 0 | 0 | 0 | 0 | 0 | 0 | 0 | 44 | 0 | 0 | 0 | 0 | 0 |
| dim | 0 | 0 | 0 | 0 | 0 | 0 | 0 | 0 | 0 | 0 | 0 | 0 | 0 | 0 | 0 | 0 | 0 | 7 | 0 | 0 | 0 | 0 | 0 |
| white | 22 | 25 | 27 | 49 | 35 | 161 | 125 | 11 | 0 | 0 | 0 | 64 | 473 | 167 | 288 | 68 | 0 | 421 | 151 | 0 | 159 | 9 | 186 |
| rock | 0 | 0 | 0 | 0 | 0 | 0 | 0 | 0 | 0 | 0 | 0 | 0 | 0 | 0 | 0 | 0 | 0 | 47 | 0 | 0 | 0 | 0 | 0 |
| autumn | 0 | 0 | 0 | 0 | 0 | 0 | 0 | 0 | 0 | 0 | 0 | 0 | 0 | 0 | 0 | 0 | 0 | 4 | 0 | 0 | 0 | 0 | 0 |
| black | 0 | 10 | 191 | 22 | 188 | 14 | 9 | 33 | 162 | 0 | 0 | 208 | 0 | 125 | 27 | 240 | 0 | 0 | 24 | 0 | 100 | 159 | 112 |
| brown | 0 | 0 | 15 | 0 | 30 | 37 | 44 | 0 | 13 | 0 | 0 | 0 | 0 | 73 | 24 | 62 | 29 | 0 | 0 | 139 | 0 | 0 | 0 |
| wood | 0 | 30 | 0 | 0 | 0 | 0 | 0 | 0 | 0 | 34 | 0 | 0 | 0 | 163 | 15 | 0 | 0 | 0 | 0 | 245 | 0 | 0 | 0 |
| blue | 294 | 0 | 32 | 31 | 19 | 47 | 30 | 0 | 57 | 19 | 0 | 0 | 0 | 23 | 40 | 111 | 0 | 0 | 0 | 0 | 98 | 0 | 80 |
| green | 20 | 0 | 0 | 24 | 13 | 0 | 0 | 0 | 15 | 24 | 0 | 0 | 0 | 0 | 27 | 29 | 456 | 0 | 0 | 0 | 26 | 0 | 32 |
| gray | 21 | 0 | 56 | 14 | 25 | 18 | 0 | 6 | 35 | 38 | 0 | 100 | 0 | 23 | 0 | 61 | 2 | 76 | 0 | 0 | 0 | 59 | 51 |
| on snow | 0 | 0 | 0 | 0 | 0 | 0 | 0 | 0 | 0 | 0 | 50 | 0 | 0 | 0 | 0 | 0 | 0 | 0 | 0 | 0 | 0 | 0 | 0 |
| large | 0 | 25 | 0 | 0 | 0 | 0 | 70 | 0 | 0 | 29 | 0 | 10 | 17 | 0 | 51 | 0 | 85 | 65 | 16 | 0 | 0 | 92 | 0 |
| eating | 0 | 0 | 0 | 0 | 0 | 0 | 0 | 0 | 0 | 0 | 0 | 176 | 0 | 0 | 0 | 0 | 0 | 0 | 0 | 0 | 0 | 0 | 0 |
| red | 0 | 0 | 9 | 99 | 0 | 34 | 15 | 0 | 10 | 0 | 0 | 0 | 0 | 24 | 17 | 74 | 4 | 0 | 0 | 0 | 47 | 0 | 49 |
| metal | 0 | 74 | 0 | 0 | 0 | 0 | 0 | 68 | 0 | 50 | 0 | 0 | 33 | 95 | 28 | 0 | 0 | 0 | 0 | 13 | 0 | 0 | 0 |
| lying | 0 | 0 | 0 | 0 | 0 | 0 | 0 | 0 | 0 | 0 | 0 | 50 | 0 | 0 | 0 | 0 | 0 | 0 | 0 | 0 | 0 | 0 | 0 |
| with people | 0 | 0 | 0 | 0 | 0 | 0 | 0 | 0 | 0 | 0 | 0 | 0 | 0 | 0 | 0 | 0 | 0 | 0 | 0 | 0 | 0 | 0 | 0 |
| on beach | 0 | 0 | 0 | 0 | 0 | 0 | 0 | 0 | 0 | 0 | 0 | 0 | 0 | 0 | 0 | 0 | 0 | 0 | 0 | 0 | 0 | 0 | 0 |
| small | 0 | 10 | 0 | 0 | 0 | 0 | 24 | 0 | 0 | 0 | 0 | 17 | 24 | 0 | 59 | 0 | 79 | 20 | 10 | 0 | 0 | 31 | 0 |
| in forest | 0 | 0 | 0 | 0 | 0 | 0 | 0 | 0 | 0 | 0 | 86 | 0 | 0 | 0 | 0 | 0 | 0 | 0 | 0 | 0 | 0 | 0 | 0 |
| on road | 0 | 0 | 0 | 0 | 0 | 0 | 0 | 0 | 0 | 0 | 0 | 0 | 0 | 0 | 0 | 0 | 0 | 0 | 0 | 0 | 0 | 0 | 0 |
| silver | 0 | 157 | 0 | 10 | 0 | 0 | 0 | 178 | 0 | 0 | 103 | 0 | 51 | 33 | 41 | 0 | 0 | 0 | 0 | 0 | 0 | 35 | 0 |
| running | 0 | 0 | 0 | 0 | 0 | 0 | 0 | 0 | 0 | 0 | 0 | 51 | 0 | 0 | 0 | 0 | 0 | 0 | 0 | 0 | 0 | 0 | 0 |
| in cage | 0 | 0 | 0 | 0 | 0 | 0 | 0 | 0 | 0 | 0 | 0 | 57 | 0 | 0 | 0 | 0 | 0 | 0 | 0 | 0 | 0 | 0 | 0 |
| open | 0 | 0 | 0 | 0 | 0 | 0 | 0 | 0 | 0 | 0 | 0 | 0 | 204 | 0 | 0 | 0 | 0 | 0 | 0 | 12 | 0 | 0 | 0 |
| yellow | 0 | 0 | 0 | 83 | 17 | 0 | 0 | 0 | 0 | 0 | 0 | 0 | 0 | 0 | 0 | 0 | 0 | 0 | 0 | 0 | 0 | 0 | 15 |
| standing | 0 | 0 | 0 | 0 | 0 | 0 | 0 | 0 | 0 | 0 | 0 | 0 | 0 | 0 | 0 | 0 | 0 | 0 | 0 | 0 | 0 | 0 | 0 |
| in city | 0 | 0 | 0 | 0 | 0 | 0 | 0 | 0 | 0 | 0 | 0 | 0 | 0 | 0 | 0 | 0 | 0 | 0 | 0 | 0 | 0 | 0 | 0 |
| sitting | 0 | 0 | 0 | 0 | 0 | 0 | 0 | 0 | 0 | 0 | 0 | 0 | 0 | 0 | 0 | 0 | 0 | 0 | 0 | 0 | 0 | 0 | 0 |
| tall | 0 | 0 | 0 | 0 | 0 | 0 | 0 | 0 | 0 | 0 | 0 | 0 | 0 | 0 | 0 | 0 | 0 | 0 | 0 | 0 | 0 | 0 | 0 |
| in river | 0 | 0 | 0 | 0 | 0 | 0 | 0 | 0 | 0 | 0 | 0 | 0 | 0 | 0 | 0 | 0 | 0 | 0 | 0 | 0 | 0 | 0 | 0 |
| in water | 0 | 0 | 0 | 0 | 0 | 0 | 0 | 0 | 0 | 0 | 85 | 0 | 0 | 0 | 0 | 0 | 0 | 0 | 0 | 0 | 0 | 0 | 0 |
| dark | 14 | 0 | 27 | 0 | 0 | 0 | 0 | 0 | 0 | 0 | 0 | 0 | 0 | 0 | 0 | 36 | 2 | 19 | 0 | 0 | 0 | 0 | 0 |
| at home | 0 | 0 | 0 | 0 | 0 | 0 | 0 | 0 | 0 | 0 | 0 | 126 | 0 | 0 | 0 | 0 | 0 | 0 | 0 | 0 | 0 | 0 | 0 |
| walking | 0 | 0 | 0 | 0 | 0 | 0 | 0 | 0 | 0 | 0 | 0 | 0 | 0 | 0 | 0 | 0 | 0 | 0 | 0 | 0 | 0 | 0 | 0 |
| glass | 0 | 0 | 0 | 0 | 0 | 0 | 0 | 0 | 0 | 0 | 0 | 0 | 0 | 43 | 0 | 0 | 0 | 0 | 0 | 0 | 0 | 0 | 0 |
| orange | 0 | 0 | 0 | 15 | 13 | 0 | 0 | 0 | 12 | 0 | 0 | 0 | 0 | 0 | 0 | 0 | 2 | 0 | 0 | 0 | 16 | 0 | 15 |
| on tree | 0 | 0 | 0 | 0 | 0 | 0 | 0 | 0 | 0 | 0 | 0 | 0 | 0 | 0 | 0 | 0 | 0 | 0 | 0 | 0 | 0 | 0 | 0 |
| in hand | 0 | 0 | 0 | 0 | 0 | 0 | 0 | 0 | 0 | 0 | 0 | 0 | 0 | 0 | 0 | 0 | 0 | 0 | 0 | 0 | 0 | 0 | 0 |
| flying | 0 | 0 | 0 | 0 | 0 | 0 | 0 | 0 | 0 | 0 | 0 | 0 | 0 | 0 | 0 | 0 | 0 | 0 | 0 | 0 | 0 | 0 | 0 |
| plastic | 0 | 31 | 0 | 0 | 0 | 0 | 0 | 18 | 0 | 0 | 0 | 0 | 0 | 0 | 27 | 0 | 0 | 0 | 0 | 0 | 0 | 0 | 0 |
| long | 0 | 0 | 0 | 0 | 0 | 0 | 0 | 0 | 0 | 0 | 0 | 0 | 0 | 0 | 0 | 0 | 0 | 0 | 0 | 0 | 21 | 0 | 0 |
| closed | 0 | 0 | 0 | 0 | 0 | 0 | 0 | 0 | 0 | 0 | 0 | 0 | 20 | 0 | 0 | 0 | 0 | 0 | 0 | 10 | 0 | 0 | 0 |
| on ground | 0 | 0 | 0 | 0 | 0 | 0 | 0 | 0 | 0 | 0 | 0 | 0 | 0 | 0 | 0 | 0 | 0 | 0 | 0 | 0 | 0 | 0 | 0 |
| aside mountain | 0 | 0 | 0 | 0 | 0 | 0 | 0 | 0 | 0 | 0 | 0 | 0 | 0 | 0 | 0 | 0 | 0 | 0 | 0 | 0 | 0 | 0 | 0 |
| pink | 0 | 0 | 0 | 0 | 0 | 18 | 0 | 0 | 8 | 0 | 0 | 0 | 8 | 0 | 0 | 22 | 2 | 0 | 0 | 0 | 60 | 0 | 0 |
| round | 0 | 0 | 0 | 0 | 0 | 0 | 0 | 0 | 0 | 0 | 0 | 0 | 40 | 0 | 0 | 0 | 0 | 0 | 0 | 0 | 0 | 0 | 0 |
| bare | 0 | 0 | 0 | 0 | 0 | 0 | 0 | 0 | 0 | 0 | 0 | 0 | 0 | 0 | 0 | 0 | 0 | 0 | 0 | 0 | 0 | 0 | 0 |
| tan | 0 | 0 | 0 | 0 | 0 | 0 | 0 | 0 | 0 | 0 | 0 | 0 | 0 | 0 | 0 | 0 | 0 | 0 | 0 | 0 | 0 | 0 | 0 |
| baby | 0 | 0 | 0 | 0 | 0 | 0 | 0 | 0 | 0 | 0 | 0 | 0 | 0 | 0 | 0 | 0 | 0 | 0 | 0 | 0 | 0 | 0 | 0 |
| in garage | 0 | 0 | 0 | 0 | 0 | 0 | 0 | 0 | 0 | 0 | 0 | 0 | 0 | 0 | 0 | 0 | 0 | 0 | 0 | 0 | 0 | 0 | 0 |
| yacht | 0 | 0 | 0 | 0 | 0 | 0 | 0 | 0 | 0 | 0 | 0 | 0 | 0 | 0 | 0 | 0 | 0 | 0 | 0 | 0 | 0 | 0 | 0 |
| on track | 0 | 0 | 0 | 0 | 0 | 0 | 0 | 0 | 0 | 0 | 0 | 0 | 0 | 0 | 0 | 0 | 0 | 0 | 0 | 0 | 0 | 0 | 0 |
| at station | 0 | 0 | 0 | 0 | 0 | 0 | 0 | 0 | 0 | 0 | 0 | 0 | 0 | 0 | 0 | 0 | 0 | 0 | 0 | 0 | 0 | 0 | 0 |
| wooden | 0 | 0 | 0 | 0 | 0 | 0 | 0 | 0 | 0 | 0 | 0 | 0 | 0 | 0 | 0 | 0 | 0 | 0 | 0 | 0 | 0 | 0 | 0 |
| on branch | 0 | 0 | 0 | 0 | 0 | 0 | 0 | 0 | 0 | 0 | 0 | 0 | 0 | 0 | 0 | 0 | 0 | 0 | 0 | 0 | 0 | 0 | 0 |
| leafy | 0 | 0 | 0 | 0 | 0 | 0 | 0 | 0 | 0 | 0 | 0 | 0 | 0 | 0 | 0 | 0 | 0 | 0 | 0 | 0 | 0 | 0 | 0 |
| shared | 0 | 0 | 0 | 0 | 0 | 0 | 0 | 0 | 0 | 0 | 0 | 0 | 0 | 0 | 0 | 0 | 0 | 0 | 0 | 0 | 0 | 0 | 0 |
| velodrome | 0 | 0 | 0 | 0 | 0 | 0 | 0 | 0 | 0 | 0 | 0 | 0 | 0 | 0 | 0 | 0 | 0 | 0 | 0 | 0 | 0 | 0 | 0 |
| at wharf | 0 | 0 | 0 | 0 | 0 | 0 | 0 | 0 | 0 | 0 | 0 | 0 | 0 | 0 | 0 | 0 | 0 | 0 | 0 | 0 | 0 | 0 | 0 |
| double decker | 0 | 0 | 0 | 0 | 0 | 0 | 0 | 0 | 0 | 0 | 0 | 0 | 0 | 0 | 0 | 0 | 0 | 0 | 0 | 0 | 0 | 0 | 0 |
| on | 0 | 0 | 0 | 0 | 0 | 0 | 0 | 0 | 0 | 0 | 0 | 98 | 0 | 0 | 0 | 0 | 0 | 0 | 0 | 0 | 0 | 111 | 0 |
| cross bridge | 0 | 0 | 0 | 0 | 0 | 0 | 0 | 0 | 0 | 0 | 0 | 0 | 0 | 0 | 0 | 0 | 0 | 0 | 0 | 0 | 0 | 0 | 0 |
| in sunset | 0 | 0 | 0 | 0 | 0 | 0 | 0 | 0 | 0 | 0 | 0 | 0 | 0 | 0 | 0 | 0 | 0 | 0 | 0 | 0 | 0 | 0 | 0 |
| at heliport | 0 | 0 | 0 | 0 | 0 | 0 | 0 | 0 | 0 | 0 | 0 | 0 | 0 | 0 | 0 | 0 | 0 | 0 | 0 | 0 | 0 | 0 | 0 |
| on bridge | 0 | 0 | 0 | 0 | 0 | 0 | 0 | 0 | 0 | 0 | 0 | 0 | 0 | 0 | 0 | 0 | 0 | 0 | 0 | 0 | 0 | 0 | 0 |
| in burrow | 0 | 0 | 0 | 0 | 0 | 0 | 0 | 0 | 0 | 0 | 0 | 0 | 0 | 0 | 0 | 0 | 0 | 0 | 0 | 0 | 0 | 0 | 0 |
| aside tree | 0 | 0 | 0 | 0 | 0 | 0 | 0 | 0 | 0 | 0 | 0 | 0 | 0 | 0 | 0 | 0 | 0 | 0 | 0 | 0 | 0 | 0 | 0 |
| in zoo | 0 | 0 | 0 | 0 | 0 | 0 | 0 | 0 | 0 | 0 | 0 | 0 | 0 | 0 | 0 | 0 | 0 | 0 | 0 | 0 | 0 | 0 | 0 |
| in pot | 0 | 0 | 0 | 0 | 0 | 0 | 0 | 0 | 0 | 0 | 0 | 0 | 0 | 0 | 0 | 0 | 0 | 0 | 0 | 0 | 0 | 0 | 0 |
| at airport | 0 | 0 | 0 | 0 | 0 | 0 | 0 | 0 | 0 | 0 | 0 | 0 | 0 | 0 | 0 | 0 | 0 | 0 | 0 | 0 | 0 | 0 | 0 |
| on sea | 0 | 0 | 0 | 0 | 0 | 0 | 0 | 0 | 0 | 0 | 0 | 0 | 0 | 0 | 0 | 0 | 0 | 0 | 0 | 0 | 0 | 0 | 0 |
| sleeping | 0 | 0 | 0 | 0 | 0 | 0 | 0 | 0 | 0 | 0 | 0 | 0 | 0 | 0 | 0 | 0 | 0 | 0 | 0 | 0 | 0 | 0 | 0 |
| in desert | 0 | 0 | 0 | 0 | 0 | 0 | 0 | 0 | 0 | 0 | 0 | 0 | 0 | 0 | 0 | 0 | 0 | 0 | 0 | 0 | 0 | 0 | 0 |
| eating grass | 0 | 0 | 0 | 0 | 0 | 0 | 0 | 0 | 0 | 0 | 0 | 0 | 0 | 0 | 0 | 0 | 0 | 0 | 0 | 0 | 0 | 0 | 0 |
| in race | 0 | 0 | 0 | 0 | 0 | 0 | 0 | 0 | 0 | 0 | 0 | 0 | 0 | 0 | 0 | 0 | 0 | 0 | 0 | 0 | 0 | 0 | 0 |
| taking off | 0 | 0 | 0 | 0 | 0 | 0 | 0 | 0 | 0 | 0 | 0 | 0 | 0 | 0 | 0 | 0 | 0 | 0 | 0 | 0 | 0 | 0 | 0 |
| on power line | 0 | 0 | 0 | 0 | 0 | 0 | 0 | 0 | 0 | 0 | 0 | 0 | 0 | 0 | 0 | 0 | 0 | 0 | 0 | 0 | 0 | 0 | 0 |
| in circus | 0 | 0 | 0 | 0 | 0 | 0 | 0 | 0 | 0 | 0 | 0 | 0 | 0 | 0 | 0 | 0 | 0 | 0 | 0 | 0 | 0 | 0 | 0 |
| mountain | 0 | 0 | 0 | 0 | 0 | 0 | 0 | 0 | 0 | 0 | 0 | 0 | 0 | 0 | 0 | 0 | 0 | 0 | 0 | 0 | 0 | 0 | 0 |
| short | 0 | 0 | 0 | 0 | 0 | 0 | 0 | 0 | 0 | 0 | 0 | 0 | 0 | 0 | 0 | 0 | 0 | 0 | 0 | 0 | 0 | 0 | 0 |

| Concepts | | | | | | | | | | | | | | | | | | | | | | |
|---|---|---|---|---|---|---|---|---|---|---|---|---|---|---|---|---|---|---|---|---|---|---|
| Contexts | tower | ocean | spoon | suit | fire hydrant | skateboard | pillow | bed | knife | backpack | bridge | rat | laptop | sink | frame | bowl | coat | bush | cloud | cabinet | shrimp | dress | television |
| open mouth | 0 | 0 | 0 | 0 | 0 | 0 | 0 | 0 | 0 | 0 | 0 | 0 | 0 | 0 | 0 | 0 | 0 | 0 | 0 | 0 | 0 | 0 | 0 |
| on booth | 0 | 0 | 0 | 0 | 0 | 0 | 0 | 0 | 0 | 0 | 0 | 0 | 0 | 0 | 0 | 0 | 0 | 0 | 0 | 0 | 0 | 0 | 0 |
| howling | 0 | 0 | 0 | 0 | 0 | 0 | 0 | 0 | 0 | 0 | 0 | 0 | 0 | 0 | 0 | 0 | 0 | 0 | 0 | 0 | 0 | 0 | 0 |
| brick | 0 | 0 | 0 | 0 | 0 | 0 | 0 | 0 | 0 | 0 | 0 | 0 | 0 | 0 | 0 | 0 | 0 | 0 | 0 | 0 | 0 | 0 | 0 |
| khaki | 0 | 0 | 0 | 0 | 0 | 0 | 0 | 0 | 0 | 0 | 0 | 0 | 0 | 0 | 0 | 0 | 0 | 0 | 0 | 0 | 0 | 0 | 0 |
| at dock | 0 | 0 | 0 | 0 | 0 | 0 | 0 | 0 | 0 | 0 | 0 | 0 | 0 | 0 | 0 | 0 | 0 | 0 | 0 | 0 | 0 | 0 | 0 |
| on head | 0 | 0 | 0 | 0 | 0 | 0 | 0 | 0 | 0 | 0 | 0 | 0 | 0 | 0 | 0 | 0 | 0 | 0 | 0 | 0 | 0 | 0 | 0 |
| around cloud | 0 | 0 | 0 | 0 | 0 | 0 | 0 | 0 | 0 | 0 | 0 | 0 | 0 | 0 | 0 | 0 | 0 | 0 | 0 | 0 | 0 | 0 | 0 |
| climbing | 0 | 0 | 0 | 0 | 0 | 0 | 0 | 0 | 0 | 0 | 0 | 0 | 0 | 0 | 0 | 0 | 0 | 0 | 0 | 0 | 0 | 0 | 0 |
| stone | 0 | 0 | 0 | 0 | 0 | 0 | 0 | 0 | 0 | 26 | 0 | 0 | 0 | 0 | 0 | 0 | 0 | 0 | 0 | 0 | 0 | 0 | 0 |
| on web | 0 | 0 | 0 | 0 | 0 | 0 | 0 | 0 | 0 | 0 | 0 | 0 | 0 | 0 | 0 | 0 | 0 | 0 | 0 | 0 | 0 | 0 | 0 |
| on a stick | 0 | 0 | 0 | 0 | 0 | 0 | 0 | 0 | 0 | 0 | 0 | 0 | 0 | 0 | 0 | 0 | 0 | 0 | 0 | 0 | 0 | 0 | 0 |
| leather | 0 | 0 | 0 | 0 | 0 | 0 | 0 | 0 | 0 | 0 | 0 | 0 | 0 | 0 | 0 | 0 | 0 | 0 | 0 | 0 | 0 | 0 | 0 |
| cutting | 0 | 0 | 0 | 0 | 0 | 0 | 0 | 0 | 0 | 0 | 0 | 0 | 0 | 0 | 0 | 0 | 0 | 0 | 0 | 0 | 0 | 0 | 0 |
| concrete | 0 | 0 | 0 | 0 | 0 | 0 | 0 | 0 | 0 | 29 | 0 | 0 | 0 | 0 | 0 | 0 | 0 | 0 | 0 | 0 | 0 | 0 | 0 |
| at park | 0 | 0 | 0 | 0 | 0 | 0 | 0 | 0 | 0 | 0 | 0 | 0 | 0 | 0 | 0 | 0 | 0 | 0 | 0 | 0 | 0 | 0 | 0 |
| on iceberg | 0 | 0 | 0 | 0 | 0 | 0 | 0 | 0 | 0 | 0 | 0 | 0 | 0 | 0 | 0 | 0 | 0 | 0 | 0 | 0 | 0 | 0 | 0 |
| on shoulder | 0 | 0 | 0 | 0 | 0 | 0 | 0 | 0 | 0 | 0 | 0 | 0 | 0 | 0 | 0 | 0 | 0 | 0 | 0 | 0 | 0 | 0 | 0 |
| at night | 0 | 0 | 0 | 0 | 0 | 0 | 0 | 0 | 0 | 0 | 0 | 0 | 0 | 0 | 0 | 0 | 0 | 0 | 0 | 0 | 0 | 0 | 0 |
| with flower | 0 | 0 | 0 | 0 | 0 | 0 | 0 | 0 | 0 | 0 | 0 | 0 | 0 | 0 | 0 | 0 | 0 | 0 | 0 | 0 | 0 | 0 | 0 |
| grazing | 0 | 0 | 0 | 0 | 0 | 0 | 0 | 0 | 0 | 0 | 0 | 0 | 0 | 0 | 0 | 0 | 0 | 0 | 0 | 0 | 0 | 0 | 0 |
| paper | 0 | 0 | 0 | 0 | 0 | 0 | 0 | 0 | 0 | 0 | 0 | 0 | 0 | 0 | 0 | 0 | 0 | 0 | 0 | 0 | 0 | 0 | 0 |
| spotted | 0 | 0 | 0 | 0 | 0 | 0 | 0 | 0 | 0 | 0 | 0 | 0 | 0 | 0 | 0 | 0 | 0 | 0 | 0 | 0 | 0 | 0 | 0 |
| at yard | 0 | 0 | 0 | 0 | 0 | 0 | 0 | 0 | 0 | 0 | 0 | 0 | 0 | 0 | 0 | 0 | 0 | 0 | 0 | 0 | 0 | 0 | 0 |
| with bee | 0 | 0 | 0 | 0 | 0 | 0 | 0 | 0 | 0 | 0 | 0 | 0 | 0 | 0 | 0 | 0 | 0 | 0 | 0 | 0 | 0 | 0 | 0 |
| hanging | 0 | 0 | 0 | 0 | 0 | 0 | 0 | 0 | 0 | 0 | 0 | 0 | 0 | 0 | 0 | 0 | 0 | 0 | 0 | 0 | 0 | 0 | 0 |
| clean | 0 | 0 | 0 | 0 | 0 | 0 | 0 | 0 | 0 | 0 | 0 | 0 | 26 | 0 | 0 | 0 | 0 | 0 | 0 | 0 | 0 | 0 | 0 |
| square | 0 | 0 | 0 | 0 | 0 | 0 | 0 | 0 | 0 | 0 | 0 | 0 | 14 | 0 | 0 | 0 | 0 | 0 | 0 | 0 | 0 | 0 | 0 |
| subway | 0 | 0 | 0 | 0 | 0 | 0 | 0 | 0 | 0 | 0 | 0 | 0 | 0 | 0 | 0 | 0 | 0 | 0 | 0 | 0 | 0 | 0 | 0 |
| jumping | 0 | 0 | 0 | 0 | 0 | 0 | 0 | 0 | 0 | 0 | 0 | 0 | 0 | 0 | 0 | 0 | 0 | 0 | 0 | 0 | 0 | 0 | 0 |
| on flower | 0 | 0 | 0 | 0 | 0 | 0 | 0 | 0 | 0 | 0 | 0 | 0 | 0 | 0 | 0 | 0 | 0 | 0 | 0 | 0 | 0 | 0 | 0 |
| landing | 0 | 0 | 0 | 0 | 0 | 0 | 0 | 0 | 0 | 0 | 0 | 0 | 0 | 0 | 0 | 0 | 0 | 0 | 0 | 0 | 0 | 0 | 0 |
| porcelain | 0 | 0 | 0 | 0 | 0 | 0 | 0 | 0 | 0 | 0 | 0 | 0 | 66 | 0 | 0 | 0 | 0 | 0 | 0 | 0 | 0 | 0 | 0 |
| at sunset | 0 | 0 | 0 | 0 | 0 | 0 | 0 | 0 | 0 | 0 | 0 | 0 | 0 | 0 | 0 | 0 | 0 | 0 | 0 | 0 | 0 | 0 | 0 |
| on desk | 0 | 0 | 0 | 0 | 0 | 0 | 0 | 0 | 0 | 0 | 0 | 0 | 0 | 0 | 0 | 0 | 0 | 0 | 0 | 0 | 0 | 0 | 0 |
| fighting | 0 | 0 | 0 | 0 | 0 | 0 | 0 | 0 | 0 | 0 | 0 | 0 | 0 | 0 | 0 | 0 | 0 | 0 | 0 | 0 | 0 | 0 | 0 |
| purple | 0 | 0 | 0 | 0 | 0 | 0 | 0 | 0 | 0 | 0 | 0 | 0 | 0 | 0 | 0 | 0 | 0 | 0 | 0 | 0 | 19 | 0 | 0 |
| off | 0 | 0 | 0 | 0 | 0 | 0 | 0 | 0 | 0 | 0 | 0 | 17 | 0 | 0 | 0 | 0 | 0 | 0 | 0 | 0 | 0 | 45 | 0 |
| gold | 0 | 0 | 0 | 0 | 0 | 0 | 0 | 0 | 0 | 0 | 0 | 0 | 0 | 37 | 0 | 0 | 0 | 0 | 0 | 0 | 0 | 0 | 0 |
| dirty | 0 | 0 | 0 | 0 | 0 | 0 | 0 | 0 | 0 | 0 | 0 | 0 | 11 | 0 | 0 | 0 | 0 | 0 | 0 | 0 | 0 | 0 | 0 |
| resting | 0 | 0 | 0 | 0 | 0 | 0 | 0 | 0 | 0 | 0 | 0 | 0 | 0 | 0 | 0 | 0 | 0 | 0 | 0 | 0 | 0 | 0 | 0 |
| beige | 0 | 0 | 0 | 0 | 0 | 0 | 0 | 0 | 0 | 0 | 0 | 0 | 0 | 0 | 0 | 0 | 0 | 0 | 0 | 0 | 0 | 0 | 0 |
| rectangular | 0 | 0 | 0 | 0 | 0 | 0 | 0 | 0 | 0 | 0 | 0 | 0 | 11 | 0 | 0 | 0 | 0 | 0 | 0 | 0 | 0 | 0 | 0 |
| thin | 0 | 0 | 0 | 0 | 0 | 0 | 0 | 0 | 0 | 0 | 0 | 0 | 0 | 0 | 0 | 0 | 0 | 18 | 0 | 0 | 0 | 0 | 0 |
| thick | 0 | 0 | 0 | 0 | 0 | 0 | 0 | 0 | 0 | 0 | 0 | 0 | 0 | 0 | 0 | 0 | 0 | 11 | 0 | 0 | 0 | 0 | 0 |
| in bucket | 0 | 0 | 0 | 0 | 0 | 0 | 0 | 0 | 0 | 0 | 0 | 0 | 0 | 0 | 0 | 0 | 0 | 0 | 0 | 0 | 0 | 0 | 0 |
| city | 0 | 0 | 0 | 0 | 0 | 0 | 0 | 0 | 0 | 0 | 0 | 0 | 0 | 0 | 0 | 0 | 0 | 0 | 0 | 0 | 0 | 0 | 0 |
| in pouch | 0 | 0 | 0 | 0 | 0 | 0 | 0 | 0 | 0 | 0 | 0 | 0 | 0 | 0 | 0 | 0 | 0 | 0 | 0 | 0 | 0 | 0 | 0 |
| in hole | 0 | 0 | 0 | 0 | 0 | 0 | 0 | 0 | 0 | 0 | 51 | 0 | 0 | 0 | 0 | 0 | 0 | 0 | 0 | 0 | 0 | 0 | 0 |
| on bird feeder | 0 | 0 | 0 | 0 | 0 | 0 | 0 | 0 | 0 | 0 | 0 | 0 | 0 | 0 | 0 | 0 | 0 | 0 | 0 | 0 | 0 | 0 | 0 |
| on shelves | 0 | 0 | 0 | 0 | 0 | 0 | 0 | 0 | 0 | 0 | 0 | 0 | 0 | 0 | 0 | 0 | 0 | 0 | 0 | 0 | 0 | 0 | 0 |
| on wall | 0 | 0 | 0 | 0 | 0 | 0 | 0 | 0 | 0 | 0 | 0 | 0 | 0 | 0 | 0 | 0 | 0 | 0 | 0 | 0 | 0 | 0 | 0 |
| on post | 0 | 0 | 0 | 0 | 0 | 0 | 0 | 0 | 0 | 0 | 0 | 0 | 0 | 0 | 0 | 0 | 0 | 0 | 0 | 0 | 0 | 0 | 0 |
| in box | 0 | 0 | 0 | 0 | 0 | 0 | 0 | 0 | 0 | 0 | 0 | 0 | 0 | 0 | 0 | 0 | 0 | 0 | 0 | 0 | 0 | 0 | 0 |
| shiny | 0 | 0 | 0 | 0 | 0 | 0 | 0 | 0 | 0 | 0 | 0 | 0 | 0 | 0 | 0 | 0 | 0 | 0 | 0 | 0 | 0 | 0 | 0 |
| sinking | 0 | 0 | 0 | 0 | 0 | 0 | 0 | 0 | 0 | 0 | 0 | 0 | 0 | 0 | 0 | 0 | 0 | 0 | 0 | 0 | 0 | 0 | 0 |
| with cargo | 0 | 0 | 0 | 0 | 0 | 0 | 0 | 0 | 0 | 0 | 0 | 0 | 0 | 0 | 0 | 0 | 0 | 0 | 0 | 0 | 0 | 0 | 0 |
| bright | 0 | 0 | 0 | 0 | 0 | 0 | 0 | 0 | 0 | 0 | 0 | 0 | 0 | 0 | 0 | 0 | 0 | 0 | 0 | 0 | 0 | 0 | 0 |
| in shell | 0 | 0 | 0 | 0 | 0 | 0 | 0 | 0 | 0 | 0 | 0 | 0 | 0 | 0 | 0 | 0 | 0 | 0 | 0 | 0 | 0 | 0 | 0 |
| aside traffic light | 0 | 0 | 0 | 0 | 0 | 0 | 0 | 0 | 0 | 0 | 0 | 0 | 0 | 0 | 0 | 0 | 0 | 0 | 0 | 0 | 0 | 0 | 0 |
| empty | 0 | 0 | 0 | 0 | 0 | 0 | 0 | 0 | 0 | 0 | 0 | 0 | 0 | 0 | 16 | 0 | 0 | 0 | 0 | 0 | 0 | 0 | 0 |
| cross tunnel | 0 | 0 | 0 | 0 | 0 | 0 | 0 | 0 | 0 | 0 | 0 | 0 | 0 | 0 | 0 | 0 | 0 | 0 | 0 | 0 | 0 | 0 | 0 |
| full | 0 | 0 | 0 | 0 | 0 | 0 | 0 | 0 | 0 | 0 | 0 | 0 | 0 | 0 | 16 | 0 | 0 | 0 | 0 | 0 | 0 | 0 | 0 |
| playing | 0 | 0 | 0 | 0 | 0 | 0 | 0 | 0 | 0 | 0 | 0 | 0 | 0 | 0 | 0 | 0 | 0 | 0 | 0 | 0 | 0 | 0 | 0 |
| light brown | 0 | 0 | 0 | 0 | 0 | 0 | 0 | 0 | 0 | 0 | 0 | 0 | 0 | 0 | 0 | 0 | 0 | 0 | 0 | 0 | 0 | 0 | 0 |
| staring | 0 | 0 | 0 | 0 | 0 | 0 | 0 | 0 | 0 | 0 | 0 | 0 | 0 | 0 | 0 | 0 | 0 | 0 | 0 | 0 | 0 | 0 | 0 |
| colorful | 0 | 0 | 0 | 0 | 0 | 0 | 0 | 0 | 0 | 0 | 0 | 0 | 0 | 0 | 0 | 0 | 0 | 0 | 0 | 0 | 0 | 0 | 0 |
| dark brown | 0 | 0 | 0 | 0 | 0 | 0 | 0 | 0 | 0 | 0 | 0 | 0 | 0 | 0 | 0 | 0 | 0 | 0 | 0 | 0 | 0 | 0 | 0 |
| young | 0 | 0 | 0 | 0 | 0 | 0 | 0 | 0 | 0 | 0 | 0 | 0 | 0 | 0 | 0 | 0 | 0 | 0 | 0 | 0 | 0 | 0 | 0 |
| dark blue | 8 | 0 | 0 | 0 | 0 | 0 | 0 | 0 | 0 | 0 | 0 | 0 | 0 | 0 | 0 | 0 | 0 | 0 | 0 | 0 | 0 | 0 | 0 |

**Contexts**

| Concepts | grass | water | outdoor | dim | white | rock | autumn | black | brown | wood | blue | green | gray | on snow | large | eating | red | metal | lying | with people | on beach | small | in forest |
|---|---|---|---|---|---|---|---|---|---|---|---|---|---|---|---|---|---|---|---|---|---|---|---|
| t-shirt | 0 | 0 | 0 | 0 | 186 | 0 | 0 | 112 | 0 | 0 | 80 | 32 | 51 | 0 | 0 | 0 | 49 | 0 | 0 | 0 | 0 | 0 | 0 |
| sweater | 0 | 0 | 0 | 0 | 62 | 0 | 0 | 102 | 36 | 0 | 90 | 33 | 116 | 0 | 0 | 0 | 47 | 0 | 0 | 0 | 0 | 0 | 0 |
| surfboard | 0 | 0 | 0 | 0 | 265 | 0 | 0 | 18 | 11 | 0 | 111 | 0 | 0 | 0 | 24 | 0 | 21 | 0 | 0 | 0 | 0 | 12 | 0 |
| tie | 0 | 0 | 0 | 0 | 44 | 0 | 0 | 113 | 19 | 0 | 104 | 21 | 36 | 0 | 0 | 0 | 75 | 0 | 0 | 0 | 0 | 0 | 0 |
| fork | 0 | 0 | 0 | 0 | 27 | 0 | 0 | 14 | 0 | 0 | 0 | 0 | 10 | 0 | 0 | 0 | 6 | 118 | 0 | 0 | 0 | 0 | 0 |
| couch | 0 | 0 | 0 | 0 | 81 | 0 | 0 | 41 | 160 | 0 | 46 | 28 | 43 | 0 | 0 | 0 | 38 | 0 | 0 | 0 | 0 | 0 | 0 |
| keyboard | 0 | 0 | 0 | 0 | 197 | 0 | 0 | 215 | 0 | 0 | 3 | 0 | 50 | 0 | 0 | 0 | 0 | 0 | 0 | 0 | 0 | 0 | 0 |
| curtain | 0 | 0 | 0 | 0 | 236 | 0 | 0 | 20 | 50 | 0 | 53 | 29 | 18 | 0 | 0 | 0 | 42 | 0 | 0 | 0 | 0 | 0 | 0 |

**Contexts**

| Concepts | on road | silver | running | in cage | open | yellow | standing | in city | sitting | tall | in river | in water | dark | at home | walking | glass | orange | on tree | in hand | flying | plastic | long | closed |
|---|---|---|---|---|---|---|---|---|---|---|---|---|---|---|---|---|---|---|---|---|---|---|---|
| t-shirt | 0 | 0 | 0 | 0 | 0 | 15 | 0 | 0 | 0 | 0 | 0 | 0 | 0 | 0 | 0 | 0 | 15 | 0 | 0 | 0 | 0 | 0 | 0 |
| sweater | 0 | 0 | 0 | 0 | 0 | 0 | 0 | 0 | 0 | 0 | 0 | 0 | 18 | 0 | 0 | 0 | 0 | 0 | 0 | 0 | 0 | 0 | 0 |
| surfboard | 0 | 0 | 0 | 0 | 0 | 36 | 0 | 0 | 0 | 0 | 0 | 0 | 0 | 0 | 0 | 0 | 21 | 0 | 0 | 0 | 0 | 0 | 0 |
| tie | 0 | 0 | 0 | 0 | 0 | 0 | 0 | 0 | 0 | 0 | 0 | 0 | 17 | 0 | 0 | 0 | 18 | 0 | 0 | 0 | 0 | 48 | 0 |
| fork | 0 | 274 | 0 | 0 | 0 | 0 | 0 | 0 | 0 | 0 | 0 | 0 | 0 | 0 | 0 | 0 | 0 | 0 | 0 | 0 | 0 | 0 | 0 |
| couch | 0 | 0 | 0 | 0 | 0 | 0 | 0 | 0 | 0 | 0 | 0 | 0 | 0 | 0 | 0 | 0 | 0 | 0 | 0 | 0 | 0 | 0 | 0 |
| keyboard | 0 | 11 | 0 | 0 | 0 | 0 | 0 | 0 | 0 | 0 | 0 | 0 | 4 | 0 | 0 | 0 | 0 | 0 | 0 | 0 | 0 | 0 | 0 |
| curtain | 0 | 0 | 0 | 0 | 0 | 15 | 0 | 0 | 0 | 0 | 0 | 0 | 0 | 0 | 0 | 0 | 0 | 0 | 0 | 0 | 0 | 0 | 0 |

**Contexts**

| Concepts | on ground | aside mountain | pink | round | bare | tan | baby | in garage | yacht | on track | at station | wooden | in farm | on branch | leafy | shared | velodrome | at wharf | double decker | on | cross bridge | in sunset | at heliport |
|---|---|---|---|---|---|---|---|---|---|---|---|---|---|---|---|---|---|---|---|---|---|---|---|
| t-shirt | 0 | 0 | 0 | 0 | 0 | 0 | 0 | 0 | 0 | 0 | 0 | 0 | 0 | 0 | 0 | 0 | 0 | 0 | 0 | 0 | 0 | 0 | 0 |
| sweater | 0 | 0 | 28 | 0 | 0 | 0 | 0 | 0 | 0 | 0 | 0 | 0 | 0 | 0 | 0 | 0 | 0 | 0 | 0 | 0 | 0 | 0 | 0 |
| surfboard | 0 | 0 | 11 | 0 | 0 | 0 | 0 | 0 | 0 | 0 | 0 | 0 | 0 | 0 | 0 | 0 | 0 | 0 | 0 | 0 | 0 | 0 | 0 |
| tie | 0 | 0 | 20 | 0 | 0 | 0 | 0 | 0 | 0 | 0 | 0 | 0 | 0 | 0 | 0 | 0 | 0 | 0 | 0 | 0 | 0 | 0 | 0 |
| fork | 0 | 0 | 0 | 0 | 0 | 0 | 0 | 0 | 0 | 0 | 0 | 0 | 0 | 0 | 0 | 0 | 0 | 0 | 0 | 0 | 0 | 0 | 0 |
| couch | 0 | 0 | 0 | 0 | 0 | 39 | 0 | 0 | 0 | 0 | 0 | 0 | 0 | 0 | 0 | 0 | 0 | 0 | 0 | 0 | 0 | 0 | 0 |
| keyboard | 0 | 0 | 0 | 0 | 0 | 0 | 0 | 0 | 0 | 0 | 0 | 0 | 0 | 0 | 0 | 0 | 0 | 0 | 0 | 0 | 0 | 0 | 0 |
| curtain | 0 | 0 | 0 | 0 | 0 | 19 | 0 | 0 | 0 | 0 | 0 | 0 | 0 | 0 | 0 | 0 | 0 | 0 | 0 | 0 | 0 | 0 | 0 |

**Contexts**

| Concepts | on bridge | in burrow | aside tree | in zoo | in pot | at airport | on sea | sleeping | in desert | eating grass | in race | taking off | on power line | in circus | mountain | short | open mouth | on booth | howling | brick | khaki | at dock | on head |
|---|---|---|---|---|---|---|---|---|---|---|---|---|---|---|---|---|---|---|---|---|---|---|---|
| t-shirt | 0 | 0 | 0 | 0 | 0 | 0 | 0 | 0 | 0 | 0 | 0 | 0 | 0 | 0 | 0 | 0 | 0 | 0 | 0 | 0 | 0 | 0 | 0 |
| sweater | 0 | 0 | 0 | 0 | 0 | 0 | 0 | 0 | 0 | 0 | 0 | 0 | 0 | 0 | 0 | 0 | 0 | 0 | 0 | 0 | 0 | 0 | 0 |
| surfboard | 0 | 0 | 0 | 0 | 0 | 0 | 0 | 0 | 0 | 0 | 0 | 0 | 0 | 0 | 0 | 0 | 0 | 0 | 0 | 0 | 0 | 0 | 0 |
| tie | 0 | 0 | 0 | 0 | 0 | 0 | 0 | 0 | 0 | 0 | 0 | 0 | 0 | 0 | 0 | 0 | 0 | 0 | 0 | 0 | 0 | 0 | 0 |
| fork | 0 | 0 | 0 | 0 | 0 | 0 | 0 | 0 | 0 | 0 | 0 | 0 | 0 | 0 | 0 | 0 | 0 | 0 | 0 | 0 | 0 | 0 | 0 |
| couch | 0 | 0 | 0 | 0 | 0 | 0 | 0 | 0 | 0 | 0 | 0 | 0 | 0 | 0 | 0 | 0 | 0 | 0 | 0 | 0 | 0 | 0 | 0 |
| keyboard | 0 | 0 | 0 | 0 | 0 | 0 | 0 | 0 | 0 | 0 | 0 | 0 | 0 | 0 | 0 | 0 | 0 | 0 | 0 | 0 | 0 | 0 | 0 |
| curtain | 0 | 0 | 0 | 0 | 0 | 0 | 0 | 0 | 0 | 0 | 0 | 0 | 0 | 0 | 0 | 0 | 0 | 0 | 0 | 0 | 0 | 0 | 0 |

**Contexts**

| Concepts | around cloud | climbing | stone | on web | on a stick | leather | cutting | concrete | at park | on iceberg | on shoulder | at night | with flower | grazing | paper | spotted | at yard | with bee | hanging | clean | square | subway | jumping |
|---|---|---|---|---|---|---|---|---|---|---|---|---|---|---|---|---|---|---|---|---|---|---|---|
| t-shirt | 0 | 0 | 0 | 0 | 0 | 0 | 0 | 0 | 0 | 0 | 0 | 0 | 0 | 0 | 0 | 0 | 0 | 0 | 0 | 0 | 0 | 0 | 0 |
| sweater | 0 | 0 | 0 | 0 | 0 | 0 | 0 | 0 | 0 | 0 | 0 | 0 | 0 | 0 | 0 | 0 | 0 | 0 | 0 | 0 | 0 | 0 | 0 |
| surfboard | 0 | 0 | 0 | 0 | 0 | 0 | 0 | 0 | 0 | 0 | 0 | 0 | 0 | 0 | 0 | 0 | 0 | 0 | 0 | 0 | 0 | 0 | 0 |
| tie | 0 | 0 | 0 | 0 | 0 | 0 | 0 | 0 | 0 | 0 | 0 | 0 | 0 | 0 | 0 | 0 | 0 | 0 | 0 | 0 | 0 | 0 | 0 |
| fork | 0 | 0 | 0 | 0 | 0 | 0 | 0 | 0 | 0 | 0 | 0 | 0 | 0 | 0 | 0 | 0 | 0 | 0 | 0 | 0 | 0 | 0 | 0 |
| couch | 0 | 0 | 0 | 0 | 0 | 0 | 0 | 0 | 0 | 0 | 0 | 0 | 0 | 0 | 0 | 0 | 0 | 0 | 0 | 0 | 0 | 0 | 0 |
| keyboard | 0 | 0 | 0 | 0 | 0 | 0 | 0 | 0 | 0 | 0 | 0 | 0 | 0 | 0 | 0 | 0 | 0 | 0 | 0 | 0 | 0 | 0 | 0 |
| curtain | 0 | 0 | 0 | 0 | 0 | 0 | 0 | 0 | 0 | 0 | 0 | 0 | 0 | 0 | 0 | 0 | 0 | 0 | 0 | 0 | 0 | 0 | 0 |

**Contexts**

| Concepts | on flower | landing | porcelain | at sunset | on desk | fighting | purple | off | gold | dirty | resting | beige | rectangular | thin | thick | in bucket | city | in pouch | in hole | on bird feeder | on shelves | on wall | on post |
|---|---|---|---|---|---|---|---|---|---|---|---|---|---|---|---|---|---|---|---|---|---|---|---|
| t-shirt | 0 | 0 | 0 | 0 | 0 | 0 | 0 | 0 | 0 | 0 | 0 | 0 | 0 | 0 | 0 | 0 | 0 | 0 | 0 | 0 | 0 | 0 | 0 |
| sweater | 0 | 0 | 0 | 0 | 0 | 0 | 0 | 0 | 0 | 0 | 0 | 0 | 0 | 0 | 0 | 0 | 0 | 0 | 0 | 0 | 0 | 0 | 0 |
| surfboard | 0 | 0 | 0 | 0 | 0 | 0 | 0 | 0 | 0 | 0 | 0 | 0 | 0 | 0 | 0 | 0 | 0 | 0 | 0 | 0 | 0 | 0 | 0 |
| tie | 0 | 0 | 0 | 0 | 0 | 0 | 15 | 0 | 0 | 0 | 0 | 0 | 0 | 0 | 0 | 0 | 0 | 0 | 0 | 0 | 0 | 0 | 0 |
| fork | 0 | 0 | 0 | 0 | 0 | 0 | 0 | 0 | 0 | 0 | 0 | 0 | 0 | 0 | 0 | 0 | 0 | 0 | 0 | 0 | 0 | 0 | 0 |
| couch | 0 | 0 | 0 | 0 | 0 | 0 | 0 | 0 | 0 | 0 | 0 | 14 | 0 | 0 | 0 | 0 | 0 | 0 | 0 | 0 | 0 | 0 | 0 |
| keyboard | 0 | 0 | 0 | 0 | 0 | 0 | 0 | 0 | 0 | 0 | 0 | 3 | 0 | 0 | 0 | 0 | 0 | 0 | 0 | 0 | 0 | 0 | 0 |
| curtain | 0 | 0 | 0 | 0 | 0 | 0 | 0 | 0 | 0 | 0 | 0 | 0 | 0 | 0 | 0 | 0 | 0 | 0 | 0 | 0 | 0 | 0 | 0 |

**Contexts**

| Concepts | in box | shiny | sinking | with cargo | bright | in shell | aside traffic light | empty | cross tunnel | full | playing | light brown | staring | colorful | dark brown | young | dark blue |
|---|---|---|---|---|---|---|---|---|---|---|---|---|---|---|---|---|---|
| t-shirt | 0 | 0 | 0 | 0 | 0 | 0 | 0 | 0 | 0 | 0 | 0 | 0 | 0 | 0 | 0 | 0 | 0 |
| sweater | 0 | 0 | 0 | 0 | 0 | 0 | 0 | 0 | 0 | 0 | 0 | 0 | 0 | 0 | 0 | 0 | 0 |
| surfboard | 0 | 0 | 0 | 0 | 0 | 0 | 0 | 0 | 0 | 0 | 0 | 0 | 0 | 0 | 0 | 0 | 0 |
| tie | 0 | 0 | 0 | 0 | 0 | 0 | 0 | 0 | 0 | 0 | 0 | 0 | 0 | 24 | 0 | 0 | 0 |
| fork | 0 | 0 | 0 | 0 | 0 | 0 | 0 | 0 | 0 | 0 | 0 | 0 | 0 | 0 | 0 | 0 | 0 |
| couch | 0 | 0 | 0 | 0 | 0 | 0 | 0 | 0 | 0 | 0 | 0 | 0 | 0 | 0 | 0 | 0 | 0 |
| keyboard | 0 | 0 | 0 | 0 | 0 | 0 | 0 | 0 | 0 | 0 | 0 | 0 | 0 | 0 | 0 | 0 | 0 |
| curtain | 0 | 0 | 0 | 0 | 0 | 0 | 0 | 0 | 0 | 0 | 0 | 0 | 0 | 0 | 0 | 0 | 0 |

Table 6: Experiments of state-of-the-art few-shot classification methods under 5-way 1- and 5-shot setting. Three common backbones are used and the 95% confidence intervals are displayed.

| Method | Backbone | Type | 5-way 1-shot | 5-way 5-shot |
|---|---|---|---|---|
| **Versa** (Gordon et al., 2018) | Conv64F | Meta | $20.00 \pm 0.00$ | $51.95 \pm 0.35$ |
| **R2D2** (Bertinetto et al., 2019) | Conv64F | Meta | $39.56 \pm 0.36$ | $51.29 \pm 0.36$ |
| **ANIL** (Raghu et al., 2020) | Conv64F | Meta | $37.24 \pm 0.36$ | $48.04 \pm 0.39$ |
| **CAN** (Hou et al., 2019) | Conv64F | Metric | $43.02 \pm 0.42$ | $55.48 \pm 0.40$ |
| **ProtoNet** (Snell et al., 2017) | Conv64F | Metric | $37.92 \pm 0.35$ | $51.39 \pm 0.38$ |
| **DN4** (Li et al., 2019a) | Conv64F | Metric | $38.87 \pm 0.36$ | $52.72 \pm 0.38$ |
| **Baseline** (Chen et al., 2019) | Conv64F | Fine-tuning | $37.97 \pm 0.32$ | $52.08 \pm 0.35$ |
| **Baseline++** (Chen et al., 2019) | Conv64F | Fine-tuning | $40.45 \pm 0.36$ | $53.42 \pm 0.37$ |
| **Versa** (Gordon et al., 2018) | ResNet12 | Meta | $39.64 \pm 0.39$ | $53.06 \pm 0.40$ |
| **R2D2** (Bertinetto et al., 2019) | ResNet12 | Meta | $45.25 \pm 0.42$ | $60.14 \pm 0.37$ |
| **ANIL** (Raghu et al., 2020) | ResNet12 | Meta | $36.58 \pm 0.38$ | $50.54 \pm 0.40$ |
| **CAN** (Hou et al., 2019) | ResNet12 | Metric | $48.93 \pm 0.42$ | $62.36 \pm 0.40$ |
| **ProtoNet** (Snell et al., 2017) | ResNet12 | Metric | $42.69 \pm 0.39$ | $59.50 \pm 0.40$ |
| **DN4** (Li et al., 2019a) | ResNet12 | Metric | $45.04 \pm 0.39$ | $57.68 \pm 0.39$ |
| **Baseline** (Chen et al., 2019) | ResNet12 | Fine-tuning | $46.78 \pm 0.41$ | $60.78 \pm 0.40$ |
| **Baseline++** (Chen et al., 2019) | ResNet12 | Fine-tuning | $46.95 \pm 0.42$ | $58.50 \pm 0.36$ |
| **Versa** (Gordon et al., 2018) | ResNet18 | Meta | $38.88 \pm 0.36$ | $53.18 \pm 0.36$ |
| **R2D2** (Bertinetto et al., 2019) | ResNet18 | Meta | $45.26 \pm 0.38$ | $58.53 \pm 0.38$ |
| **ANIL** (Raghu et al., 2020) | ResNet18 | Meta | $37.26 \pm 0.36$ | $49.42 \pm 0.39$ |
| **CAN** (Hou et al., 2019) | ResNet18 | Metric | $48.18 \pm 0.42$ | $62.38 \pm 0.41$ |
| **ProtoNet** (Snell et al., 2017) | ResNet18 | Metric | $43.14 \pm 0.37$ | $57.84 \pm 0.39$ |
| **DN4** (Li et al., 2019a) | ResNet18 | Metric | $43.73 \pm 0.38$ | $56.56 \pm 0.39$ |
| **Baseline** (Chen et al., 2019) | ResNet18 | Fine-tuning | $46.49 \pm 0.41$ | $60.22 \pm 0.39$ |
| **Baseline++** (Chen et al., 2019) | ResNet18 | Fine-tuning | $47.26 \pm 0.40$ | $60.10 \pm 0.38$ |

Table 7: Experiments of state-of-the-art few-shot classification methods under 10-way 1- and 5-shot setting. Three common backbones are used and the 95% confidence intervals are displayed.

| Method | Backbone | Type | 10-way 1-shot | 10-way 5-shot |
|---|---|---|---|---|
| **Versa** (Gordon et al., 2018) | Conv64F | Meta | $10.00 \pm 0.00$ | $36.88 \pm 0.21$ |
| **R2D2** (Bertinetto et al., 2019) | Conv64F | Meta | $24.80 \pm 0.20$ | $35.98 \pm 0.21$ |
| **ANIL** (Raghu et al., 2020) | Conv64F | Meta | $22.39 \pm 0.19$ | $33.05 \pm 0.22$ |
| **CAN** (Hou et al., 2019) | Conv64F | Metric | $28.19 \pm 0.23$ | $39.06 \pm 0.23$ |
| **ProtoNet** (Snell et al., 2017) | Conv64F | Metric | $23.72 \pm 0.19$ | $35.49 \pm 0.22$ |
| **DN4** (Li et al., 2019a) | Conv64F | Metric | $26.26 \pm 0.21$ | $37.54 \pm 0.22$ |
| **Baseline** (Chen et al., 2019) | Conv64F | Fine-tuning | $23.71 \pm 0.18$ | $35.70 \pm 0.21$ |
| **Baseline++** (Chen et al., 2019) | Conv64F | Fine-tuning | $25.55 \pm 0.20$ | $36.34 \pm 0.21$ |
| **Versa** (Gordon et al., 2018) | ResNet12 | Meta | $10.00 \pm 0.00$ | $40.21 \pm 0.22$ |
| **R2D2** (Bertinetto et al., 2019) | ResNet12 | Meta | $31.16 \pm 0.23$ | $42.10 \pm 0.22$ |
| **ANIL** (Raghu et al., 2020) | ResNet12 | Meta | $22.94 \pm 0.20$ | $33.77 \pm 0.22$ |
| **CAN** (Hou et al., 2019) | ResNet12 | Metric | $31.92 \pm 0.24$ | $44.78 \pm 0.23$ |
| **ProtoNet** (Snell et al., 2017) | ResNet12 | Metric | $29.59 \pm 0.22$ | $42.38 \pm 0.24$ |
| **DN4** (Li et al., 2019a) | ResNet12 | Metric | $30.69 \pm 0.22$ | $40.31 \pm 0.21$ |
| **Baseline** (Chen et al., 2019) | ResNet12 | Fine-tuning | $31.88 \pm 0.24$ | $43.40 \pm 0.23$ |
| **Baseline++** (Chen et al., 2019) | ResNet12 | Fine-tuning | $31.01 \pm 0.23$ | $39.89 \pm 0.21$ |
| **Versa** (Gordon et al., 2018) | ResNet18 | Meta | $10.00 \pm 0.00$ | $36.51 \pm 0.21$ |
| **R2D2** (Bertinetto et al., 2019) | ResNet18 | Meta | $29.73 \pm 0.22$ | $39.74 \pm 0.21$ |
| **ANIL** (Raghu et al., 2020) | ResNet18 | Meta | $10.47 \pm 0.10$ | $33.83 \pm 0.23$ |
| **CAN** (Hou et al., 2019) | ResNet18 | Metric | $33.16 \pm 0.25$ | $43.47 \pm 0.23$ |
| **ProtoNet** (Snell et al., 2017) | ResNet18 | Metric | $29.34 \pm 0.22$ | $41.34 \pm 0.22$ |
| **DN4** (Li et al., 2019a) | ResNet18 | Metric | $29.05 \pm 0.21$ | $30.12 \pm 0.78$ |
| **Baseline** (Chen et al., 2019) | ResNet18 | Fine-tuning | $33.12 \pm 0.25$ | $44.63 \pm 0.24$ |
| **Baseline++** (Chen et al., 2019) | ResNet18 | Fine-tuning | $32.19 \pm 0.23$ | $43.32 \pm 0.22$ |

