# OpenReview forum: "MetaCoCo: A New Few-Shot Classification Benchmark with Spurious Correlation"
_ICLR.cc/2024/Conference — ICLR 2024 poster_

### Official Review · Reviewer_xD6Q · 2023-10-29

**Soundness:** 3 good
**Presentation:** 3 good
**Contribution:** 3 good
**Rating:** 8
**Confidence:** 3

**Summary:**

The authors present a large-scale, diverse, and realistic environment benchmark for spurious-correlation few-shot classification. Extensive experiments on the proposed benchmark are performed to evaluate the state-of-the-art methods in FSC, cross-domain shifts, and self-supervised learning. The experimental results show that the performance of the existing methods degrades significantly in the presence of spurious correlation shifts.

**Strengths:**

1. The paper is well-written and neatly presented.
2. Exploring the spurious-correlation few-shot classification is a very interesting task.
3. Sufficient experimental results show that spurious correlation shifts damage the model performance.

**Weaknesses:**

1. Contributions are not clearly and accurately stated.
2. The explanation for the spurious-correlation few-shot classification is not easy to understand.

**Questions:**

1. In the experiment, why not use the most recent methods for comparison? I have only listed some references for 2021-2022, and the authors can use other recent methods rather than the enumerated ones.  I'm just curious about how recent methods perform on the MetaCoCo dataset.
2. How to understand the conceptual and contextual information mentioned in the manuscript?
3. The network structure of ResNet-50 is not explained in EXPERIMENTAL SETUP, it is suggested that the author explain it in backbone architectures.

[1]  Yang, Zhanyuan, Jinghua Wang, and Yingying Zhu. "Few-shot classification with contrastive learning." European Conference on Computer Vision. Cham: Springer Nature Switzerland, 2022.
[2]  Wertheimer, Davis, Luming Tang, and Bharath Hariharan. "Few-shot classification with feature map reconstruction networks." Proceedings of the IEEE/CVF conference on computer vision and pattern recognition. 2021.
[3]  Wang, Heng, et al. "Revisit Finetuning strategy for Few-Shot Learning to Transfer the Emdeddings." The Eleventh International Conference on Learning Representations. 2022.

---

> ### Author Response · Authors · 2023-11-21
> **Responses by Authors (Part 1): Added contribution summary and further concept clarifications**
>
> We sincerely appreciate the reviewer’s great efforts and insightful comments to improve our manuscript. In below, we address these concerns point by point and try our best to update the manuscript accordingly.
>
> > **[W1] Contributions are not clearly and accurately stated.**
>
> **Response:** We thank the reviewer for pointing out this issue. **We have added a concise and clear contribution summary of our paper in the “Introduction” section.** In a nutshell, our main contributions can be summarized as follows:
>
> -	We propose a large-scale few-shot classification benchmark with spurious-correlation shifts arising from various contexts in real-world scenarios, named MetaCoCo, in which each class (or concept) consists of different backgrounds (or contexts).
>
> -	We further propose a metric by using a pre-trained vision-language model to quantify
> and compare the extent of spurious correlations on MetaCoCo and other benchmarks.
>
> -	We conduct extensive experiments on MetaCoCo to evaluate and compare most recent methods in few-shot classification, domain shifts, overfitting, and self-supervised learning.
>
> > **[W2] How to understand the conceptual and contextual information mentioned in the manuscript?**
>
> **Response:** Thank you for the comment. In this paper, we follow the existing literature addressing out-of-distribution problems in computer vision [1, 2, 3] to adopt the terminologies “conceptual information” and “contextual information”. Specifically, **the conceptual information in an image refers to the object of the class,** while **the contextual information in an image refers to the background or the environment.** For example, consider an image with a dog in green grass, and the true class label is "dog". Then "dog's nose" can be regarded as a conceptual information, while "green grass" can be regarded as a contextual information. As shown in Figure 1 (b), the conceptual information includes “dog”, “bicycle”, “train”, “cat”, etc., whereas the contextual information includes “autumn”, “snow”, “rock”, “water”, etc.
>
> > **[W3] The explanation for the spurious-correlation few-shot classification is not easy to understand.**
>
> **Response:** We continue with our response to W2 to further explain the spurious-correlation in few-shot classification. For the aforementioned example, that is, an image with a dog in green grass, a well-trained classifier may capture both conceptual and contextual information: "dog's nose" and "green grass". However, given that the true label of the image is “dog”, we know that **the classifier produces a spurious correlation between the contextual information "green grass" and the image label “dog”.**
>
> Different from the widely known Meta-dataset with cross-domain shifts shown in Figure 1 (a), the proposed MetaCoCo as shown in Figure 1 (b) further emphasizes the existence of spurious-correlations in the real-world scenarios. For example, in base classes we have an image "dog in green grass" (and the true class label is "dog"), while in novel classes we instead have an image "cat in green grass" (and the true class label is "cat"). **Thus, we need the classifier to be able to distinguish between these conceptual and contextual information.**

---

> ### Author Response · Authors · 2023-11-21
> **Responses by Authors (Part 2): Added extensive experiments comparing the most recent few-shot classification methods**
>
> > **[W4] How recent methods perform on the MetaCoCo dataset?**
>
> **Response:** We thank the useful suggestions from the reviewer. **We added more experiments to compare the performance of the 8 most recent few-shot classification methods on the proposed MetaCoCo.** The experimental results are shown as below.
>
> | Method | Conference | Backbone | Type | GL | LL | TT | $1$-shot | $5$-shot |
> |:---------| :---------: | :---------: | :---------: | :---------: | :---------: | :---------: | :---------: | :---------: |
> | CDKT+ML [4] | NeurIPS 2023 | ResNet18 | Meta |  | &check; | &check; | 44.86 | 61.42  |
> | CDKT+PL [4] | NeurIPS 2023 | ResNet18 | Meta |  | &check; | &check; |43.21 | 59.87 |
> | DeepBDC [5] | CVPR 2022 | ResNet12 | Metric |  | &check; | &check; |46.78 | 62.54 |
> | TSA+DETA [6] | ICCV 2023 | ResNet18 | Metric | | &check; | &check; | 51.42 | 61.58 |
> | PUTM [7] | ICCV 2023 | ResNet18 | Metric | | &check; | &check; | 60.23 | 72.36 |
> | FRN [8] | CVPR 2021 | ResNet12 | Fine-tuning | &check; | | &check; | 50.23 | 60.56 |
> | Yang et al [9] | ECCV 2022 | ResNet12 | Fine-tuning | &check; | | &check; | 58.01 | 69.32 |
> | LP-FT-FB [10] | ICLR 2023 | ResNet12 | Fine-tuning | &check; | &check; | &check; | 56.21 | 70.21 |
> | | | | | | | | | |
>
> From the table, we find that: (1) The performance of these methods is significantly reduced on our proposed MetaCoCo dataset. (2) The performance of fine-tuning based methods is better than other methods, such as LP-FT-FB [10].
>
> **In addition to the methods mentioned by the reviewer, we also verify diverse methods including vision-language pre-trained methods.**
>
> | Method | Conference | Backbone | Type | Base (1-shot) | New (1-shot) | HM (1-shot) | Base (16-shot) | New (16-shot) | HM (16-shot) |
> |:---------| :---------: | :---------: | :---------: | :---------: | :---------: | :---------: | :---------: | :---------: | :---------: |
> | CLIP [11] | ICML 2021 | ResNet50 | Zero-shot CLIP | 68.12 | 72.19 | 70.10 | 68.12 | 72.19 | 70.10 |
> | CoOp [12] | IJCV 2022 | ResNet50 | Prompt-tuning CLIP | 61.22 | 63.62 | 62.40 | 47.55 | 40.90 | 43.98 |
> | CoCoOp [13] | CVPR 2022 | ResNet50 | Prompt-tuning CLIP | 59.86 | 63.88 | 61.80 | 36.88 | 46.10 | 40.98 |
> | KgCoOp [14] | CVPR 2023 | ResNet50 | Prompt-tuning CLIP | 67.95 | 72.05 | 69.94 | 49.18 | 48.78 | 48.98 |
> | LFA [15] | ICCV 2023 | ResNet50 | Prompt-tuning CLIP | 66.87 | 73.01 | 69.81 | 49.23 | 48.12 | 48.67 |
> | ProGrad [16] | ICCV 2023 | ResNet50 |Prompt-tuning CLIP | 65.23 | 69.39 | 67.25 | 49.14 | 47.36 | 48.23 |
> | | | | | | | | | | |
> | CLIP [11] | ICML 2021 | ViT-B/16 | Zero-shot CLIP | 73.19 | 75.29 | 74.23 | 73.19 | 75.29 | 74.23  |
> | CoOp [12] | IJCV 2022 | ViT-B/16 | Prompt-tuning CLIP | 64.95 | 68.87 | 66.85 |  49.98 | 45.40 | 47.58 |
> | CoCoOp [13] | CVPR 2022 | ViT-B/16 | Prompt-tuning CLIP | 65.60 | 69.50 | 67.49 | 49.35 | 47.84 | 48.58 |
> | KgCoOp [14] | CVPR 2023 | ViT-B/16 | Prompt-tuning CLIP | 71.22 | 74.74 | 72.94 | 51.21 | 50.99 | 51.10 |
> | LFA [15] | ICCV 2023 | ViT-B/16 | Prompt-tuning CLIP | 71.56 | 74.98 | 73.23 | 52.04 | 49.65 | 50.82 |
> ProGrad [16] | ICCV 2023 | ViT-B/16 | Prompt-tuning CLIP | 70.81 | 73.27 | 72.02 | 51.48 | 48.02 | 49.69 |
> | MaPLe [17] | CVPR 2023 | ViT-B/16 | Prompt-tuning CLIP | 60.31 | 68.94 | 64.34 | 48.04 | 49.47 | 48.74 |
> | | | | | | | | | | |
>
> The above table shows the experimental results of VLMs on MetaCoCo, where base and new represent the accuracy of the base and new classes, respectively. HM is the harmonic mean of accuracy on base and new data. From the above table, we find two interesting phenomenons: (1) The more samples are used in the model prompt-tuning phase, the worse the performance of the model will be. For example, the performance drops from 62.40% with 1-shot to 43.98% with 16-shot in CoOp with ResNet50. This further indicates that the more samples used, the greater the shifts in spurious correlations introduced, resulting in reduced model performance. (2) The performance of prompt-tuning methods is lower than zero-shot CLIP in all settings. The possible reason is that prompt-tuning methods overfit to spurious information, therefore, affecting the generalization ability of the model.

---

> ### Author Response · Authors · 2023-11-21
> **Responses by Authors (Part 3): Experimental details including backbone architectures and parameter tuning**
>
> > **[W5] The network structure of ResNet-50 is not explained in EXPERIMENTAL SETUP, it is suggested that the author explain it in backbone architectures.**
>
> **Response:** We thank the useful suggestions from the reviewer. For **network structure,** we follow [18] to implement ResNet50 using 50 layers, including 48 convolutional layers and 2 fully connected layers, with the detailed network structure summarized in below.
>
> | Layer name | output size | ResNet50 |
> |:---------| :---------: | :---------: |
> | conv1 | 112x112 | 7x7, 64, stride 2 |
> | conv2_x | 56x56 | 3x3 max pool, stride 2 |
> | conv2_x | 56x56 | 1x1, 64 |
> | conv2_x | 56x56 | 3x3, 64 |
> | conv2_x | 56x56 | 1x1, 256 |
> | conv3_x | 28x28 | 1x1, 128 |
> | conv3_x | 28x28 | 3x3, 128 |
> | conv3_x | 28x28 | 1x1, 512 |
> | conv4_x | 14x14 | 1x1, 256 |
> | conv4_x | 14x14 | 3x3, 256 |
> | conv4_x | 14x14 | 1x1, 1024 |
> | conv5_x | 7x7 | 1x1, 512 |
> | conv5_x | 7x7 | 3x3, 512 |
> | conv5_x | 7x7 | 1x1, 2048 |
> | | 1x1 | average pool, 1000-d fc, softmax |
> | | | |
>
> Specifically, each layer in conv2_x and conv5_x are repeated with 3 times, each layer in conv3_x are repeated with 4 times, and each layer in conv4_x are repeated with 6 times. The building block of ResNet50 is the residual block, which typically contains two convolutional layers along with the skip connection. The network architecture includes global average pooling and a softmax layer for classification in the final layers.
>
> For all experiments, we use the A100 80G GPU as the computational resource. We tune batch size in $\{128, 256, 512\}$. For learning rate, optimizer, and weight decay, **we follow [20,21] as a comprehensive library for few-shot learning to implement using fixed parameter combinations as summarized in below.**
>
> | Method | Conference | Backbone | Type | Learning Rate | Optimizer | Decay |
> |:---------| :---------: | :---------: | :---------: | :---------: | :---------: | :---------: |
> | MoCo [20] | CVPR 2020 | ResNet50 | Self-supervised learning | 0.03 | SGD | Cosine |
> | SimCLR [21] | ICML 2020 | ResNet50 |  Self-supervised learning | 0.001 |  Adam | Cosine |
> | | | | | | | |
>
> **For ease of reproducibility, we also added Table 4 in Appendix C showing the optimal parameter combinations when implementing the other 17 methods using ResNet12 as the backbone model following [19].**
>
> ***
>
> **We hope the above discussion will fully address your concerns about our work, and we would really appreciate it if you could be generous in raising your score.** We look forward to your insightful and constructive responses to further help us improve the quality of our work. Thank you!
>
> ***
>
> > **Reference**
>
> [1] Haotian Wang et al. Out-of-distribution generalization with causal feature separation. TKDE 2023.
>
> [2] Xingxuan Zhang et al. Deep Stable Learning for Out Of Distribution Generalization. CVPR 2021.
>
> [3] Mingjun Xu et al. Multi-view Adversarial Discriminator: Mine the Non-causal Factors for Object Detection in Unseen Domains. CVPR 2023.
>
> [4] Tianjun Ke et al. Revisiting Logistic-softmax Likelihood in Bayesian Meta-learning for Few-shot Classification. NeurIPS 2023.
>
> [5] Jiangtao Xie et al. Joint Distribution Matters: Deep Brownian Distance Covariance for Few-Shot Classification. CVPR 2022.
>
> [6] Ji Zhang et al. DETA: Denoised Task Adaptation for Few-Shot Learning. ICCV 2023.
>
> [7] Long Tian et al. Prototypes-oriented Transductive Few-shot Learning with Conditional Transport. ICCV 2023.
>
> [8] Wertheimer Davis et al. Few-shot classification with feature map reconstruction networks. CVPR 2021.
>
> [9] Zhanyuan Yang et al. Few-shot classification with contrastive learning. ECCV 2022.
>
> [10] Heng Wang et al. Revisit Finetuning strategy for Few-Shot Learning to Transfer the Emdeddings. ICLR 2023.
>
> [11] Radford Alec et al. Learning transferable visual models from natural language supervision. ICML 2021.
>
> [12] Kaiyang Zhou et al. Learning to prompt for vision-language models. IJCV 2022.
>
> [13] Kaiyang Zhou et al. Conditional prompt learning for vision-language models. CVPR 2022.
>
> [14] Haotao Yao et al. Visual-Language Prompt Tuning with Knowledge-guided Context Optimization. CVPR 2023.
>
> [15] Yassine Ouali et al. Black Box Few-Shot Adaptation for Vision-Language models. ICCV 2023.
>
> [16] Beier Zhu et al. Prompt-aligned Gradient for Prompt Tuning. ICCV 2023.
>
> [17] Muhammad Uzair Khattak et al. MaPLe: Multi-modal Prompt Learning. CVPR 2023.
>
> [18] Kaiming He et al. Deep Residual Learning for Image Recognition. CVPR 2016.
>
> [19] Wenbin Li et al. Libfewshot: A comprehensive library for few-shot learning. TPAMI 2023.
>
> [20] Kaiming He et al. Momentum contrast for unsupervised visual representation learning. CVPR 2020.
>
> [21] Ting Chen et al. A simple framework for contrastive learning of visual representation. ICML 2020.

---

> > ### Author Response · Authors · 2023-11-22
> > **Thanks for your detailed and insightful comments, and we would like to address your concerns and questions.**
> >
> > We thank the reviewer for the detailed and insightful comments. We kindly remind you that the author-reviewer discussion period is about to close and we hope that the response in the rebuttal could address all your concerns. **Specifically, we added a concise and clear contribution summary of our paper (in Introduction, Section 1), more experiments comparing 9 most recent few-shot classification methods on the proposed MetaCoCo (in Table 2, Section 5), and further clarified our experimental details in terms of the backbone architectures and parameter tunings (in Appendix C).** We hope the above discussion will fully address your concerns about our work.
> >
> > We are anticipating any further advice, inquiries or remaining concerns you may have. We will make our best effort to handle them. Thank you!

---

> ### Comment · Reviewer_xD6Q · 2023-11-22
> **Thanks for the replies!**
>
> The replies are very detailed and mostly addressed my concerns. I've raised my score to 8.

---

> > ### Author Response · Authors · 2023-11-22
> > **Thank you for your constructive comments and raising the score!**
> >
> > We are glad to know that your concerns have been effectively addressed. We are very grateful for your constructive comments and questions, which helped improve the clarity and quality of our paper. Thanks again!

---

### Official Review · Reviewer_QLBK · 2023-10-31

**Soundness:** 2 fair
**Presentation:** 2 fair
**Contribution:** 2 fair
**Rating:** 5
**Confidence:** 2

**Summary:**

This paper presents a novel dataset, called METACoCo. The dataset targets at one research void that existing public datasets cannot evaluate – the spurious-correlation problem in few-shot classification. One metric is proposed to evaluate the spurious-correlation shifts. Experiments are conducted to evaluate the performance of existing few-shot classification models etc. The results shows that the evaluated models degrade significantly in the presence of spurious-correlation shifts.

**Strengths:**

The main contribution of this paper is the created dataset. The dataset enables researchers to study spurious-correlation problem in few shot learning. The conducted experimental analysis is persuasive.

**Weaknesses:**

One of my concerns is the technical contribution of this work. Although this paper points out spurious-correlation that have not been well addressed by existing methods and a dataset is proposed, the technical contribution of this work would be more significant if a model can be provided to address the problem.

**Questions:**

What is the difference between spurious-correlation studied in this paper and overfitting in few shot learning? Can we view spurious-correlation as one type of model overfitting? If this is true, it would be interesting to have a comprehensive study of overfitting problem in few-shot learning, in addition to the spurious correlation studied in this paper.

---

> ### Author Response · Authors · 2023-11-21
> **Responses by Authors (Part 1): Added contribution summary and experiments comparing methods for addressing the overfitting problem**
>
> We sincerely appreciate the reviewer’s great efforts and insightful comments to improve our manuscript. In below, we address these concerns point by point and try our best to update the manuscript accordingly.
>
> > **[W1] The technical contribution of this work would be more significant if a model can be provided to address the problem.**
>
> **Response:** We thank the reviewer for pointing out this issue. Nonetheless, we kindly remind our paper is submitted to the **“datasets and benchmarks” track** in ICLR 24, and we agree with the reviewer that **the main contribution is the proposed large-scale few-shot classification benchmark collected from real-world scenarios, rather than a specific method.** To the best of our knowledge, **this is the first benchmark for studying the spurious-correlation shifts in few-shot classification.** Moreover, we further propose a novel metric by using a pre-trained vision-language model to measure the extent of spurious correlations.
>
> To make our contributions more recognizable, **we have added a concise and clear contribution summary of our paper in the “Introduction” section.** In a nutshell, our main contributions can be summarized as follows:
>
> -	We propose a large-scale few-shot classification benchmark with spurious-correlation shifts arising from various contexts in real-world scenarios, named MetaCoCo, in which each class (or concept) consists of different backgrounds (or contexts).
>
> -	We further propose a metric by using a pre-trained vision-language model to quantify
> and compare the extent of spurious correlations on MetaCoCo and other benchmarks.
>
> -	We conduct extensive experiments on MetaCoCo to evaluate and compare most recent methods in few-shot classification, domain shifts, overfitting, and self-supervised learning.
>
>
> > **[W2] What is the difference between spurious-correlation studied in this paper and overfitting in few shot learning?**
>
> **Response:** Thank you for raising such an interesting and insightful concern. In our opinion, we consider overfitting to be a broader concept than spurious-correlation, that is, **spurious-correlation in the training data can cause model overfitting when naively using empirical risk minimization, but not necessary.** In fact, **many other factors can also lead to model overfitting, such as long-tailed data [1], noisy labels [2], cross-domain shifts [3, 4], etc.**
>
> > **[W3] Can we view spurious-correlation as one type of model overfitting?**
>
> **Response:** Yes, please kindly refer to our response to W2 that **the spurious-correlation problem is one of the causes resulting in model overfitting.**
>
> > **[W4] If this is true, it would be interesting to have a comprehensive study of overfitting problem in few-shot learning, in addition to the spurious correlation studied in this paper.**
>
> **Response:** As suggested by the reviewer, **we added a comprehensive study comparing 9 most recent methods for addressing the overfitting problem** in few-shot learning on MetaCoCo. The experimental results are shown as below.
>
> |Method | Conference | Backbone | Type | GL | LL | TT | 1-shot | 5-shot |
> |:---------| :---------: | :---------: | :---------: | :---------: | :---------: | :---------: | :---------: | :---------: |
> | MTL [5] | CVPR 2019 | ResNet12 | Meta | &check; | &check; | &check; | 44.23 | 58.04 |
> |ProtoNet [6] | NeurIPS 2017 | ResNet12 |Metric |  | &check; | &check; | 42.69 | 59.50 |
> | BF3S [7] | ICCV 2019 | WRN-28-10 | Metric | | &check; | &check; | 43.78 | 57.64 |
> | FEAT [8] | CVPR 2020 | ResNet18 | Metric | &check; | &check; | &check; | 50.23 | 64.12 |
> |DSN [9] | CVPR 2020 | ResNet12 | Metric | | &check; | &check; | 46.21 | 58.01 |
> | SVM with DC [10]| TPAMI 2021 | ResNet18 | Metric | &check; | &check; | &check; | 43.56 | 58.96 |
> | SMKD with Prototype [11] | CVPR 2023 | ViT-S | Metric | &check; | &check; | &check; | 56.89 | 69.54 |
> | FGFL [12] | ICCV 2023 | ResNet12 | Metric |  | &check; | &check; | 46.78 | 64.32 |
> |LP-FT-FB [13] | ICLR 2023 | ResNet12 | Fine-tuning | &check; | &check; | &check; | 56.21 | 70.21 |
> |  |  |  |  |  |  |  |  |
>
> From the table, we find that these methods do not show excellent performance on MetaCoCo with spurious-correlation. One possible reason is that **these methods aim to address the overfitting problem of base classes in the few-shot training phase,** but not for the spurious-correlation problem, **making the trained model tend to overfit to the context information,** *e.g.*, **background knowledge.**
>
> ***
> **We hope the above discussion will fully address your concerns about our work, and we would really appreciate it if you could be generous in raising your score.** We look forward to your insightful and constructive responses to further help us improve the quality of our work. Thank you!

---

> > ### Author Response · Authors · 2023-11-21
> > **Responses by Authors (Part 2): References**
> >
> > > **References**
> >
> > [1] Zhisheng Zhong et al. Improving calibration for long-tailed recognition. CVPR 2021.
> >
> > [2] Yingbin Bai et al. Understanding and Improving Early Stopping for Learning with Noisy Labels. NeurIPS 2021.
> >
> > [3] Eleni Triantafillou et al. Meta-dataset: A dataset of datasets for learning to learn from few examples. International Conference on Learning Representations. ICLR 2020.
> >
> > [4] Yong Feng et al. Globally localized multisource domain adaptation for cross-domain fault diagnosis with category shift. TNNLS 2021.
> >
> > [5] Qianru Sun et al. Meta-transfer learning for few-shot learning. CVPR 2019.
> >
> > [6] Jake Snell et al. Prototypical Networks for Few-shot Learning. NeurIPS 2017.
> >
> > [7] Spyros Gidaris et al. Boosting Few-Shot Visual Learning With Self-Supervision. ICCV 2019.
> >
> > [8] Hanjia Ye et al. Few-Shot Learning via Embedding Adaptation with Set-to-Set Functions. CVPR 2020.
> >
> > [9] Christian Simon et al. Adaptive Subspaces for Few-Shot Learning. CVPR 2020.
> >
> > [10] Shuo Yang et al. Bridging the Gap Between Few-Shot and Many-Shot Learning via Distribution Calibration. TPAMI 2021.
> >
> > [11] Han Lin et al. Supervised Masked Knowledge Distillation for Few-Shot Transformers. CVPR 2023.
> >
> > [12] Hao Cheng et al. Frequency Guidance Matters in Few-Shot Learning. ICCV 2023.
> >
> > [13] Heng Wang et al. Revisit Finetuning strategy for Few-Shot Learning to Transfer the Emdeddings. ICLR 2023.

---

> ### Author Response · Authors · 2023-11-22
> **Thanks for your detailed and insightful comments, and we would like to address your concerns and questions.**
>
> We thank the reviewer for the detailed and insightful comments. We kindly remind you that the author-reviewer discussion period is about to close and we hope that the response in the rebuttal could address all your concerns. Specifically, we added **a concise and clear contribution summary** of our paper (in Introduction, Section 1) and **a comprehensive study comparing the 9 most recent methods for addressing the overfitting problem** in few-shot learning on MetaCoCo (in Appendix D) to better address your concerns. We also added extensive experiments comparing the **9 most recent few-shot classification methods** on MetaCoCo (in Table 2, Section 5), as well as the **7 most recent vision-language pre-trained models (VLMs)** using ResNet50 and ViT-B/16 as the vision backbone, and **further clarified our experimental details** in terms of the backbone architectures and parameter tunings (in Appendix C). We hope the above discussion will fully address your concerns about our work.
>
> We are anticipating any further advice, inquiries or remaining concerns you may have. We will make our best effort to handle them. Thank you!

---

> > ### Author Response · Authors · 2023-11-22
> > **We are looking forward to your response on the last day of the authors-reviewer discussion period!**
> >
> > We thank the reviewer for the detailed and insightful comments. During the rebuttal period, we tried our best to address your current concerns by conducting a comprehensive study on MetaCoCo and revising our manuscript. **We sincerely hope the aforementioned changes can fully address your current concerns about our work, and we would really appreciate it if you could be generous in raising your score.** We look forward to your insightful and constructive responses to further help us improve the quality of our work. Thank you!

---

> > > ### Author Response · Authors · 2023-11-23
> > > **We are still expecting to have your response in the final author-reviewer discussion phase!**
> > >
> > > Dear Reviewer QLBK,
> > >
> > > We sincerely appreciate your time and efforts in the review and discussion phases. We have tried our best to respond to all your questions and concerns in detail. **As the author-reviewer discussion phase is only one hour left, if our responses have addressed your remaining concerns, we would really appreciate it if you could be generous in raising your score**; if not, we look forward to your further insightful and constructive responses to further help us improve the quality of our work. Thank you!
> > >
> > > Best regards,
> > >
> > > Submission7015 Authors.

---

### Official Review · Reviewer_F7nA · 2023-11-01

**Soundness:** 3 good
**Presentation:** 3 good
**Contribution:** 3 good
**Rating:** 8
**Confidence:** 4

**Summary:**

Out-of-distribution problems in few-shot classification are important and challenging, which considerably degrades the performance of deep learning models deployed in real-world applications. However, although cross-domain shifts and spurious-correlation shifts are two important real-world distributions, the lack of spurious dataset has left a gap in this important research area. In this paper, the authors present a new benchmark to accelerate the development of the spurious-correlation few-shot classification problem. Overall, I think it is important to study the spurious-correlation few-shot classification and the authors have done a great job to this area.

**Strengths:**

S1: The paper is well-motivated to propose a benchmark with spurious-correlation shifts collected from real-world scenarios. The presentation of this paper is fairly clear, and the details of the dataset construction are also explained thoroughly.

S2: The authors further proposed to use of a pre-training vision-language model CLIP for measuring the quantifying and comparing the extent of spurious correlations on MetaCoCo and other FSC benchmarks, which is quite impressive.

S3: Extensive experiments are conducted to investigate the ability of current methods for addressing real-world spurious correlations, with the baselines including a wide range of existing representative approaches.

**Weaknesses:**

W1: Currently, many vision-language pre-training models, such as CLIP [1] and CoOp [2], evaluate the performance in few-shot setting. I hope to see the performance of pre-trained large models like CLIP on the MetaCoCo with spurious-correlation shifts.

[1] Radford, Alec et al. Learning transferable visual models from natural language supervision, ICML.

[2] Zhou, Kaiyang and Yang, Jingkang and Loy, Chen Change and Liu, Ziwei, Learning to prompt for vision-language models, IJCV.

**Questions:**

See Weaknesses.

---

> ### Author Response · Authors · 2023-11-21
> **Responses by Authors: Added experiments for evaluating the vision-language pre-trained models**
>
> We sincerely appreciate the reviewer’s great efforts and insightful comments to improve our manuscript. In below, we address these concerns point by point and try our best to update the manuscript accordingly.
>
> > **[W1] The performance of pre-trained large models like CLIP on the proposed MetaCoCo dataset.**
>
> **Response:** We thank the reviewer for pointing out this issue. **We added extensive experiments for evaluating the performance of vision-language pre-trained models (VLMs) on our proposed MetaCoCo dataset.** These VLMs are broadly categorized into zero-shot CLIP and prompt-tuning CLIP. We follow the original papers to implement using ResNet50 and ViT-B/16 as backbones with 1-shot and 16-shot samples, and the experimental results are shown as below.
>
> | Method | Conference | Backbone | Type | Base (1-shot) | New (1-shot) | HM (1-shot) | Base (16-shot) | New (16-shot) | HM (16-shot) |
> |:---------| :---------: | :---------: | :---------: | :---------: | :---------: | :---------: | :---------: | :---------: | :---------: |
> | CLIP [1] | ICML 2021 | ResNet50 | Zero-shot CLIP | 68.12 | 72.19 | 70.10 | 68.12 | 72.19 | 70.10 |
> | CoOp [2] | IJCV 2022 | ResNet50 | Prompt-tuning CLIP | 61.22 | 63.62 | 62.40 | 47.55 | 40.90 | 43.98 |
> | CoCoOp [3] | CVPR 2022 | ResNet50 | Prompt-tuning CLIP | 59.86 | 63.88 | 61.80 | 36.88 | 46.10 | 40.98 |
> | KgCoOp [4] | CVPR 2023 | ResNet50 | Prompt-tuning CLIP | 67.95 | 72.05 | 69.94 | 49.18 | 48.78 | 48.98 |
> | LFA [5] | ICCV 2023 | ResNet50 | Prompt-tuning CLIP | 66.87 | 73.01 | 69.81 | 49.23 | 48.12 | 48.67 |
> | ProGrad [6] | ICCV 2023 | ResNet50 |Prompt-tuning CLIP | 65.23 | 69.39 | 67.25 | 49.14 | 47.36 | 48.23 |
> || |  |  | | | | | | |
> | CLIP [1] | ICML 2021 | ViT-B/16 | Zero-shot CLIP | 73.19 | 75.29 | 74.23 | 73.19 | 75.29 | 74.23  |
> | CoOp [2] | IJCV 2022 | ViT-B/16 | Prompt-tuning CLIP | 64.95 | 68.87 | 66.85 |  49.98 | 45.40 | 47.58 |
> | CoCoOp [3] | CVPR 2022 | ViT-B/16 | Prompt-tuning CLIP | 65.60 | 69.50 | 67.49 | 49.35 | 47.84 | 48.58 |
> | KgCoOp [4] | CVPR 2023 | ViT-B/16 | Prompt-tuning CLIP | 71.22 | 74.74 | 72.94 | 51.21 | 50.99 | 51.10 |
> | LFA [5] | ICCV 2023 | ViT-B/16 | Prompt-tuning CLIP | 71.56 | 74.98 | 73.23 | 52.04 | 49.65 | 50.82 |
> ProGrad [6] | ICCV 2023 | ViT-B/16 | Prompt-tuning CLIP | 70.81 | 73.27 | 72.02 | 51.48 | 48.02 | 49.69 |
> | MaPLe [7] | CVPR 2023 | ViT-B/16 | Prompt-tuning CLIP | 60.31 | 68.94 | 64.34 | 48.04 | 49.47 | 48.74 |
> || |  |  | | | | | | |
>
>
> In the above table, “base” and “new” represent the accuracy of the base and new classes, respectively, and HM is the harmonic mean of accuracy on “base” and “new” data. From the table, we find two interesting phenomenons: (1) The more samples are used in the model prompt-tuning phase, the worse the performance of the model will be. For example, the performance drops from 62.40% with 1-shot to 43.98% with 16-shot in CoOp with ResNet50. This further indicates that the more samples used, the greater the shifts in spurious correlations introduced, resulting in reduced model performance. (2) The performance of prompt-tuning methods is lower than zero-shot CLIP in all settings. The possible reason is that prompt-tuning methods overfit to spurious information, therefore, affecting the generalization ability of the model.
>
> ***
> **We hope the above discussion will fully address your concerns about our work.** We look forward to your insightful and constructive responses to further help us improve the quality of our work. Thank you!
> ***
>
> > **References**
>
> [1] Radford Alec et al. Learning transferable visual models from natural language supervision. ICML 2021.
>
> [2] Kaiyang Zhou et al. Learning to prompt for vision-language models. IJCV 2022.
>
> [3] Kaiyang Zhou et al. Conditional prompt learning for vision-language models. CVPR 2022.
>
> [4] Haotao Yao et al. Visual-Language Prompt Tuning with Knowledge-guided Context Optimization. CVPR 2023.
>
> [5] Yassine Ouali et al. Black Box Few-Shot Adaptation for Vision-Language models. ICCV 2023.
>
> [6] Beier Zhu et al. Prompt-aligned Gradient for Prompt Tuning. ICCV 2023.
>
> [7] Muhammad Uzair Khattak et al. MaPLe: Multi-modal Prompt Learning. CVPR 2023.

---

> ### Author Response · Authors · 2023-11-22
> **Thanks for your detailed and insightful comments, and we would like to address your concerns and questions.**
>
> We thank the reviewer for the detailed and insightful comments. We kindly remind you that the author-reviewer discussion period is about to close and we hope that the response in the rebuttal could address all your concerns. **Specifically, we added experiments comparing 7 most recent vision-language pre-trained models (VLMs) using ResNet50 and ViT-B/16 as the vision backbone, respectively.** We hope the above discussion will fully address your concerns about our work.
>
> We are anticipating any further advice, inquiries or remaining concerns you may have. We will make our best effort to handle them. Thank you!

---

> > ### Comment · Reviewer_F7nA · 2023-11-22
> > **Thanks for the authors' replies, and my positive recommendation remains unchanged.**
> >
> > I would like to thank the authors for their great efforts during the rebuttal period, the added comparison experiments with VLMs enhanced the quality of this paper. My positive recommendation remains unchanged.

---

> ### Author Response · Authors · 2023-11-22
> **Thank you for your positive recommendation!**
>
> We are glad to know that your concerns have been effectively addressed. We are very grateful for your constructive comments and questions, which helped improve the clarity and quality of our paper. Thanks again!

---

### Author Response · Authors · 2023-11-21
**General responses and manuscript revision summary**

Dear reviewers and AC,

We sincerely thank all reviewers and AC for their great effort and constructive comments on our manuscript.

As reviewers highlighted, we believe our paper tackles an interesting and important problem (**Reviewer F7nA**, **Reviewer QLBK**, **Reviewer xD6Q**), providing a novel and effective (**Reviewer F7nA**, **Reviewer QLBK**, **Reviewer xD6Q**) benchmark for spurious-correlation few-shot classification, validated with extensive experiments (**Reviewer F7nA**, **Reviewer QLBK**, **Reviewer xD6Q**) with a clear presentation (**Reviewer F7nA**, **Reviewer xD6Q**).

Moreover, we thank the reviewers for pointing out the connection between spurious-correlation and overfitting (**Reviewer QLBK**), as well as for the suggestions for comparing most recent vision-language pre-training models (**Reviewer F7nA**) and few-shot classification methods (**Reviewer xD6Q**). In response to these comments, we have carefully revised and enhanced our manuscript with the following addition discussions and extensive experiments:

- [Reviewer xD6Q] We add **a concise and clear contribution summary** of our paper (in Introduction, Section 1).
- [Reviewer xD6Q] We add experiments comparing **9 most recent few-shot classification methods** on the proposed MetaCoCo (in Table 2, Section 5).
- [Reviewer xD6Q] We clarify our experimental details in terms of the **backbone architectures** and **parameter tunings** (in Appendix C).
- [Reviewer QLBK] We add comprehensive studies comparing **9 methods for addressing overfitting problem** in few-shot learning on MetaCoCo (in Appendix D).
- [Reviewer F7nA] We add experiments comparing **7 most recent vision-language pre-trained models (VLMs)** using ResNet50 and ViT-B/16 as the vision backbone, respectively (in Appendix D).

These updates are temporarily highlighted in "$\textcolor{purple}{purple}$" for facilitating checking.

We hope our response and revision could address all the reviewers' concerns, and are more than eager to have further discussions with the reviewers in response to these revisions.

Thanks,

Submission7015 Authors.

---

### Meta-Review · Area_Chair_7wvD · 2023-12-07

**Metareview:**

The submission presents a new, large-scale, diverse, and realistic benchmark for spurious-correlation in few-shot classification called MetaCoCo. The authors evaluate several state-of-the-art methods and demonstrate that the performance of these approaches degrades significantly in the presence of spurious correlation shifts.

Reviewers note the submission's clarity (F7nA, xD6Q) and its extensive and convincing experiments and baselines (F7nA, QLBK, xD6Q). They find the problem studied to be relevant and well-motivated (F7nA, xD6Q). Reviewer F7nA also notes the submission's significant contribution towards quantifying spurious-correlation shifts in the benchmark. Concerns over the lack of evaluation of VLMs (F7nA), the clarity of the statement of contributions (xD6Q), and the clarity of the explanation of spurious-correlation few-shot classification (xD6Q) have been addressed to the reviewers' satisfaction. From my vantage point, Reviewer QLBK's concern over the lack of technical innovation towards solving the problem of spurious correlations is also adequately addressed by the authors in their response: the ICLR call for papers explicitly welcomes datasets and benchmarks contributions, and given the submission's extensive contributions in that regard, it meets the bar for acceptance in terms if significance.

Given all the above, I recommend accepting the submission.

**Justification For Why Not Higher Score:**

The paper is acceptable but not comparable in quality and potential impact to the benchmark-track papers selected for oral presentations.

**Justification For Why Not Lower Score:**

The submission makes an important benchmarking contribution along the line of benchmarks that highlight failure modes in few-shot classifiers, and it has a strong potential impact in terms of driving the field forward.

---

### Decision · Program_Chairs · 2024-01-16

Accept (poster)